# Advancing LLM Reasoning with Natural Language and Numerical Feedback

Xiaoying Zhang[*,†,‡,1,4] Yipeng Zhang[*1] Hao Sun[2] Kaituo Feng[3] Chaochao Lu[4]
Chao Yang[‡,4] Helen Meng[‡,1]

## Abstract

Recent advances in reinforcement learning (RL) using numerical rewards have significantly enhanced the complex reasoning capabilities of large language models (LLMs). However, we identify three fundamental limitations of purely numerical feedback: performance plateaus, ineffective spontaneous self-reflection, and persistent failures. We show that plateaued RL models can successfully refine failed solutions when given natural language critiques. Motivated by this, we propose Critique-GRPO, an online RL framework that integrates both natural language and numerical feedback for policy optimization. This approach enables LLMs to learn simultaneously from initial responses and critique-guided refinements, effectively internalizing the exploration benefits of both stages. Extensive experiments show that Critique-GRPO outperforms all compared supervised and RL-based fine-tuning methods, achieving average Pass@1 improvements of approximately +15.0-21.6% on various Qwen models and +7.3% on Llama-3.2-3B-Instruct across eight challenging reasoning tasks. Notably, Critique-GRPO facilitates effective self-improvement through self-critiquing, achieving substantial gains over GRPO, *e.g.,* a +16.7% Pass@1 improvement on AIME 2024. The code and models are released at: https://github.com/zhangxy-2019/critique-GRPO

## 1. Introduction

RL has been a key driver of recent advancements in improving the reasoning capabilities of LLMs (Yang et al., 2025a; DeepSeek-AI et al., 2025; OpenAI et al., 2024; OpenAI, 2025). Specifically, online RL with numerical feedback (*e.g.,* scalar rewards)—exemplified by the R1-Zero paradigm (DeepSeek-AI et al., 2025)—enables LLMs to learn through trial-and-error (Silver & Sutton, 2025), yielding substantial improvements.

Despite its success, we identify three key limitations of RL with solely numerical feedback (Section 3): (i) *Performance Plateaus:* Performance stagnates despite $8\times$ training data scaling or extended training. (ii) *Ineffective Spontaneous Self-Reflection:* Spontaneous "Aha moments" (DeepSeek-AI et al., 2025) rarely improve problem-solving success. (iii) *Persistent Failures:* Models consistently fail on specific training problems despite extensive iterations. While dense intermediate rewards (Cui et al., 2025a; Feng et al., 2025) can partially mitigate plateaus and persistent failures through fine-grained credit assignment (Sutton, 1984), they remain insufficient for rectifying ineffective self-reflection. Crucially, numerical feedback inherently lacks the expressivity to explain *why* a response fails or *how* to correct it. These limitations highlight the need for richer feedback mechanisms to enable effective RL scaling (Karpathy, 2025).

Natural Language Feedback (NLF) (*e.g.,* textual critiques) offers a promising solution by providing explicit guidance (Saunders et al., 2022). However, existing methods primarily utilize supervised fine-tuning (SFT) to imitate static, pre-collected critiques or critique-guided refinements (Wang et al., 2025; Chen et al., 2024; Xi et al., 2024). While effective for narrow policy alignment, these offline approaches lack the capacity for active exploration and real-time adaptation, which are essential for robust generalization. Crucially, the integration of expressive critiques into online RL loops—to augment purely numerical feedback—remains unexplored. This work addresses that gap by asking: *Can we incorporate critiques into an online RL framework to enable LLMs to learn simultaneously from both natural language and numerical feedback?*

To investigate this, we first show that critiques enable even plateaued RL models to correct persistent failures (Sec-

---

[†]Project Lead. [*]Equal contribution [‡]Corresponding authors. [1]HCCL, The Chinese University of Hong Kong, Hong Kong, China [2]University of Cambridge, Cambridge, United Kingdom [3]MMLab, The Chinese University of Hong Kong, Hong Kong, China [4]Shanghai Artificial Intelligence Laboratory, Shanghai, China. Correspondence to: Xiaoying Zhang <zhangxy@se.cuhk.edu.hk>.

*Proceedings of the $43^{rd}$ International Conference on Machine Learning*, Seoul, South Korea. PMLR 306, 2026. Copyright 2026 by the author(s).

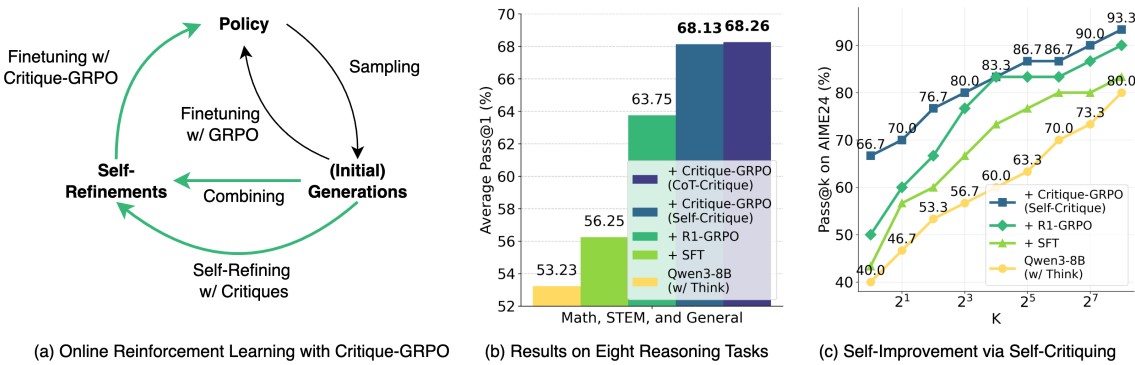

(a) Online Reinforcement Learning with Critique-GRPO    (b) Results on Eight Reasoning Tasks    (c) Self-Improvement via Self-Critiquing

*Figure 1.* (a) Critique-GRPO enhances online reinforcement learning by enabling the model to learn from both initial responses and self-refinements through natural language feedback (critiques), highlighted in green, instead of relying solely on numerical feedback (scalar rewards). (b) Critique-GRPO improves the average Pass@1 score on Qwen3-8B by approximately 4.5% across eight reasoning tasks compared to GRPO. (c) Critique-GRPO facilitates self-improvement via self-critiquing, obtaining 66.7% pass@1 on AIME 2024 and consistent gains across pass@k scores (k=1-256) over GRPO.

tion 3). This efficacy extends from simple indicators to detailed chain-of-thought (CoT) evaluations (Wang et al., 2025). This suggests that verbal credit assignment from critiques enables models to access high-quality refinement trajectories via in-context learning (Liu et al., 2022)—trajectories unreachable via standard exploration with scalar rewards alone. This aligns with theoretical work showing that granular diagnostic signals in informative language feedback can exponentially accelerate learning by reducing search-space complexity (Xu et al., 2025). Building on this insight, we propose Critique-GRPO, a novel online RL framework that synergizes numerical and natural language feedback. As depicted in Figure 1, Critique-GRPO employs a dual mechanism: (i) *learning from standard generation*, where the model generates initial responses via standard exploration; and (ii) *learning from critique-guided refinement*, where the model performs self-refinement via in-context learning based on critiques from model- or rule-based reward systems. This integrates targeted natural language feedback through high-quality refinements while preserving standard exploration. To mitigate entropy explosion and performance degradation arising from significant distribution shifts induced by refinements, we prioritize initial responses and incorporate refinements selectively. Additionally, a shaping function (Yan et al., 2025) reinforces successful, yet unfamiliar refinements while penalizing incorrect attempts.

We evaluate Critique-GRPO using five different models across eight challenging reasoning tasks. Extensive results demonstrate that Critique-GRPO significantly outperforms supervised and RL-based fine-tuning methods, achieving average Pass@1 improvements of +15.0-21.6% on Qwen models (Qwen et al., 2025; Yang et al., 2025a) and +7.3% on Llama-3.2-3B-Instruct (Grattafiori et al., 2024). Notably, the framework is robust to diverse critique sources and ex-

hibits strong self-improvement capabilities; for instance, employing self-critiques yields a +16.7% Pass@1 gain over GRPO on AIME 2024.

In summary, our contributions are three-fold: (i) We identify three key limitations of RL relying solely on numerical feedback and highlight the potential of natural language feedback to address them. (ii) We propose Critique-GRPO, an online RL framework that enables LLMs to learn from both initial responses and their refinements by integrating numerical and natural language feedback. (iii) We validate the efficacy of Critique-GRPO through extensive experiments, demonstrating superior performance across eight challenging reasoning benchmarks.

## 2. Related Work

**Enhancing LLM Reasoning with RL.**    Reinforcement learning with numerical feedback has proven highly effective for improving the reasoning capabilities of LLMs (OpenAI et al., 2024; DeepSeek-AI et al., 2025; Fatemi et al., 2025; Li et al., 2025; Liu et al., 2025a; Yu et al., 2025). However, numerical feedback is inherently limited in expressivity; sparse scalar rewards provide little diagnostic information about where or why errors occur, making it difficult for models to identify and correct specific reasoning failures (Xi et al., 2024; Gandhi et al., 2025). To bridge this gap, recent work has combined online RL with expert demonstrations (Yan et al., 2025; Lv et al., 2025; Lanchantin et al., 2025); however, these approaches are constrained by their reliance on curated, high-quality data. In contrast, Critique-GRPO integrates expressive natural language critiques directly into online RL loops, eliminating the dependency on expert demonstrations. Further discussions on learning from NLF are provided in Appendix A.

# 3. Limitations of RL and the Promise of Natural Language Guidance

## 3.1. Three Limitations of Learning with Solely Numerical Feedback

In this section, we identify three key limitations of RL fine-tuning that relies solely on numerical feedback: (i) *performance plateaus*, (ii) *ineffective spontaneous self-reflection*, and (iii) *persistent failures*. We investigate these limitations using mathematical and STEM reasoning tasks on three models: Qwen2.5-7B-Base (Qwen et al., 2025) (non-reasoning), Qwen3-8B (Yang et al., 2025a) (reasoning), and Qwen3-8B-Base. We summarize key findings for Qwen2.5-7B-Base below; complete analyses appear in Appendices E.

**Performance plateaus despite scaling data and compute.** As shown in Figure 5a (Appendix E), Qwen2.5-7B-Base performance saturates after 120 steps; neither an 8-fold increase in training prompts (4k to 32k) nor extended training yields significant gains.

**Spontaneous self-reflection contributes minimally to problem-solving.** We analyze six cognitive behaviors emerging during RL fine-tuning: (i) planning behaviors (subgoal setting, summarization) and (ii) self-reflection behaviors (verification, backtracking, backward chaining (Gandhi et al., 2025), anticipation). For problems that only the RL-finetuned model solves correctly, we measure each behavior's contribution to success. Figure 6a (Appendix E) shows self-reflection behaviors contribute minimally across both reasoning tasks.

**Persistent failures on substantial training subsets.** As shown in Table 1 (left panel), even the best-performing RL-finetuned Qwen2.5-7B-Base consistently fails on approximately 29% of training questions (Pass@4 = 0), indicating fundamental limitations of numerical-only feedback.

## 3.2. Promise of Learning from Natural Language Feedback

To investigate whether natural language feedback can overcome the identified scaling bottlenecks, we evaluate three critique variants: (i) *Indicative Critique*, which provides only a binary correctness signal; (ii) *Indicative Critique with Ground Truth (w/ GT)*, which includes the final answer but *excludes expert demonstrations*; (iii) *CoT Critique*, a model-generated step-by-step evaluation that concludes with a correctness label, also *without expert demonstrations*. We apply these critiques to the subset of training questions where RL-finetuned models persistently fail (Pass@4 = 0) and prompt them to refine their initial responses. Our key findings, summarized below, demonstrate that even minimal language feedback can catalyze error correction where scalar rewards failed. Detailed analyses, refinement strate-

gies, and examples are available in Appendix E.

**Deliberate critiques outperform spontaneous self-reflection.** All three critique types enable successful refinement of previously unsolvable problems, demonstrating that deliberate critiques succeed where spontaneous self-reflection fails.

---

**Three Types of Critiques**

**Indicative Critique:** The generated solution is incorrect.

**Indicative Critique w/ Ground Truth:** The generated solution is incorrect, the ground truth is $\frac{7}{25}$.

**CoT Critique:** Let's analyze the student's solution step-by-step and identify any errors:
### Step 1: Geometry Understanding ...
### Step 7: Precise Calculation ...
Conclusion: incorrect [END]

---

**CoT critiques enable substantial improvement.** Table 1 shows that CoT critiques achieve the highest valid refinement rate (36.47%) and successfully refine 55.37% of persistently failed questions for Qwen2.5-7B-Base. The effectiveness of CoT critiques can be attributed to their richness, *e.g.,* providing a step-by-step explanation.

# 4. Critique-GRPO

In this section, we introduce Critique-GRPO, an online RL framework built upon Group Relative Policy Optimization (GRPO) (Shao et al., 2024a) that enables LLMs to learn from both natural language and numerical feedback.

## 4.1. From GRPO to Critique-GRPO

GRPO is an online RL algorithm designed for efficient LLM fine-tuning. Unlike Proximal Policy Optimization (PPO) (Schulman et al., 2017), it eliminates the need for a value function approximation by estimating advantages based on the relative performance of a group of sampled responses for the same query. For an LLM policy $\pi_\theta$, GRPO operates as follows: For each query in the dataset $q \in Q$, $n$ responses $\{y^{(i)}\}_{i=1}^{n}$ are sampled from the old policy $\pi_{\text{old}}$ and scored to obtain rewards $\{R^{(i)}\}_{i=1}^{n}$. The training objective is:

$$\mathcal{J}_{\text{GRPO}}(\theta) = \mathbb{E}_{q \sim Q, \{y^{(i)}\}_{i=1}^{n} \sim \pi_{\text{old}}(\cdot|q)} \left[ \frac{1}{n} \sum_{i=1}^{n} \frac{1}{|y^{(i)}|} \sum_{t=1}^{|y^{(i)}|} \right.$$

$$\left. \min \left( r_t^{(i)}(\theta) \hat{A}_t^{(i)}, \text{clip}(r_t^{(i)}(\theta), 1 - \epsilon, 1 + \epsilon) \hat{A}_t^{(i)} \right) \right]$$

$$(1)$$

*Table 1.* Analysis of performance gains from critique-based self-refinement.

| Method | % Failed Questions (Pass@4=0) | Critique Type | % Valid Critiques | % Valid Refinements | % Critiqued Questions | % Questions Refined |
|---|---|---|---|---|---|---|
| RL-finetuned Qwen2.5-7B-Base | 29.07 | Indicative Critique | 100.00 | 2.09 | 100.00 | 7.05 |
| | | Indicative Critique w/ GT | 100.00 | 1.98 | 100.00 | 6.88 |
| | | CoT Critique | 60.06 | **36.47** | 95.10 | **55.37** |

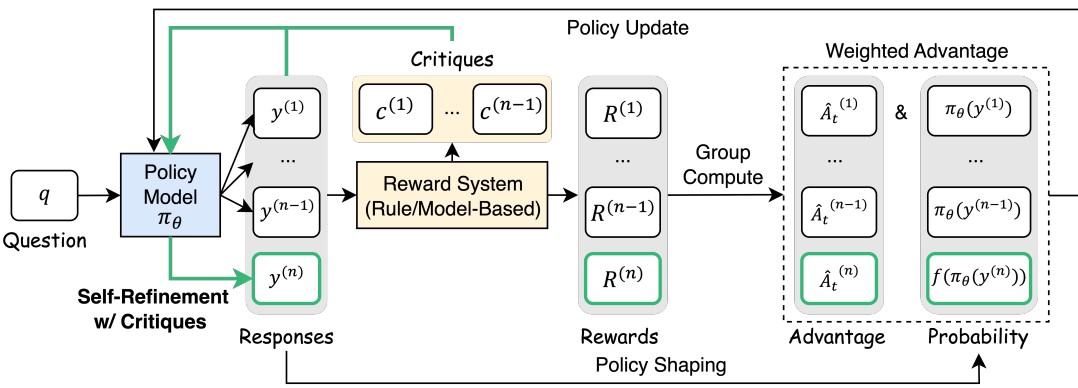

*Figure 2.* Overview of Critique-GRPO. Given a question, Critique-GRPO samples initial responses and then refines these responses via in-context learning, utilizing critiques generated by a reward system (either rule-based or model-based). These refinements are combined with the initial responses to optimize the policy within an online RL framework. A weighted advantage function, combined with policy shaping, emphasizes correct refinements while penalizing incorrect ones.

where $r_t^{(i)}(\theta) = \frac{\pi_\theta(y_t^{(i)}|q,y_{<t}^{(i)})}{\pi_{\text{old}}(y_t^{(i)}|q,y_{<t}^{(i)})}$ is the probability ratio, comparing the current policy $\pi_\theta$ to the old policy $\pi_{\text{old}}$. The advantage $\hat{A}_t^{(i)} = \frac{R^{(i)} - \text{mean}(\{R^{(1)},...,R^{(n)}\})}{\text{std}(\{R^{(1)},...,R^{(n)}\})}$ for all tokens in a response is calculated by normalizing the rewards $\{R^{(i)}\}_{i=1}^n$ using the group mean and standard deviation. The hyperparameter $\epsilon$ sets the clipping range for the probability ratio, preventing overly large policy updates. For simplicity, we omit the KL divergence penalty in Equation 1. Following Liu et al. (2025a), we exclude the length-normalization factor $1/|y^{(i)}|$ and the reward standard deviation $\text{std}(\{R^{(i)}\})$ to avoid biased gradients.

### 4.2. Online RL with Critique-GRPO

We introduce Critique-GRPO, which optimizes $\pi_\theta$ using trajectories from both standard generation and critique-guided refinement. As illustrated in Figure 2, the process consists of three steps:

**Step 1: Initial Response Sampling.** For each query $q \in Q$, we sample $n$ initial responses $\{y^{(i)}\}_{i=1}^n$ from the old policy $\pi_{\text{old}}(\cdot \mid q)$. A reward system evaluates these responses, generating critiques $\{c^{(i)}\}_{i=1}^n$ and scalar rewards $\{R^{(i)}\}_{i=1}^n$, with positive rewards for correct responses and zero reward for incorrect ones: $c^{(i)}, R^{(i)} \leftarrow$ Reward$(q, y^{(i)}), \forall i$. Critique-GRPO supports two types of reward systems: (i) *Rule-based:* Rewards are binary

based on string matching with the ground truth: $R^{(i)} =$ is_equivalent$(y^{(i)}, y_{\text{GT}})$. Based on these evaluations, two indicative critiques are heuristically constructed: one without ground truth ($c_{\text{I}}^{(i)}$) and one with ground truth ($c_{\text{GT}}^{(i)}$): $c_{\text{I}}^{(i)}, c_{\text{GT}}^{(i)} \leftarrow R^{(i)}$. (ii)*Model-based:* A reward model $\pi_{RM}$ generates CoT critiques: $c_{\text{CoT}}^{(i)} \sim \pi_{RM}(\cdot \mid I_c, q, y^{(i)})$, where $I_c$ is the critique instruction. The binary correctness of the critique determines the scalar reward: $R^{(i)} \leftarrow c_{\text{CoT}}^{(i)}$ (detailed in Appendix E).

**Step 2: Critique-Guided Self-Refinement.** To optimize computational efficiency, we initiate refinement only when the initial response set $\{y^{(i)}\}_{i=1}^n$ contains zero correct solutions. We generate refined responses via in-context learning conditioned on the question-response-critique triplet $(q, y^{(i)}, c^{(i)})$ and a refinement instruction $I_{\text{refine}}$ (detailed in Appendix M and Appendix E): $y_{\text{refined}}^{(i)} \sim \pi_{\text{old}}(\cdot \mid I_{\text{refine}}, q, y^{(i)}, c^{(i)})$, where $c^{(i)} \in \{c_{\text{CoT}}^{(i)}, c_{\text{GT}}^{(i)}, c_{\text{I}}^{(i)}\}$. These refinements are evaluated by the reward system to obtain scores $\{R_{\text{refine}}^{(i)}\}_{i=1}^n$. To mitigate entropy explosion from significant distribution shifts induced by refinements, we sample a subset of $k$ refinements $\{y_{\text{refined}}^{(i')}\}_{i'=1}^k$ from the full set, prioritizing correct solutions; if no correct refinements exist, we sample randomly. This subset is combined with the initial responses to form the final training group: $\{y^{(i)}\}_{i=1}^n \cup \{y_{\text{refined}}^{(i')}\}_{i'=1}^k$.

We justify our critique mechanism's sample efficiency via the Transfer Eluder Dimension framework (Xu et al., 2025). By characterizing critiques as *Reward-Informative Feedback* (Cheng et al., 2023), we show they distinguish hypotheses more efficiently than scalar rewards.

**Proposition 4.1** (Complexity Reduction via Critique-Guided Exploration). *Consider a reasoning problem where the goal is to construct a hidden optimal solution sequence of $L$ steps, $a^* = (s_1^*, \ldots, s_L^*)$, with each $s_i \in \mathcal{S}$. The action set is $\mathcal{A} = \bigcup_{k=1}^{L} \mathcal{S}^k$. The hypothesis space is $\mathcal{F}$.*

*Standard Generation (Reward-Only): Binary rewards only indicate if a final state is correct ($R = 1$) or not ($R = 0$). Since incorrect sequences provide identical zero-information, the agent must effectively enumerate the action space $\mathcal{A}$. The Eluder dimension (Russo & Van Roy, 2013) scales exponentially:* $\dim_E(\mathcal{F}) \approx O(|\mathcal{S}|^L)$.

*Critique-Guided Refinement (Informative Feedback): (1) Indicative Feedback ($c_I$, $c_{GT}$): While the worst-case complexity remains $O(|\mathcal{S}|^L)$, the critique acts as a pruning signal. By conditioning on the failure and critique, the policy restricts its search to a subspace $\mathcal{A}_c \subset \mathcal{A}$ consistent with the critique. (2) Constructive Feedback ($c_{CoT}$): If the critique localizes the first error step $t$, the problem decomposes into $L$ independent sub-problems of size $|\mathcal{S}|$, reducing complexity to linear $O(|\mathcal{S}|L)$. If the critique provides the correction suggestion for the first error $s_t^*$, complexity becomes independent of the search space $O(L)$.*

*Consequently, for a fixed computational budget $M$ where $|\mathcal{S}|L \ll M \ll |\mathcal{S}|^L$, critique-guided exploration yields a higher probability of success:* $P(a^* \in \{y_{refined}^{(j)}\}_{j=1}^M) \gg P(a^* \in \{y^{(i)}\}_{i=1}^M)$. *(See Appendix F.4 for the full analysis).*

**Step 3: Online Policy Optimization.** Finally, we fine-tune the model on the mixed set of initial and refined responses using scalar rewards. The training objective is:

$$\mathcal{J}_{\text{Critique-GRPO}}(\theta) = \mathcal{J}_{\text{init}}(\theta) + \mathcal{J}_{\text{refi}}(\theta), \quad (2)$$

The objective for *initial responses* follows the standard GRPO formulation:

$$\mathcal{J}_{\text{init}}(\theta) = \mathbb{E}_{q \sim Q, \{y^{(i)}\}_{i=1}^n \sim \pi_{\text{old}}(\cdot|q)} \left[ \frac{1}{n} \sum_{i=1}^n \sum_{t=1}^{|y^{(i)}|} \right.$$
$$\left. \min \left( r_t^{(i)}(\theta) \hat{A}_t^{(i)}, \text{clip}(r_t^{(i)}(\theta), 1-\epsilon, 1+\epsilon) \hat{A}_t^{(i)} \right) \right] \quad (3)$$

The objective for *refined responses* are presented as follows:

$$\mathcal{J}_{\text{refi}}(\theta) = \mathbb{E}_{q \sim Q, \{y_{\text{refined}}^{(i')}\}_{i'=1}^k \sim \pi_{\text{old}}(\cdot|q)} \left[ \frac{1}{k} \sum_{i'=1}^k \sum_{t=1}^{|y_{\text{refined}}^{(i')}|} \right.$$
$$\left. \min \left( \rho_t^{(i')}(\theta) \hat{A}_t^{(i')}, \text{clip}(\rho_t^{(i')}(\theta), 1-\epsilon, 1+\epsilon) \hat{A}_t^{(i')} \right) \right] \quad (4)$$

The advantages $\hat{A}_t^{(i)}$ and $\hat{A}_t^{(i')}$ are computed using the group mean of rewards from *both* the initial and refined sets to ensure a unified baseline: $\hat{A}_t^{(i/i')} = R^{(i/i')} - \text{mean}(\{R^{(i)}\}_{i=1}^n \cup \{R_{\text{refined}}^{(i')}\}_{i'=1}^k)$. For initial responses, $r_t^{(i)}(\theta)$ is the standard importance sampling ratio. For refined responses, we employ a shaping function to define the policy ratio $\rho_t^{(i')}(\theta)$ (Yan et al., 2025): $\rho_t^{(i')}(\theta) = \frac{\pi_\theta(y_{\text{refined},t}^{(i')}|q, y_{\text{refined},<t}^{(i')})}{\pi_\theta(y_{\text{refined},t}^{(i')}|q, y_{\text{refined},<t}^{(i')}) + \gamma}$ ($0 < \gamma < 1$). The shaping term $\gamma$ reweights gradients to assign higher importance to tokens in the refined responses that are currently low-probability under $\pi_\theta$, facilitating efficient learning from valid but unfamiliar refinements. We remove the KL-divergence penalty to enable substantial policy updates towards these refinements. Detailed analyses are provided in Appendix F.

# 5. Experiments

## 5.1. Experimental Setup

**Datasets and Evaluation Metrics.** We use randomly sampled subsets of 4k examples from a reorganized 46k subset (Yan et al., 2025) of OpenR1-Math-220k (Bakouch et al., 2025) as the training set. For validation, we utilize the curated validation set provided by (Yan et al., 2025). The model is evaluated on five established mathematical reasoning benchmarks: MATH-500 (Hendrycks et al., 2021), Minerva-Math, OlympiadBench (He et al., 2024a), AIME 2024/2025 (Li et al., 2024), and AMC 2023 (Li et al., 2024). To analyze out-of-distribution generalization, we further evaluate the model on three reasoning tasks spanning scientific and general domains: TheoremQA (Chen et al., 2023), GPQA-Diamond, and MMLU-Pro (Wang et al., 2024). For evaluation, we use greedy decoding (temperature = 0) and report pass@1 over three runs.

**Compared Methods.** We compare Critique-GRPO against the following approaches. All differences are considered significant at $p < 0.01$. During RL fine-tuning, we utilize binary scalar rewards (+1 for correct responses and 0 for incorrect ones). *1. Supervised Learning-based Finetuning:* (i) *Supervised Fine-tuning (SFT)*: Finetuning on high-quality annotated training data. (ii) *Reward Ranked Finetuning (RAFT)* (Dong et al., 2023): Finetuning on correct initial responses. (iii) *Refinement Finetuning (Refinement FT)* (Chen et al., 2024): Finetuning on correct refinements generated conditionally on CoT critiques. (iv) *Critique Finetuning (Critique FT)* (Wang et al., 2025): Finetuning on CoT critiques. (v) *Critique-in-the-Loop Finetuning (CITL-FT)* (Xi et al., 2024): Fine-tuning on both correct initial responses and their refinements generated conditionally on CoT critiques. *2. Reinforcement Learning-based Finetuning:* (vi) *R1-GRPO* (DeepSeek-AI et al., 2025): Fine-tuning on initial responses using the GRPO algorithm with binary scalar rewards. (vii) *R1-Dr.GRPO* (Liu et al., 2025a): Fine-tuning

*Table 2.* Evaluation results (Pass@1) on mathematical reasoning (ID) and scientific and general (OOD) reasoning tasks. "Expert Demo." refers to expert demonstrations. "Num. Feedback" and "Lang. Feedback" denote numerical and natural language feedback, respectively. **The natural language feedback provides critique without including expert demonstrations.**

| Method | Supervision | | | Math (ID) | | | | | Science & General (OOD) | | | Avg. |
|---|---|---|---|---|---|---|---|---|---|---|---|---|
| | Expert Demo. | Num. FB | Lang. FB | MATH 500 | Minerva MATH | Olympiad Bench | AMC23 | AIME24 | Theorem QA | GPQA Diamond | MMLU Pro | |
| *Non-Reasoning Model* | | | | | | | | | | | | |
| Qwen2.5-7B-Base | - | - | - | 60.80 | 20.20 | 30.40 | 35.00 | 13.30 | 21.60 | 28.79 | 46.24 | 32.04 |
| *Supervised Learning-based Finetuning* | | | | | | | | | | | | |
| + SFT | ✓ | ✗ | ✗ | 61.60 | 24.30 | 23.40 | 40.00 | 6.70 | 26.50 | 30.30 | 51.49 | 33.04 |
| + RAFT | ✗ | ✓ | ✗ | 67.00 | 19.50 | 32.40 | 50.00 | 10.00 | 24.40 | 23.74 | 47.12 | 34.27 |
| + Refinement FT | ✗ | ✓ | ✓ | 65.80 | 21.30 | 32.10 | 47.50 | 13.30 | 24.40 | 29.80 | 47.51 | 35.21 |
| + Critique FT | ✗ | ✗ | ✓ | 66.00 | 19.10 | 29.30 | 47.50 | 13.3 | 29.60 | 28.79 | 44.46 | 34.76 |
| + CITL-FT | ✗ | ✓ | ✓ | 70.20 | 19.90 | 30.70 | 42.50 | 16.70 | 28.70 | 28.28 | 48.31 | 35.66 |
| *Reinforcement Learning-based Finetuning (w/o SFT)* | | | | | | | | | | | | |
| + R1-GRPO | ✗ | ✓ | ✗ | 74.00 | 32.00 | 38.50 | 42.50 | 16.70 | 40.60 | 33.33 | 51.81 | 41.18 |
| + R1-Dr.GRPO | ✗ | ✓ | ✗ | **78.40** | 34.90 | 39.90 | 40.00 | 13.30 | 43.10 | 38.89 | 52.83 | 42.66 |
| + Critique-GRPO (Ours) (Indicative Critique) | ✗ | ✓ | ✓ | 76.00 | 36.00 | 41.00 | 55.00 | 13.30 | 41.80 | 37.88 | **55.97** | 44.62 |
| + Critique-GRPO (Ours) (Critique w/ GT) | ✗ | ✓ | ✓ | 76.80 | 35.70 | 39.60 | 62.50 | 10.00 | 44.00 | **38.89** | 54.88 | 45.30 |
| + Critique-GRPO (Ours) (CoT Critique) | ✗ | ✓ | ✓ | 77.80 | **36.80** | **42.40** | 62.50 | 20.00 | 44.00 | 37.88 | 55.28 | **47.08** |
| *Reasoning Model (w/ Thinking)* | | | | | | | | | | | | |
| Qwen3-8B | - | - | - | 82.00 | 41.20 | 44.10 | 67.50 | 40.00 | 46.90 | 35.86 | 68.25 | 53.23 |
| + SFT | ✓ | ✗ | ✗ | 83.20 | 43.80 | 46.40 | 82.50 | 40.00 | 48.90 | 38.38 | 66.81 | 56.25 |
| + RAFT | ✗ | ✓ | ✗ | 82.80 | 44.10 | 46.40 | 75.00 | 36.70 | 46.80 | 37.88 | 69.00 | 54.84 |
| + Refinement FT | ✗ | ✓ | ✓ | 87.40 | 46.00 | 54.50 | 80.00 | 40.00 | 55.40 | 45.45 | 66.82 | 59.45 |
| + Critique FT | ✗ | ✗ | ✓ | 84.40 | 37.10 | 49.80 | 80.00 | 36.70 | 46.40 | 35.35 | 64.10 | 54.23 |
| + CITL-FT | ✗ | ✓ | ✓ | 85.00 | 43.00 | 46.80 | 70.00 | 43.30 | 48.00 | 41.92 | 68.73 | 55.84 |
| + R1-GRPO | ✗ | ✓ | ✗ | 91.00 | 52.60 | 65.60 | 82.50 | 50.00 | 57.90 | 40.40 | 70.00 | 63.75 |
| + R1-Dr.GRPO | ✗ | ✓ | ✗ | 91.20 | 51.10 | 63.60 | 82.50 | 53.30 | 59.00 | 44.44 | 70.51 | 64.46 |
| + Critique-GRPO (Ours) (Indicative Critique) | ✗ | ✓ | ✓ | 91.00 | 47.80 | 63.30 | 85.00 | **63.30** | 60.40 | 47.47 | 70.00 | 66.03 |
| + Critique-GRPO (Ours) (Critique w/ GT) | ✗ | ✓ | ✓ | 92.00 | 50.00 | 66.80 | 87.50 | 56.70 | 59.00 | 47.47 | **70.87** | 66.29 |
| + Critique-GRPO (Ours) (CoT Critique) | ✗ | ✓ | ✓ | **92.00** | **52.90** | **66.80** | 92.50 | 63.30 | 60.10 | **47.98** | 70.47 | **68.26** |

on initial responses using the Dr.GRPO algorithm, which removes optimization bias terms, with binary scalar rewards. We implement Critique-GRPO using asynchronous rollouts via the VERL framework (Sheng et al., 2024) for computational efficiency. Implementation details are in Appendix G.

## 5.2. Main Results

Table 2 presents our evaluation results, revealing three key findings:

**Natural language feedback enhances online RL policy optimization.** Critique-GRPO consistently outperforms both supervised and RL-based fine-tuning methods on Qwen2.5-7B-Base and Qwen3-8B across all tasks. Compared to R1-GRPO and R1-Dr.GRPO, Critique-GRPO improves average pass@1 by +4.4 points (42.66% → 47.08%) on Qwen2.5-7B-Base and +3.8 points (64.46% → 68.26%) on Qwen3-8B. See Appendix J for qualitative analysis.

**Online self-refinement outperforms offline approaches.** Critique-GRPO (CoT critique) surpasses Refinement FT by +11.9 points (47.08% vs. 35.21%) and +8.81 points (68.26%

vs. 59.45%) in average pass@1 on Qwen2.5-7B-Base and Qwen3-8B, respectively. It also exceeds CITL-FT by +11.4 points (47.08% vs. 35.66%) on Qwen2.5-7B-Base and +12.4 points (68.26% vs. 55.84%) on Qwen3-8B.

**Richer critiques yield superior refinements and policy optimization.** Critique-GRPO with CoT critiques consistently outperforms its indicative critique variant, achieving average pass@1 improvements of +1.8–2.4% on Qwen2.5-7B-Base and +2.0–2.3% on Qwen3-8B. This advantage arises from CoT critiques' ability to guide more effective refinements, as demonstrated in Section 3. See Appendices B and J for computational cost and qualitative analyses.

## 5.3. Investigation on Math-Centric Backbone Models

We validate Critique-GRPO on the math-centric backbone model Qwen2.5-Math-7B-Base, comparing it against three RL fine-tuning approaches that use only numerical feedback: *SimpleRL-Zero* (Zeng et al., 2025), *PRIME-Zero* (Cui et al., 2025a), and *Oat-Zero* (Liu et al., 2025a) (see Appendix G for details).

*Table 3.* Investigation of RL finetuning with Critique-GRPO on Qwen2.5-Math-7B-Base (Yang et al., 2024). Results marked with an asterisk are cited from (Yan et al., 2025).

| Method | Training Data Volume | Math (ID) | | | | | Science & General (OOD) | | | Avg. |
| --- | --- | --- | --- | --- | --- | --- | --- | --- | --- | --- |
| | | MATH 500 | Minerva MATH | Olympiad Bench | AMC23 | AIME24 | Theorem QA | GPQA Diamond | MMLU Pro | |
| Qwen2.5-Math-7B-Base | - | 51.20 | 13.20 | 17.60 | 47.50 | 13.30 | 26.40 | 26.77 | 39.70 | 29.46 |
| + SimpleRL-Zero* | 46k | 76.00 | 25.00 | 34.70 | 54.90 | 27.00 | - | 23.20 | 34.50 | - |
| + PRIME-Zero* | 46k | 81.40 | 39.00 | 40.30 | 54.00 | 17.00 | - | 18.20 | 32.70 | - |
| + Oat-Zero* | 46k | 78.00 | 34.60 | 43.40 | 61.20 | **33.40** | - | 23.70 | 41.70 | - |
| + Critique-GRPO (Ours) (CoT-Critique) | 4k | **84.20** | **40.10** | **48.60** | **70.00** | 30.00 | **51.40** | **40.40** | **43.79** | **51.06** |

*Table 4.* Potential for self-improvement via self-critiquing using Critique-GRPO on Qwen3-8B.

| Method | Math (ID) | | | | | Science & General (OOD) | | | Avg. | AIME24 (Avg@32) | AIME25 (Avg@32) |
| --- | --- | --- | --- | --- | --- | --- | --- | --- | --- | --- | --- |
| | MATH 500 | Minerva MATH | Olympiad Bench | AMC23 | AIME24 | Theorem QA | GPQA Diamond | MMLU Pro | | | |
| Qwen3-8B (w/ Think) | 82.00 | 41.20 | 44.10 | 67.50 | 40.00 | 46.90 | 35.86 | 68.25 | 53.23 | 30.00 | 23.30 |
| + SFT | 83.20 | 43.80 | 46.40 | 82.50 | 40.00 | 48.90 | 38.38 | 66.81 | 56.25 | 33.30 | 40.00 |
| + R1-GRPO | 91.00 | 52.60 | 65.60 | 82.50 | 50.00 | 57.90 | 40.40 | 70.00 | 63.75 | 53.33 | 43.30 |
| + Critique-GRPO (Ours) (Self-Critique) | **92.60** | **52.60** | **66.20** | **95.00** | **60.00** | **60.60** | **47.98** | **70.03** | **68.13** | **60.00** | **46.70** |

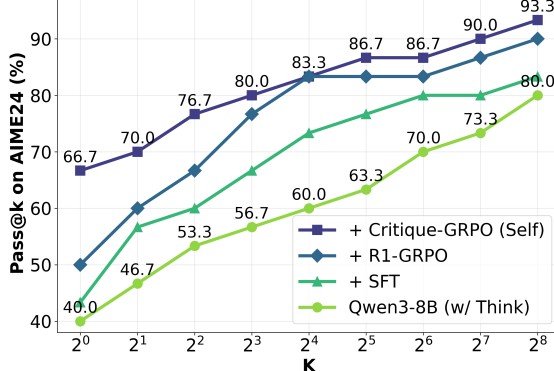

*Figure 3.* Comparison of Pass@k for self-improvement via Critique-GRPO (self-critique), R1-GRPO, SFT, and the base model Qwen3-8B on AIME24.

**Natural language feedback overcomes performance plateaus from numerical-only approaches.** Table 3 shows that Critique-GRPO achieves a 21.6% average pass@1 improvement over Qwen2.5-Math-7B-Base using only 4k RL training prompts, substantially outperforming numerical-feedback approaches that require 46k prompts.

**Verbal credit assignment surpasses numerical credit assignment.** Despite using only 4k prompts, Critique-GRPO (guided by outcome rewards and language critiques) consistently outperforms PRIME-Zero (which uses 46k prompts with dense intermediate rewards) by large margins.

### 5.4. Self-Improvement via Self-Critiquing

We explore Critique-GRPO's capacity for LLM self-improvement through self-critiquing, yielding Critique-

GRPO (self-critique). See Appendix H for details.

**Critique-GRPO enables self-improvement via self-critiquing.** Table 4 shows that Critique-GRPO (self-critique) improves average pass@1 by 4.5% over R1-GRPO and 12.0% over SFT.

**Self-critiquing enhances exploration.** Figures 3 and 9b demonstrate that Critique-GRPO (self-critique) consistently outperforms R1-GRPO and SFT across pass@k metrics on AIME24 and AIME25 for $k \in [1, 256]$, indicating genuine improvements. See Appendix H for additional results.

### 5.5. Exploration with Varying Models

We validate Critique-GRPO's robustness by (i) testing on different model architectures and scales, including Llama-3.2-3B-Instruct (Grattafiori et al., 2024) and Qwen3-32B (Yang et al., 2025a) (Appendix D), and (ii) evaluating compatibility with alternative critique models: Llama3.1-405B (Grattafiori et al., 2024) and DeepCritic-7B-RL1.5-PRM800K (Yang et al., 2025b).

**Critique-GRPO generalizes across model architectures and critique models.** Table 5 (upper section) shows that Critique-GRPO consistently outperforms GRPO on all eight reasoning tasks with Llama-3.2-3B-Instruct and Qwen3-32B (Appendix D), achieving average gains of 5.1 and 4.0 points, respectively, confirming robustness across architectures and scales. The lower section demonstrates compatibility with both proprietary and fine-tuned critique models, yielding average improvements of 5.6–6.8 points over GRPO, underscoring Critique-GRPO's versatility.

*Table 5.* Evaluation of Critique-GRPO with varying policy model architectures and critique models.

| Method | Critique Model | Math (ID) | | | | | Science & General (OOD) | | | Avg. |
|---|---|---|---|---|---|---|---|---|---|---|
| | | MATH 500 | Minerva MATH | Olympiad Bench | AMC23 | AIME24 | Theorem QA | GPQA Diamond | MMLU Pro | |
| **Evaluation with varying policy model architectures** | | | | | | | | | | |
| Llama-3.2-3B-Instruct | - | 46.60 | 17.30 | 15.30 | 27.50 | 13.30 | 21.80 | 22.22 | 38.16 | 22.52 |
| R1-GRPO | - | 53.60 | 19.50 | 18.80 | 27.50 | 13.30 | 24.60 | 29.29 | 40.72 | 24.79 |
| Critique-GRPO | GPT-4o | **58.80** | **25.70** | **24.10** | **35.00** | **20.00** | **31.10** | **36.87** | **43.79** | **29.86** |
| **Evaluation with varying critique models** | | | | | | | | | | |
| Qwen2.5-7B-Base | - | 60.80 | 20.20 | 30.40 | 35.00 | 13.30 | 21.60 | 28.79 | 46.24 | 32.04 |
| R1-GRPO | - | 74.00 | 32.00 | 38.50 | 42.50 | 16.70 | 40.60 | 33.33 | 51.81 | 41.18 |
| Critique-GRPO | Llama3.1-405B | 78.40 | 37.90 | 41.00 | **62.50** | 16.70 | 42.80 | **38.89** | 56.12 | 46.79 |
| Critique-GRPO | GPT-4o | 77.80 | 36.80 | 42.40 | **62.50** | 20.00 | **44.00** | 37.88 | 55.28 | 47.08 |
| Critique-GRPO | DeepCritic-7B | **79.20** | **40.10** | **44.00** | 60.00 | **23.30** | 43.10 | 36.87 | **57.28** | **47.98** |

*Table 6.* Fine-grained ablation of Critique-GRPO objective modifications on Qwen2.5-7B-Base.

| Method | Modifications | | | | Math (ID) | | | | | Science & General (OOD) | | | Avg. |
|---|---|---|---|---|---|---|---|---|---|---|---|---|---|
| | KL Loss Removal | Language Feedback | Refinement Sampling | Policy Shaping | MATH 500 | Minerva MATH | Olympiad Bench | AMC23 | AIME24 | Theorem QA | GPQA Diamond | MMLU Pro | |
| + R1-GRPO | × | × | × | × | 74.00 | 32.00 | 38.50 | 42.50 | 16.70 | 40.60 | 33.33 | 51.81 | 41.18 |
| + Critique-GRPO (CoT Critique) | ✓ | × | × | × | **78.40** | 34.90 | 39.90 | 40.00 | 13.30 | 43.10 | **38.89** | 52.83 | 42.66 |
| | ✓ | ✓ | × | × | 77.00 | 30.90 | 42.40 | 52.50 | 13.30 | 41.80 | 33.33 | 54.88 | 43.26 |
| | ✓ | ✓ | ✓ | × | 77.40 | **41.00** | 39.70 | 45.00 | 16.70 | 42.60 | 34.34 | 54.88 | 43.95 |
| | ✓ | ✓ | ✓ | ✓ | 77.80 | 36.80 | **42.40** | **62.50** | **20.00** | **44.00** | 37.88 | **55.28** | **47.08** |

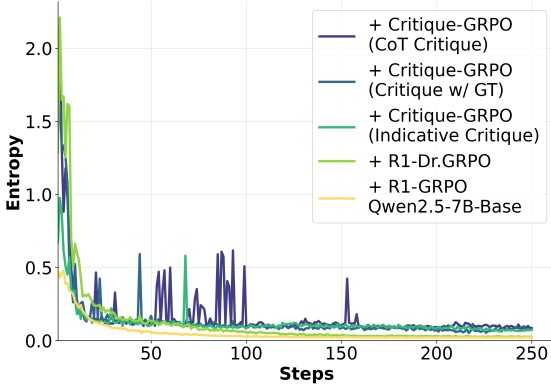

*Figure 4.* Entropy dynamics on Qwen2.5-7B-Base.

### 5.6. Investigation of Policy Exploration

To investigate whether models learn from valuable explorations in refinements, we analyze the entropy dynamics of Qwen2.5-7B-Base in Figure 4.

**Learning from natural language feedback sustains exploration.** Critique-GRPO maintains higher policy entropy than R1-GRPO and R1-Dr.GRPO, indicating more consistent exploration. Early entropy peaks (before step 200) occur when self-generated refinements deviate significantly from initial responses, increasing entropy and enabling beneficial distributional shifts. The subsequent entropy decrease reflects rapid internalization of these refinements. This pat-

tern aligns with prior findings that rare, high-advantage actions increase policy entropy (*i.e.,* unfamiliar but correct responses promote effective exploration), while common, high-advantage actions reduce it (Cui et al., 2025b). Together with Table 2, Critique-GRPO's superior performance demonstrates that maintaining higher entropy enhances policy optimization. This observation is consistent with theoretical analysis in Section 4 and Xu et al. (2025), showing that critiques enable valuable explorations unreachable through standard generation. We provide a comprehensive analysis in Appendix I.

### 5.7. Fine-Grained Ablation Studies

We ablate Critique-GRPO's objective modifications from GRPO: (i) removing KL regularization; (ii) adding natural language feedback (fine-tuning on initial generations and one random refinement); (iii) selecting higher-quality refinements; (iv) applying policy shaping via token-level probability ratios.

**Looser optimization and language feedback enable effective learning.** Table 6 shows cumulative gains: removing KL (+1.5%), adding language feedback (+0.6% to 43.26%), quality-based selection (+0.7% to 43.95%), and policy shaping (+3.1% to 47.08%). The substantial gain from policy shaping demonstrates that amplifying rare but valuable reasoning patterns—those with low initial probability but high success rates—is crucial for RL fine-tuning.

*Table 7.* Exploration of RL fine-tuning with Critique-GRPO for weak-to-strong generalization on Qwen3-8B. Refinements, termed as "weaker refinement," are generated by a weaker model.

| Method | Math (ID) | | | | | Science & General (OOD) | | | Avg. |
|---|---|---|---|---|---|---|---|---|---|
| | MATH 500 | Minerva MATH | Olympiad Bench | AMC23 | AIME24 | Theorem QA | GPQA Diamond | MMLU Pro | |
| Qwen3-8B (w/ Think) | 82.00 | 41.20 | 44.10 | 67.50 | 40.00 | 46.90 | 35.86 | 68.25 | 53.23 |
| + R1-GRPO | **91.00** | **52.60** | **65.60** | 82.50 | 50.00 | 57.90 | 40.40 | 70.00 | 63.75 |
| + Critique-GRPO (Ours) (Weaker Refinement via Critique w/ GT) | 90.40 | 50.70 | 64.90 | **85.00** | **60.00** | **59.00** | **43.43** | **70.94** | **65.55** |

*Table 8.* Exploration of online joint optimization against a sequential baseline (*R1-GRPO + Refinement-SFT*). Metrics represent Pass@1 accuracy (%). The results demonstrate that Critique-GRPO significantly outperforms the decoupled baseline across both in-distribution (ID) and out-of-distribution (OOD) benchmarks. Critique-GRPO consistently achieves a statistically significant improvement ($p < 0.01$) over the sequential baseline.

| Method | Math (ID) | | | | | Science & General (OOD) | | | Avg. |
|---|---|---|---|---|---|---|---|---|---|
| | MATH 500 | Minerva MATH | Olympiad Bench | AMC23 | AIME24 | Theorem QA | GPQA Diamond | MMLU Pro | |
| Qwen-2.5-7B-Base | 60.80 | 20.20 | 30.40 | 35.00 | 13.30 | 21.60 | 28.79 | 46.24 | 32.04 |
| R1-GRPO | 74.00 | 32.00 | 38.50 | 42.50 | 16.70 | 40.60 | 33.33 | 51.81 | 41.18 |
| R1-GRPO + Ref.-SFT | 75.40 | 33.80 | 40.10 | 47.50 | 20.00 | 41.20 | 34.34 | 52.83 | 43.15 |
| **Critique-GRPO** | **77.80** | **36.80** | **42.40** | **62.50** | **20.00** | **44.00** | **37.88** | **55.28** | **47.08** |

### 5.8. Exploration of Weak-to-Strong Generalization

We investigate the potential of weak-to-strong generalization (Burns et al., 2023) using Critique-GRPO, where a strong model learns from refinements generated by a weaker teacher model. Specifically, we use Qwen3-8B-Base (Yang et al., 2025a) as the weaker teacher to generate refinements based on indicative critiques with the ground truth answers, guiding the improvement of Qwen3-8B.

**Critique-GRPO enables effective weak-to-strong generalization.** As shown in Table 7, Critique-GRPO (weaker refinement via critique with ground truth) achieves a +12.3% average pass@1 improvement over Qwen3-8B and outperforms R1-GRPO (65.55% *v.s.* 63.75%). This demonstrates that refinements from a weaker model can significantly enhance the performance of a stronger model.

### 5.9. Online Joint Optimization vs. Sequential Baseline

To isolate the contribution of our online joint optimization, we compare Critique-GRPO against a sequential baseline, *R1-GRPO + Refinement-SFT (CoT Critique)*, executed in two decoupled stages: (1) running standard GRPO to convergence, and (2) critiquing/refining remaining failures followed by SFT on filtered refinements.

As shown in Table 8, Critique-GRPO significantly outperforms this baseline across both ID math and OOD science & general benchmarks. For instance, Critique-GRPO advances

the MATH-500 accuracy to 77.80% and markedly surges to 62.50% on AMC23. On heavy OOD reasoning tasks like GPQA-Diamond, the sequential baseline yields marginal gains (34.34% *v.s.* 33.33%), whereas Critique-GRPO promotes it to 37.88%. These results strongly support the integration of online joint optimization via dual numeric and language feedback.

## 6. Conclusion

We identified three key challenges in RL approaches using only numerical feedback and proposed Critique-GRPO, an online RL framework that learns from both natural language and numerical feedback. Specifically, it enables learning from initial responses and critique-guided self-refinements simultaneously. Experiments across eight reasoning tasks demonstrate that Critique-GRPO consistently outperforms existing SFT and numerical-feedback-only RL methods, paving the way for scaling RL with diverse real-time feedback in real-world post-deployment scenarios.

## Acknowledgments

We sincerely thank Zichen Liu, Zhanhui Zhou, and the anonymous reviewers for their valuable feedback.

## Impact Statement

Throughout our research, we have adhered to the ICML Code of Ethics, prioritizing privacy, fairness, and individual well-being. All benchmark datasets used were strictly for research purposes and contained no personally identifiable information to safeguard privacy. Prompts were carefully designed to minimize bias or discriminatory language and reduce potential negative impacts. Additionally, modelgenerated responses were verified to ensure they were free from offensive content, misinformation, and personally identifiable information.

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

# Appendix

## A. Additional Discussion on Related Work

**Learning from Natural Language Feedback.** Natural Language Feedback (NLF) provides textual critiques with granular diagnostic signals that facilitate precise response refinement (Saunders et al., 2022; Chen et al., 2024; Zhang et al., 2026). Current literature typically integrates NLF by: (1) translating linguistic feedback into scalar reward signals for RL fine-tuning (Kim et al., 2023; Whitehouse et al., 2025; Liu et al., 2025b; Lightman et al., 2024; Ouyang et al., 2022; Casper et al., 2023; Rafailov et al., 2024); or (2) utilizing SFT to imitate static critiques or critique-guided refinements (Wang et al., 2025; Xi et al., 2024; Chen et al., 2024). However, these approaches are primarily offline and lack the capacity for active exploration, limiting their ability to adapt dynamically to idiosyncratic errors encountered during training. While emerging research in natural language RL (Feng et al., 2024; Hong et al., 2025; Luo et al., 2025) has begun to explore linguistic feedback, it remains predominantly confined to offline settings. In contrast, Critique-GRPO integrates real-time NLF directly into the online RL loop. By enabling the model to learn from guided refinements during active exploration, our approach internalizes error-correction mechanisms more effectively, overcoming the performance plateaus inherent in systems relying solely on numerical rewards.

**Enhancing LLM Reasoning with Reinforcement Learning.** Reinforcement Learning (RL) has demonstrated significant effectiveness in improving the reasoning capabilities of Large Language Models (LLMs) (OpenAI et al., 2024; DeepSeek-AI et al., 2025; Fatemi et al., 2025; Li et al., 2025; Feng et al., 2026). This is commonly achieved by fine-tuning models on complex reasoning tasks to encourage diverse and robust reasoning behaviors (Gandhi et al., 2025; Yue et al., 2025). Recent advancements leverage RL with numerical feedback, such as assigning positive rewards (*e.g.,* +1) for correct responses and negative rewards (*e.g.,* -1) for incorrect ones (OpenAI et al., 2024; DeepSeek-AI et al., 2025; Liu et al., 2025a; Yu et al., 2025). These methods often employ online policy optimization algorithms, including Proximal Policy Optimization (PPO) (Schulman et al., 2017), Group Relative Policy Optimization (GRPO) (Shao et al., 2024a), REINFORCE (Williams, 1992), and Decoupled Clip and Dynamic Sampling Policy Optimization (DAPO) (Yu et al., 2025). However, numerical feedback tends to be sparse, and models often struggle with tasks that exceed their current knowledge boundaries, limiting the potential for substantial improvement (Xi et al., 2024; Gandhi et al., 2025). To address this limitation, recent approaches have incorporated high-quality expert demonstrations alongside online exploration (Yan et al., 2025). In contrast, our approach enables models to refine their outputs through the integration of various textual feedback, which helps identify and address potential errors. This combination of textual feedback with online exploration for policy optimization enhances flexibility and scalability.

## B. Analysis of Training Computational Cost

*Table 9.* Analysis of Training Computational Cost.

| Model | Method | Training Time (GPU-Hours) | Average Accuracy (%) |
|---|---|---|---|
| Qwen2.5-7B-Base | R1-GRPO | 240 | 41.18 |
| | Critique-GRPO (CoT-Critique) | 245 | 47.08 |
| Qwen3-8B | R1-GRPO | 296 | 63.75 |
| | Critique-GRPO (CoT-Critique) | 315 | 68.26 |

We utilize asynchronous rollouts to minimize the additional computational cost introduced by incorporating critiques and self-refinement in Critique-GRPO. Table 9 presents the training time required for 300 steps on a single NVIDIA A800 GPU.

The results show that the additional cost is limited to only 5 and 19 GPU-hours for Qwen2.5-7B-Base and Qwen3-8B, respectively, representing approximately a 2% and 6.5% increase over GRPO. This overhead is marginal compared to the significant performance improvement, with average gains of +6 and +4.5 percentage points across 8 tasks, achieved using only 4K prompts. These findings highlight the substantial gains in data efficiency and model performance, demonstrating that the additional computational cost is highly justified.

## C. Impact of Critique Length on Critique-GRPO

*Table 10.* Evaluation of Critique-GRPO with varying critique length.

| Method | Critique Length | Math (ID) | | | | | Science & General (OOD) | | | Avg. |
|---|---|---|---|---|---|---|---|---|---|---|
| | | MATH 500 | Minerva MATH | Olympiad Bench | AMC23 | AIME24 | Theorem QA | GPQA Diamond | MMLU Pro | |
| Qwen2.5-7B-Base | - | 60.80 | 20.20 | 30.40 | 35.00 | 13.30 | 21.60 | 28.79 | 46.24 | 32.04 |
| + R1-GRPO | - | 74.00 | 32.00 | 38.50 | 42.50 | 16.70 | 40.60 | 33.33 | 51.81 | 41.18 |
| + Critique-GRPO (Indicative Critique) | 7 tokens | 76.00 | 36.00 | 41.00 | 55.00 | 13.30 | 41.80 | 37.88 | 55.97 | 44.62 |
| + Critique-GRPO (Critique w/ GT) | 15 tokens | 76.80 | 35.70 | 39.60 | **62.50** | 10.00 | **44.00** | **38.89** | 54.88 | 45.30 |
| + Critique-GRPO (CoT Critique) | 683 tokens | 77.80 | 36.80 | **42.40** | **62.50** | **20.00** | **44.00** | 37.88 | 55.28 | **47.08** |
| + Critique-GRPO (CoT Critique) | 2079 tokens | **77.80** | **37.90** | 38.80 | **62.50** | 16.70 | 42.80 | 35.35 | **56.12** | 46.00 |

We investigate the impact of critique token length on the performance of Critique-GRPO by varying the level of detail in generated critiques. As shown in Table 10, shorter CoT critiques (approximately 683 tokens) generally achieve strong performance. However, increasing the token length to around 2079 tokens by prompting the model to produce highly detailed critiques results in a slight performance drop (from 47.08 to 46.00). This decline is likely due to the increased difficulty of extracting critical information from longer contexts. Despite this, all results using Critique-GRPO consistently surpass those of GRPO, demonstrating the robustness of Critique-GRPO.

In addition, we evaluate the system against adversarial critiques to verify that improvements stem from valid guidance rather than spurious natural language signals. For example, by inverting feedback conclusions (*e.g.,* swapping "correct" and "incorrect"), we observe entropy explosion and subsequent performance degradation.

## D. Scalability of Critique-GRPO to Large-Scale Models

*Table 11.* Evaluation of the scalability of Critique-GRPO on a larger-scale model. (The decoding budget is set to 16,384 tokens for Qwen3-32B.)

| Method | Critique Model | Math (ID) | | | | | Science & General (OOD) | | | Avg. |
|---|---|---|---|---|---|---|---|---|---|---|
| | | MATH 500 | Minerva MATH | Olympiad Bench | AMC23 | AIME24 | Theorem QA | GPQA Diamond | MMLU Pro | |
| Qwen3-32B | - | 91.80 | 53.70 | 62.70 | 92.50 | 63.30 | 62.40 | 45.96 | 77.06 | 68.68 |
| R1-GRPO | - | 93.00 | 54.00 | 66.40 | 97.50 | 80.00 | 64.00 | 56.57 | 77.97 | 73.68 |
| Critique-GRPO | GPT-4o | **96.40** | **59.60** | **69.30** | **100.00** | **90.00** | **66.40** | **60.61** | **78.49** | **77.60** |

We assess the scalability of Critique-GRPO on the large-scale Qwen3-32B model (Yang et al., 2025a). As shown in Table 11, Critique-GRPO consistently surpasses GRPO across all eight reasoning tasks.

# E. Limitations of RL with Numerical Feedback and the Promise of Natural Language Guidance (Detailed Analyses)

## E.1. Three Limitations of Learning with Numerical Feedback

We investigate the limitations of fine-tuning with RL relying solely on numerical feedback from three key perspectives: (i) How best performance improves as the number of training examples increases. (ii) How cognitive behaviors contribute to improving successful problem-solving. (iii) The model's ability to solve previously failed problems through trial-and-error.

**Setup.** We conduct experiments on non-reasoning models, Qwen2.5-7B-Base (Qwen et al., 2025) and Qwen3-8B-Base (Yang et al., 2025a), and a reasoning model, Qwen3-8B (Yang et al., 2025a), for mathematical reasoning tasks. Specifically, we fine-tune the models using GRPO (Shao et al., 2024b) with numerical feedback.[1]

*Datasets and Evaluation Metrics.* We utilize randomly sampled subsets of 4k, 8k, 16k, and 32k examples from a reorganized 46k subset (Yan et al., 2025) of OpenR1-Math-220k (Bakouch et al., 2025). The prompts are sourced from NuminaMath 1.5 (Li et al., 2024), while the ground truth chain-of-thought (CoT) reasoning paths are generated by Deepseek-R1 (DeepSeek-AI et al., 2025). Unless otherwise specified, experiments primarily use 4k training prompts. For validation, we randomly sample 500 examples from the validation set curated by (Yan et al., 2025), which includes examples from Olympiad Bench (He et al., 2024b), MATH (Hendrycks et al., 2021), Minerva-Math (Lewkowycz et al., 2022), AIME 2024 (Li et al., 2024), and AMC 2023 (Li et al., 2024). To provide a comprehensive evaluation, we assess performance on in-distribution (ID) tasks using Minerva-Math (Lewkowycz et al., 2022) and on out-of-distribution (OOD) tasks using GPQA-Diamond, which covers physics, chemistry, and biology (Rein et al., 2024). For evaluation, we employ greedy decoding (temperature = 0) and report accuracy as pass@1, following prior work (Zeng et al., 2025; Yang et al., 2025a; Liu et al., 2025a).[2]

*Reward Design.* We employ rule-based evaluation to provide numerical feedback (scalar rewards), using Math-Verify[3] to validate the correctness of generated answers against ground truth during fine-tuning. Binary rewards are assigned as follows: $+1$ for correct final answers and $0$ for incorrect ones. These rewards serve as a proxy for assessing the accuracy of generated responses.

*Implementation Details.* Our implementation leverages the VERL library (Sheng et al., 2024) and samples four candidate responses per prompt during fine-tuning.

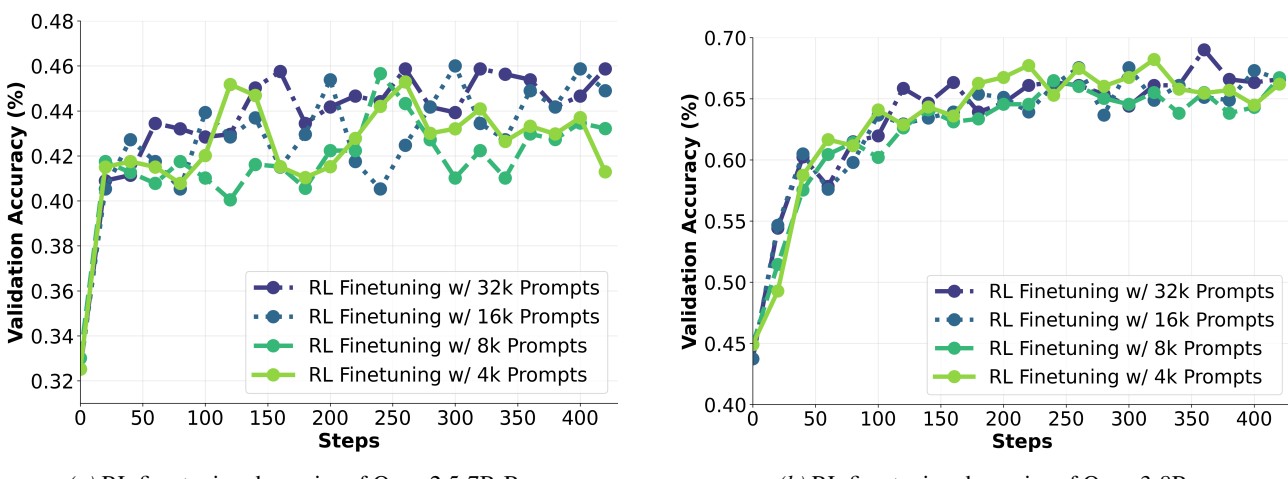

*(a)* RL fine-tuning dynamics of Qwen2.5-7B-Base.      *(b)* RL fine-tuning dynamics of Qwen3-8B.

*Figure 5.* RL fine-tuning dynamics of Qwen2.5-7B-Base (left) and Qwen3-8B (w/ Thinking) (right) using GRPO with numerical feedback on the validation set with varying numbers of training examples.

**Results.** **RL with solely numerical feedback frequently encounters performance plateaus.** Figure 5 illustrates the RL fine-tuning dynamics of Qwen2.5-7B-Base and Qwen3-8B across varying numbers of training examples. On-policy

---

[1]GRPO is used without loss of generality, as RL algorithms such as PPO and GRPO exhibit comparable performance.

[2]The pass@k metric represents the percentage of problems where the model produces a correct solution within its first $k$ attempts.

[3]https://github.com/huggingface/Math-Verify

RL frequently stagnates, as reflected in validation set accuracy: Qwen2.5-7B-Base reaches its highest performance at approximately 45-46% accuracy after 120 steps (Figure 5a), while Qwen3-8B plateaus at 65-67% accuracy after 200 steps (Figure 5b). Critically, peak accuracy shows negligible improvement despite an 8-fold increase in training prompts or extending the training duration to 420 steps. These results underscore a fundamental scaling bottleneck in scalar-reward RL, where increased compute and data fail to drive further reasoning gains.

**Spontaneous self-reflection has limited impact on enhancing problem-solving success.** Cognitive behaviors are widely recognized as key contributors to successful complex reasoning (DeepSeek-AI et al., 2025; Gandhi et al., 2025). In particular, increased self-reflection behaviors after RL fine-tuning, which mimic humans reflecting on past experiences and refining their approach to reach a solution (commonly referred to as the "Aha moment" (DeepSeek-AI et al., 2025)), have drawn significant attention. However, does spontaneous self-reflection play the most critical role in improved performance?

To address this question, we characterize six key cognitive behaviors that contribute to self-improving reasoning during RL fine-tuning:

- **Subgoal Setting**: Decomposing complex problems into smaller, manageable subtasks.

- **Summarization**: Summarizing the current state by identifying completed subtasks and determining the next steps in reasoning.

- **Verification**: Systematically checking intermediate results or computations to ensure correctness.

- **Backtracking**: Identifying errors or dead-ends in reasoning and revising previous methods or approaches.

- **Backward Chaining**: Reasoning from desired outcomes back to the initial inputs or steps required to achieve the result. This is particularly useful for multiple-choice questions with provided answer options (Gandhi et al., 2025).

- **Anticipation**: Anticipating potential inaccuracies or exploring alternative solutions to a problem.

We categorize the first two behaviors as **planning behaviors** and the remaining four as **self-reflection behaviors**. To analyze their contributions, we evaluate problems previously unsolved by the base model. For Qwen2.5-7B-Base, we identify 87 unsolved problems from the Minerva-Math dataset and 37 from the GPQA-Diamond dataset. For Qwen3-8B, we identify 33 unsolved problems from the Minerva-Math dataset and 15 from the GPQA-Diamond dataset. We then examine the cognitive behaviors exhibited in the correct responses generated by RL-finetuned models for these problems.

To detect these behaviors, we use GPT-4o (Hurst et al., 2024) as an automatic judge with manually crafted prompts (Appendix M). Each behavior $b_i$ is scored as:

$$s_i = \begin{cases} 1 & \text{if behavior } b_i \text{ is present,} \\ 0 & \text{otherwise.} \end{cases} \tag{5}$$

The average success contribution of each behavior is computed as follows:

$$\text{Contribution}(b_i) = \frac{\sum_{j=1}^{N} s_{i,j}}{N}, \tag{6}$$

where $N$ is the total number of analyzed responses, and $s_{i,j}$ indicates whether behavior $b_i$ appears in response $j$. Further details are provided in Appendix L.

Figure 6 presents the average success contribution of various behaviors, showing that planning behaviors are the primary contributors to successful problem-solving, while self-reflection behaviors contribute less in both the mathematical (Minerva-Math) and STEM (GPQA-Diamond) domains. In Figure 6a, self-reflection behaviors barely contribute in the mathematical domain for the non-reasoning model. Thanks to extensive training on expert demonstrations with diverse reasoning behaviors in the mathematical domain (Yang et al., 2025a), the reasoning model shows that self-reflection behaviors make a noticeable contribution (Figure 6b). Nevertheless, self-correction-related behaviors, such as backtracking, backward chaining, and anticipation, still contribute considerably less. These observations suggest the limited effectiveness of

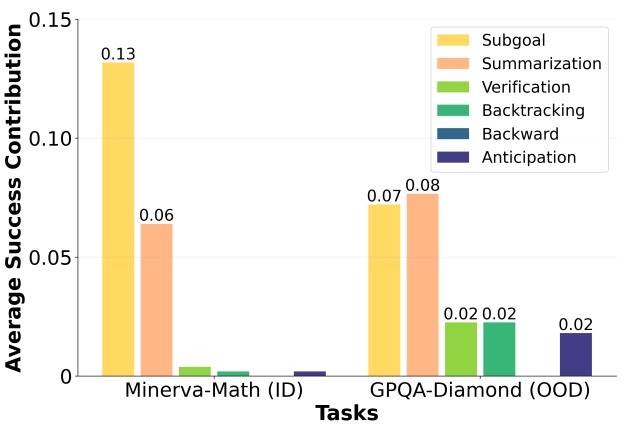

*(a)* Reasoning behavior contributions to successful problem-solving in RL-tuned Qwen2.5-7B-Base.

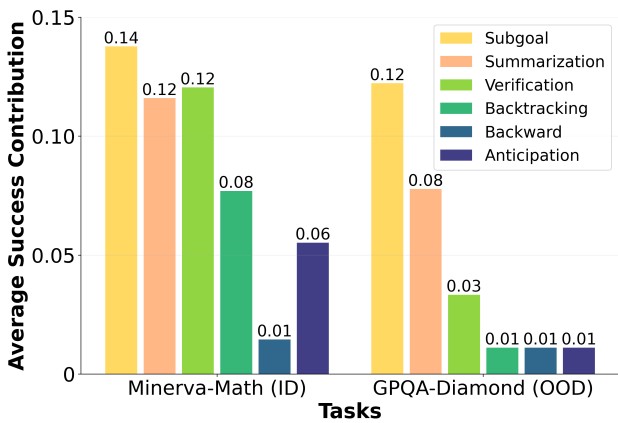

*(b)* Reasoning behavior contributions to successful problem-solving in RL-tuned Qwen3-8B.

*Figure 6.* Impact of different reasoning behaviors on solving previously failed problems after RL fine-tuning with numerical feedback for Qwen2.5-7B-Base (left) and Qwen3-8B (right) on test tasks.

spontaneous self-reflection. This underscores the unreliability of relying on spontaneous self-reflection for self-correction to improve problem-solving success.

**Models exhibit persistent failures on a subset of problems despite trial-and-error fine-tuning.** We evaluated the best-performing RL-finetuned Qwen2.5-7B-Base and Qwen3-8B models on 4k training prompts. As shown in the left panel of Table 12, these models consistently failed on approximately 29% and 3.75% of problems, respectively, with pass@4 = 0. This occurred despite undergoing trial-and-error fine-tuning, where correct responses are rewarded, and incorrect responses are penalized. While the reasoning model (Qwen3-8B) exhibits evident spontaneous self-reflection and significantly better performance with fewer persistent failures, it still struggles with certain problems when relying solely on numerical feedback.

A likely cause of these performance plateaus and persistent failures is the sparse informational content of numerical feedback. Scalar rewards often fail to convey *why* a response is correct or incorrect or *how* to improve multi-step reasoning. Furthermore, the limited effectiveness of spontaneous self-reflection exacerbates these challenges. Together, these limitations highlight the necessity of richer feedback mechanisms to enable more effective learning.

### E.2. Promise of Learning from Natural Language Feedback

To move beyond the limitations of purely numerical reward signals, we explore the potential of leveraging natural language feedback to help models identify errors and refine their responses. Specifically, we examine three types of critiques:

- **Indicative Critique**: A heuristic-based critique that merely indicates the binary correctness of the generated solution, **without an expert trajectory**.

- **Indicative Critique with Ground Truth**: A heuristic-based critique providing both binary correctness and the ground truth answer, **without an expert trajectory**.

- **CoT Critique**: A model-generated critique offering a step-by-step evaluation to justify the assessment, concluding with a binary correctness label, **without an expert trajectory**.

Examples of these critique types are presented below. We summarize our method for leveraging textual critiques to guide response refinement in Algorithm 1. A detailed description is provided in Section E.3, and an illustrative example of refinement using CoT critique is available in Appendix N.

> **Indicative Critique**   The generated solution is incorrect.

> **Indicative Critique w/ Ground Truth**   The generated solution is incorrect, the ground truth is $\frac{7}{25}$.

**CoT Critique**

Let's analyze the student's solution step-by-step and identify any errors:

### Step 1: Geometry Understanding The student correctly identifies the geometry of the problem, noting that the sphere is tangent to the base and sides of the cone, and that the axial section forms an isosceles triangle.

### Step 2: Variable Definitions The student defines $r$ as the radius of the sphere, $R$ as the radius of the base of the cone, and $h$ as the height of the cone. These definitions are appropriate.

### Step 3: Relating Sphere and Cone The student attempts to relate $r$, $R$, and $h$ using similar triangles. However, the equation derived:

$$r = \frac{Rh}{h + R}$$

is not directly applicable to the problem as it does not consider the specific condition that the areas are equal.

### Step 4: Equating Areas The student correctly equates the areas of the sphere and the base of the cone:

$$4\pi r^2 = \pi R^2 \implies R = 2r$$

This step is correct.

### Step 5: Substitution and Error The student substitutes $R = 2r$ into the earlier equation, leading to:

$$r = \frac{(2r)h}{h + 2r}$$

This substitution is unnecessary and results in incorrect simplification. The correct approach would involve using the inradius formula for the triangle formed by the axial section of the cone.

### Step 6: Angle Calculation The student calculates $\tan \theta = 1$ and concludes $\theta = 45°$, leading to $\cos(90°) = 0$. This is incorrect because the angle calculation should be based on the relationship between $h$ and $R$, derived from the inradius formula.

### Step 7: Precise Calculation The student attempts to verify the angle calculation but repeats the same incorrect logic, leading to the same incorrect conclusion.

### Correct Solution Approach The correct approach involves using the inradius formula for the triangle formed by the axial section of the cone: - Calculate the inradius using the area and semi-perimeter of the triangle. - Substitute $R = 2r$ and solve for $h$ in terms of $r$. - Relate $h$ and $\theta$ using trigonometric identities. - Calculate $\cos(2\theta)$ using the double angle identity.

The correct final answer is:

$$\cos(2\theta) = \frac{7}{25}.$$

Conclusion: incorrect [END]

**Setup.** As described in the previous section, we evaluate the best-performing RL-finetuned Qwen2.5-7B-Base and Qwen3-8B models, generating four responses per question for a set of 4k prompts. Persistently failed question-response pairs are identified as those with pass@4 = 0. For each such pair, we prompt GPT-4o (Hurst et al., 2024) to generate a CoT critique and then construct the two corresponding heuristic-based critiques. Examples of these critique types are shown below. Subsequently, we prompt the models to refine these failed responses.

*Evaluation Metrics.* To assess the efficacy of the critique and refinement process, we define the following metrics: (i) **Valid Critiques Generated**: The percentage of initially erroneous responses for which the critique model ($\pi_\phi$) successfully generated a well-formed and usable critique. This accounts for potential failures in the critique generation process itself. (ii) **Successful Refinements**: The percentage of initially erroneous responses (that received a valid critique) for which the LLM ($\pi_\theta$) subsequently generated a correct refined response. (iii) **Critiqued Questions**: The percentage of unique questions (all of whose initial $k$ responses were erroneous) for which at least one of their erroneous responses received a valid critique. (iv) **Questions Successfully Refined**: The percentage of unique questions (all of whose initial $k$ responses were erroneous) for which at least one of their erroneous responses was successfully corrected through the refinement process. We evaluate the correctness of the refinements using the rule-based method described earlier.

**Results. Deliberate critique is more effective than spontaneous self-reflection for self-correction.** Incorporating all three types of critiques yields non-zero valid refinements and correctly refined questions. This suggests that critiques can

---

**Algorithm 1** Leveraging Textual Critiques for Refinement of LLM-Generated Responses

---

1: **Input:** LLM $\pi_\theta$, Reasoning-based reward model $\pi_{RM}$, Evaluation Eval, Questions $Q = \{q\}$, Instructions $I_c, I_{\text{refine}}$, Sample count $k$
2: **Output:** Refined responses $\{y_{\text{refined}}^{(j)}\}$ for failed questions
3: *// Step 1: Initial Response Sampling*
4: **for** each question $q \in Q$ **do**
5:      Sample $k$ initial responses: $\{y_0^{(i)}\}_{i=1}^k \sim \pi_\theta(\cdot \mid q)$
6: **end for**
7: *// Step 2: Response Evaluation and Critique Generation*
8: **for** each question $q \in Q$ **do**
9:      **for** each initial response $y_0^{(i)} \in \{y_0^{(i)}\}_{i=1}^k$ **do**
10:          Generate CoT critique: $c_{\text{CoT}}^{(i)} \sim \pi_{RM}(\cdot \mid I_c, q, y_0^{(i)})$
11:          Evaluate correctness: $\text{Eval}(q, y_0^{(i)}) \in \{0, 1\}$
12:          **if** $\text{Eval}(q, y_0^{(i)}) = 0$ **then**
13:              Construct heuristic-based critiques: $c_I^{(i)}$ (indicative) and $c_{\text{GT}}^{(i)}$ (w/ GT)
14:          **end if**
15:      **end for**
16: **end for**
17: Identify failed questions: $Q_{\text{failed}} \leftarrow \{q \in Q \mid \forall i : \text{Eval}(q, y_0^{(i)}) = 0\}$
18: Form triplets $(q, y_0^{(j)}, c^{(j)})$ for $q \in Q_{\text{failed}}$ where $c^{(j)} \in \{c_{\text{CoT}}^{(j)}, c_{\text{GT}}^{(j)}, c_I^{(j)}\}$
19: *// Step 3: Self-Refinement Generation*
20: **for** each triplet $(q, y_0^{(j)}, c^{(j)})$ **do**
21:      Generate refined response: $y_{\text{refined}}^{(j)} \sim \pi_\theta(\cdot \mid I_{\text{refine}}, q, y_0^{(j)}, c^{(j)})$
22: **end for**
23: **return** $\{y_{\text{refined}}^{(j)}\}$

---

enable both models to successfully correct some persistently failed responses that cannot be resolved through spontaneous self-reflection, as discussed in Section 3.1.

**CoT Critiques facilitate effective model refinement.** Table 12 shows that refinement guided by CoT critiques achieves the highest valid refinement rate (36.47% and 10.63%) and the largest percentage of successfully refined questions (55.37% and 20.00%) on Qwen2.5-7B-Base and Qwen3-8B, respectively. This performance significantly surpasses refinement based on indicative critiques or critiques with ground truth, even though the CoT critique generation process produces valid critiques for only 60.06% and 50.17% of erroneous responses. The effectiveness of CoT critiques can be attributed to their richness: by providing a step-by-step evaluation of the reasoning (*potentially along with* targeted guidance on the correct solution approach).

**Binary correctness signals alone can provide refinement benefits.** Refinement with indicative critiques with/without ground truth also yields some successful refinements, albeit at a substantially lower rate (approximately 2%-4% valid refinements). This suggests that even simply indicating the correctness of a response can provide a minimal benefit, indicating some promise in leveraging natural language feedback to augment learning from numerical signals. However, the lack of a substantial difference between indicative critiques and critiques with ground truth suggests that providing only the ground-truth answer, without any explanation or reasoning, provides little additional guidance to the model.

### E.3. Leveraging Textual Critiques for Refining LLM Responses

We describe the process for leveraging these textual critiques to guide the refinement of LLM-generated responses:

1. **Initial Response Sampling**: Given an LLM $\pi_\theta$ parameterized by $\theta$ and a set of questions $\{q\}$, we sample multiple initial responses for each question $\{y_0^{(i)}\}_{i=1}^k \sim \pi_\theta(\cdot \mid q)$, where $k$ is the number of samples.

2. **Response Evaluation and Critique Generation**: We use an evaluation function $\text{Eval}(q, y_0)$ to assess the correctness of each response $y_0$. The function outputs 1 if $y_0$ is correct and 0 otherwise. Specifically, we adopt a model-based

*Table 12.* Analysis of performance gains from critique-based self-refinement.

| Method | % Failed Questions (Pass@4=0) | Critique Type | % Valid Critiques | % Valid Refinements | % Critiqued Questions | % Questions Refined |
|---|---|---|---|---|---|---|
| RL-finetuned Qwen2.5-7B-Base | 29.07 | Indicative Critique | 100.00 | 2.09 | 100.00 | 7.05 |
| | | Indicative Critique w/ GT | 100.00 | 1.98 | 100.00 | 6.88 |
| | | CoT Critique | 60.06 | **36.47** | 95.10 | **55.37** |
| RL-finetuned Qwen3-8B (w/ Thinking) | 3.75 | Indicative Critique | 100.00 | 3.33 | 100.00 | 8.67 |
| | | Indicative Critique w/ GT | 100.00 | 3.67 | 100.00 | 10.67 |
| | | CoT Critique | 50.17 | **10.63** | 88.67 | **20.00** |
| RL-finetuned Qwen3-8B-Base | 17.18 | Indicative Critique | 100.00 | 3.57 | 100.00 | 11.21 |
| | | Indicative Critique w/ GT | 100.00 | 3.93 | 100.00 | 12.23 |
| | | CoT Critique | 66.08 | **44.71** | 98.25 | **66.96** |

evaluation with a reasoning-based reward model $\pi_{RM}$. The reasoning-based reward model generates a CoT critique $c_{\text{CoT}}^{(i)} \sim \pi_{RM}(\cdot \mid I_c, q, y_0^{(i)})$, where $I_c$ is a predefined instruction (detailed in Appendix M). Based on the binary correctness label within $c_{\text{CoT}}^{(i)}$, we construct the corresponding heuristic-based critiques: an indicative critique $c_{\text{I}}^{(i)}$ (containing only the correctness label) and a critique with ground truth $c_{\text{GT}}^{(i)}$ (correctness label plus the known ground truth answer for $q$).

To focus on the model's ability to learn from critiques for initially incorrect solutions and to control for spontaneous self-correction, we identify persistently failed questions. A question $q$ is classified as persistently failed if all $k$ of its initial responses $\{y_0^{(i)}\}_{i=1}^k$ are deemed incorrect based on the labels from their respective CoT critiques. For each such incorrect response $y_0^{(j)}$ from a persistently failed question, we form a triplet $(q, y_0^{(j)}, c^{(j)})$, where $c^{(j)}$ is one of the three critique types: $c_{\text{CoT}}^{(j)}, c_{\text{GT}}^{(j)},$ or $c_{\text{I}}^{(j)}$.

3. **Self-Refinement Generation**: For each selected triplet $(q, y_0^{(j)}, c^{(j)})$ corresponding to an initial incorrect response, we prompt the original LLM $\pi_\theta$ to generate a refined response $y_{\text{refined}}^{(j)} \sim \pi_\theta(\cdot \mid I_{\text{refine}}, q, y_0^{(j)}, c^{(j)})$. This generation is conditioned on a specific refinement instruction $I_{\text{refine}}$ (detailed in Appendix M), the original question $q$, the initial failed response $y_0^{(j)}$, and its associated critique $c^{(j)}$.

The full process is summarized in Algorithm 1. An example illustrating the self-refinement process, including the application of a CoT critique, is provided in Appendix N.

### E.4. Investigation on Qwen3-8B-Base

We identify 50 previously unsolved problems from the Minerva-Math dataset and 15 from the GPQA-Diamond dataset for Qwen3-8B-Base. Figure 7 shows the average contribution of reasoning behaviors to successful completions of previously failed questions by Qwen3-8B-Base. Notably, self-reflection behaviors contribute minimally to successful problem-solving.

Table 12 reveals that the best-performing RL-finetuned Qwen3-8B-Base persistently failed on 17.18% of training problems. In addition, all three types of critiques facilitate the LLM's self-refinements.

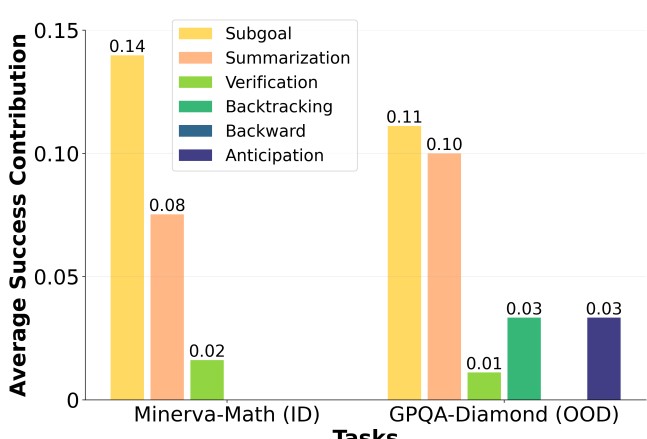

*Figure 7.* Average contribution of reasoning behaviors to successful completions of previously failed questions by Qwen3-8B-Base on Minerva-Math (in-distribution) and GPQA-Diamond (out-of-distribution) datasets.

# F. More Details about Critique-GRPO

## F.1. The Critique-GRPO Algorithm

The Critique-GRPO algorithm is summarized in Algorithm 2.

---

**Algorithm 2** Critique-GRPO: Online Policy Optimization Framework

---

1: **Input:** Pretrained policy $\pi_{\text{old}}(\theta)$, reward model $\pi_{RM}$, questions $Q$, instructions $I_{\text{refine}}, I_c$
2: **Goal:** Improve policy by learning from initial responses and critiques
3: // *Step 1: Initial Response Sampling*
4: **for** each question $q \in Q$ **do**
5:     Sample $n$ initial responses: $\{y^{(i)}\}_{i=1}^n \sim \pi_{\text{old}}(\cdot|q)$
6:     Generate CoT critiques: $\{c_{\text{CoT}}^{(i)}\}_{i=1}^n \sim \pi_{RM}(\cdot \mid I_c, q, y^{(i)})$
7:     Obtain scalar rewards: $\{R^{(i)}\}_{i=1}^n$ from $\pi_{RM}$ labels
8: **end for**
9: // *Step 2: Critique-Guided Self-Refinement*
10: **for** each response $y^{(i)} \in \{y^{(i)}\}_{i=1}^n$ **do**
11:     Generate refinement: $y_{\text{refined}}^{(i)} \sim \pi_{\text{old}}(\cdot \mid I_{\text{refine}}, q, y^{(i)}, c_{\text{CoT}}^{(i)})$
12:     Score refinement: $R_{\text{refine}}^{(i)} \leftarrow \text{Eval}(y_{\text{refined}}^{(i)})$
13: **end for**
14: Sample $k$ refinements to mitigate shift: $\{y_{\text{refined}}^{(i')}\}_{i'=1}^k \subset \{y_{\text{refined}}^{(i)}\}_{i=1}^n$
15: Form mixed group: $\mathcal{Y}_{\text{mix}} = \{y^{(i)}\}_{i=1}^n \cup \{y_{\text{refined}}^{(i')}\}_{i'=1}^k$
16: // *Step 3: Online Policy Optimization*
17: Compute token-level advantage $\hat{A}_t$ for $\mathcal{Y}_{\text{mix}}$ using group relative rewards
18: Calculate ratios $r_t(\theta)$ and apply policy shaping for refinements $r_{\text{refine},t}(\theta)$
19: Update $\theta$ by maximizing $\mathcal{J}_{\text{Critique-GRPO}}$ (Eq. 2)
20: **Output:** Fine-tuned policy $\pi_\theta$

---

## F.2. The shaping Function in Critique-GRPO

We adopt a shaping function $f(x) = x/(x + \gamma)$(Yan et al., 2025) ($0 < \gamma < 1$), depicted in Figure8, to reweight gradients and emphasize low-probability tokens in refined responses. As illustrated in Figure 8, this function is bounded between $(0, 1)$, where $x$ represents the token probability of the policy. When $\gamma$ is small (*i.e.*, 0.1), the function significantly amplifies low probabilities, with this amplification decreasing as $x$ increases. Larger $\gamma$ values (0.9) produce less pronounced scaling effects. The black dashed diagonal line indicates no shaping (*i.e.*, $f(x) = x$). We set $\gamma = 0.1$ to optimize learning from unfamiliar yet correct refinements while strongly penalizing unfamiliar incorrect ones.

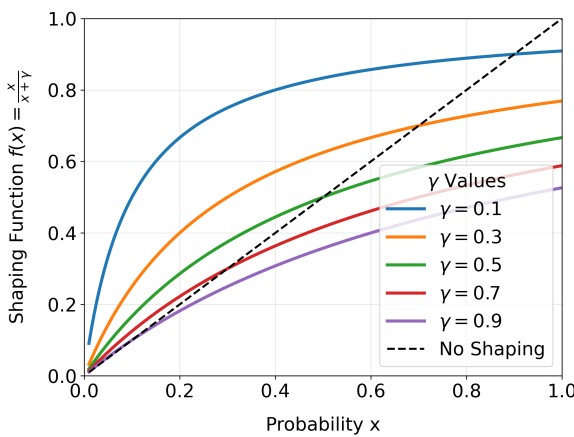

*Figure 8.* Effect of $\gamma$ on the shaping function $f(x)$. The black dashed diagonal line indicates no shaping, $f(x) = x$.

## F.3. Ratio of Responses per Prompt in Critique-GRPO

We experiment with the ratio of initial responses to refinements per prompt, ranging from 1:1 to 7:1. Our results indicate that a 7:1 ratio achieves both stable training and optimal performance. Lower ratios lead to performance degradation due to sudden increases in entropy loss, caused by distribution shifts introduced by the refinements, during later training stages.

## F.4. Theoretical Analysis of Sample Efficiency

In this section, we provide a theoretical justification for the superior sample efficiency of critique-guided refinement compared to standard reward-based learning. We utilize the *Transfer Eluder Dimension* (Xu et al., 2025) framework to quantify the information gain provided by language feedback. We first formalize our critique mechanism as *Reward-Informative Feedback*, establishing that it allows the policy to distinguish hypotheses significantly more efficiently than scalar rewards alone. We then apply this framework to reasoning tasks to demonstrate an exponential reduction in sample complexity.

**Preliminaries: Complexity Measures and Verifiers.** To characterize the complexity of the hypothesis space $\mathcal{H}$, we first introduce the Eluder Dimension (Russo & Van Roy, 2013), a standard measure in reinforcement learning that quantifies how many observations are required to reduce uncertainty about a reward function.

**Definition F.1** (Eluder Dimension (Russo & Van Roy, 2013))**.** An action $a \in \mathcal{A}$ is $\epsilon$-dependent on actions $\{a_1, \ldots, a_n\} \subset \mathcal{A}$ with respect to a reward class $\mathcal{R}$ if any pair of reward functions $r, r' \in \mathcal{R}$ satisfying $\sum_{i=1}^{n}(r(a_i) - r'(a_i))^2 \le \epsilon^2$ also satisfies $|r(a) - r'(a)| \le \epsilon$. The $\epsilon$-eluder dimension, denoted $\dim_E(\mathcal{R}, \epsilon)$, is the length of the longest sequence in $\mathcal{A}$ such that every element is $\epsilon'$-independent of its predecessors for some $\epsilon' \ge \epsilon$.

In Language Learning from Feedback (LLF) (Cheng et al., 2023), agents leverage feedback $o$ (*e.g.,* critiques) beyond scalar rewards. To quantify the information in $o$, we assume the existence of a *verifier*.

**Assumption F.2** (Verifier (Xu et al., 2025))**.** The verifier defines a loss $\ell : \mathcal{A} \times \mathcal{O} \times \mathcal{H} \to [0, 1]$ measuring the alignment between a hypothesis $\eta$ and feedback $o$ on action $a$. Consistency yields $\ell(a, o, \eta) = 0$, while inconsistency incurs a non-zero penalty.

To accommodate potential noise, we assume feedback is unbiased: each hypothesis minimizes the expected verifier loss under its induced distribution. Defining the expected minimum loss as $\ell_\eta^{\min}(a) := \min_{\eta'} \mathbb{E}_{o \sim f_\eta(a)}[\ell(a, o, \eta')]$, we utilize this to define the *Transfer Eluder Dimension*:

**Definition F.3** (Transfer Eluder Dimension (Xu et al., 2025))**.** An action $a$ is $\epsilon$-*transfer dependent* on actions $\{a_1, \ldots, a_n\} \subset \mathcal{A}$ with respect to $\mathcal{H}$ and verifier $\ell$ if any pair $\eta, \eta' \in \mathcal{H}$ satisfying:

$$\sum_{i=1}^{n} \left( \mathbb{E}_{o \sim f_{\eta'}(a_i)}[\ell(a_i, o, \eta)] - \ell_{\eta'}^{\min}(a_i) \right) \le \epsilon^2 \tag{7}$$

also satisfies $|r_\eta(a) - r_{\eta'}(a)| \le \epsilon$. The $\epsilon$-transfer eluder dimension, $\dim_{TE}(\mathcal{H}, \ell, \epsilon)$, is the length of the longest sequence of transfer-independent actions.

**Critique as Reward-Informative Feedback.** Intuitively, the Transfer Eluder Dimension measures how effectively feedback reduces uncertainty about the reward. For this framework to apply, we must establish that our critique mechanism is at least as useful as the reward signal. We rely on the concept of *Reward-Informative Feedback*.

**Definition F.4** (Reward-Informative Feedback)**.** A feedback function $f_\eta$ is reward-informative of $r_\eta$ with respect to a verifier $\ell$ if there exists a constant $C_F > 0$ such that for all $\eta' \in \mathcal{H}$ and $a \in \mathcal{A}$:

$$|r_\eta(a) - r_{\eta'}(a)|^2 \le C_F \mathbb{E}_{o \sim f_\eta(a)}[\ell(a, o, \eta') - \ell_\eta^{\min}(a)]. \tag{8}$$

This condition implies that if two hypotheses differ in reward, the verifier can distinguish them via feedback. Our employed critiques (both indicative and constructive) satisfy this condition by providing strictly more information than a binary reward $r \in \{0, 1\}$, enabling the rejection of incorrect hypotheses without needing to observe the final scalar reward.

**Proposition F.5** (Generalization Bound)**.** *For reward-informative LLF problems with constant $C_F$, the Transfer Eluder Dimension is bounded by the standard Eluder Dimension of the effective reward class $\mathcal{R}_\mathcal{H} = \{r_\eta : \eta \in \mathcal{H}\}$:*

$$\dim_{TE}(\mathcal{H}, C_F\ell, \epsilon) \le \dim_E(\mathcal{R}_\mathcal{H}, \epsilon). \tag{9}$$

Proposition F.5 guarantees that learning from critiques is *no harder* than learning from rewards. However, in practice, the inequality is strict and the gap is large. Below, we quantify this gap for reasoning tasks.

**Complexity Reduction Analysis.** We analyze the reduction in the effective search space for a reasoning task where the goal is to construct a hidden optimal solution sequence $a^* = (s_1^*, \ldots, s_L^*)$ of length $L$ over a vocabulary $\mathcal{S}$. The action space is defined as $\mathcal{A} = \mathcal{S}^L$.

**Proposition F.6** (Sample Efficiency of Critique-Guided Refinement). *We compare the dimension of the hypothesis space under Reward-Only Learning versus Critique-Guided Learning.*

1. ***Standard Generation – Reward-Only Learning (Eluder Dimension):*** *With binary rewards $r(a) = \mathbb{I}(a = a^*)$, the signal is sparse (a "needle in a haystack"). Observing $r(a) = 0$ eliminates only the specific sequence $a$, providing no information about the correctness of other sequences $a' \neq a$. Consequently, the agent must effectively enumerate the action space to find $a^*$. The Eluder dimension scales exponentially with the sequence length:*

$$\dim_E(\mathcal{R}, \epsilon) \approx O(|\mathcal{S}|^L). \tag{10}$$

2. ***Critique-Guided Refinement – Critique-Guided Learning (Transfer Eluder Dimension):*** *We analyze two types of feedback mechanisms:*

   - *Indicative Feedback ($c_I$, $c_{GT}$): This feedback indicates failure but lacks specific error localization. While the worst-case complexity remains $\dim_{TE}(\mathcal{H}, \ell, \epsilon) \approx O(|\mathcal{S}|^L)$, the critique acts as a pruning signal. Conditioning on the failure and the specific content of the critique restricts the search to a subspace $\mathcal{A}_c \subset \mathcal{A}$, reducing the effective search space size by a constant factor $\alpha < 1$.*
   - *Constructive Feedback ($c_{CoT}$): If the critique localizes the first error at step $t$, the problem decomposes into $L$ sequential sub-problems, each of size $|\mathcal{S}|$. This reduces the complexity from exponential to linear, i.e., $\dim_{TE}(\mathcal{H}, \ell, \epsilon) \approx O(L|\mathcal{S}|)$. Furthermore, if the critique provides the correction suggestion for the first error $s_t^*$, the complexity becomes independent of the vocabulary size, scaling as $\dim_{TE}(\mathcal{H}, \ell, \epsilon) \approx O(L)$.*

Consequently, for a fixed computational budget $M$ where $L|\mathcal{S}| \ll M \ll |\mathcal{S}|^L$, critique-guided exploration yields a significantly higher probability of success:

$$P(a^* \in \{y_{\text{refined}}^{(j)}\}_{j=1}^M) \gg P(a^* \in \{y^{(i)}\}_{i=1}^M). \tag{11}$$

*Proof.* We prove this by estimating the Transfer Eluder Dimension $d$, which quantifies the effective search space size. We assume a uniform exploration strategy over the effective search space.

**1. Derivation of Effective Dimension ($d$)**

*Case A: Reward-Only.* The binary indicator problem is equivalent to a standard bandit problem with $|\mathcal{S}|^L$ arms. In this setting, the feedback function is simply the reward itself, $f_\eta(a) = r(a)$. Observing $r(a) = 0$ provides information only about action $a$ and no other $a' \neq a$. Thus, $a'$ is $\epsilon$-independent of $a$ (Definition F.1). We can construct a sequence of independent actions spanning the entire space. Thus, the dimension is bounded by the cardinality of the action space:

$$d_{\text{std}} = |\mathcal{S}|^L. \tag{12}$$

*Case B: Critique-Guided (Constructive).* Let the feedback $f(a)$ return the index of the first error: $t = \min\{i \mid s_i \neq s_i^*\}$. We partition the action space $\mathcal{A}$ into $L$ disjoint sets, where $\mathcal{A}_t = \{(s_1, \ldots, s_L) \mid s_{1:t-1} \text{ are correct}, s_t \text{ is incorrect}\}$. According to Definition F.3, an action is transfer-dependent if similar feedback implies similar rewards. If we select more than $|\mathcal{S}|$ actions from a partition $\mathcal{A}_t$, by the Pigeonhole Principle, at least two actions must share the same incorrect token $s_t$ (given they share the same correct prefix). Observing the error index $t$ for the first action effectively predicts the error index for the second. This reduces the problem to solving $L$ sequential classification problems of size $|\mathcal{S}|$. The effective dimension is the sum of these sub-problems:

$$d_{\text{crit}} = \sum_{t=1}^{L} |\mathcal{S}| = L|\mathcal{S}|. \tag{13}$$

If the critique also provides the correction suggestion for the first error $s_t^*$, the sequence of error indices in an independent action sequence becomes strictly monotonic, further reducing the dimension to $d_{\text{corr}} \approx O(L)$.

**2. Probability of Success**

Let $M$ be the computational budget. The probability of finding the unique optimal solution $a^*$ is equivalent to sampling the correct element from a set of size $d$ without replacement. The probability of success is approximately:

$$P(\text{success}) \approx 1 - \exp\left(-\frac{M}{d}\right). \tag{14}$$

Given the regime $L|\mathcal{S}| \ll M \ll |\mathcal{S}|^L$:

For **Reward-Only Learning**, substituting $d_{\text{std}} = |\mathcal{S}|^L$:

$$P_{\text{std}} \approx 1 - \exp\left(-\frac{M}{|\mathcal{S}|^L}\right) \approx \frac{M}{|\mathcal{S}|^L} \approx 0. \tag{15}$$

For **Critique-Guided Learning**, substituting $d_{\text{crit}} = L|\mathcal{S}|$:

$$P_{\text{crit}} \approx 1 - \exp\left(-\frac{M}{L|\mathcal{S}|}\right) \approx 1. \tag{16}$$

**Conclusion.** The ratio of success probabilities is $\frac{P_{\text{crit}}}{P_{\text{std}}} \approx \frac{|\mathcal{S}|^L}{M} \gg 1$. This confirms that critique-guided refinement is strictly more sample-efficient than standard reward-based generation. □

### F.5. Theoretical Analysis of Convergence and Policy Improvement

Building upon the sample efficiency established in Appendix F.4, we now analyze the optimization dynamics. We prove that while standard GRPO converges given sufficient samples, Critique-GRPO accelerates this process by altering the trajectory distribution. We utilize the framework of Policy Gradient methods (Sutton et al., 1999; Agarwal et al., 2021) to demonstrate that Critique-GRPO learns from *higher quality trajectories*, guaranteeing a steeper monotonic policy improvement.

**Preliminaries and Problem Setup.** We consider the reasoning task as a contextual bandit problem (Lu et al., 2010). Let $\pi_\theta(y|x)$ be the policy parameterized by $\theta$.

**Definition F.7** (Sparse Reward Landscape). The reward function $R(y, x)$ is binary and sparse. Let $\mathcal{Y}_x^*$ be the set of correct reasoning chains.

$$R(y, x) = \mathbb{I}(y \in \mathcal{Y}_x^*). \tag{17}$$

**Definition F.8** (GRPO Gradient Estimate). Following Shao et al. (2024a), GRPO estimates the gradient using a group of $G$ outputs $\{y_1, \ldots, y_G\}$ sampled from $\pi_{\theta_{\text{old}}}$ for a single input $x$. The gradient estimator is:

$$\hat{g}_{\text{GRPO}} = \frac{1}{G} \sum_{i=1}^{G} \hat{A}_i \nabla_\theta \log \pi_\theta(y_i|x), \quad \text{where} \quad \hat{A}_i = \frac{R(y_i) - \mu_R}{\sigma_R + \epsilon}. \tag{18}$$

#### F.5.1. INEFFICIENCY OF STANDARD EXPLORATION

In complex reasoning tasks, the probability of generating a correct solution $y^*$ via random sampling, denoted as $p^* = P_{y \sim \pi_\theta}(y \in \mathcal{Y}_x^*)$, is small but non-negligible. While standard GRPO is an unbiased estimator, its convergence rate is limited by the sparsity of the signal.

**Proposition F.9** (Signal-to-Noise Ratio in Standard GRPO). *For a group size $G$, the expected number of positive rewards in a standard GRPO batch is $\mathbb{E}[N^+] = Gp^*$. The gradient contribution from optimal trajectories is weighted by $p^*$. If $p^* < 1/G$, many batches will contain no positive signal ($\sigma_R = 0$), resulting in wasted computation steps where $\Delta\theta = 0$. Even when $\sigma_R > 0$, the magnitude of the update in the direction of the optimal policy is proportional to $p^*$.*

#### F.5.2. POLICY IMPROVEMENT VIA CRITIQUE-GUIDED REFINEMENT

Critique-GRPO modifies the sampling distribution. Instead of optimizing solely on $y \sim \pi_\theta$, we optimize on a mixture distribution induced by the critique mechanism. Let the refined set be $\mathcal{Y}_{\text{ref}} = \{\text{Refine}(y) \mid y \sim \pi_\theta\}$.

**Lemma F.10** (Trajectory Quality Enhancement). *Let $P_{crit}$ be the success rate of the critique mechanism in generating a correct solution $y^*$. As shown in Proposition 4.1, $P_{crit} \gg p^*$. By including $\mathcal{Y}_{ref}$ in the GRPO group, the effective probability of observing an optimal trajectory in the batch becomes $P_{eff} \approx P_{crit}$. This ensures that the gradient estimator is dominated by high-quality trajectories, significantly reducing the variance of the advantage function and ensuring $\sigma_R > 0$ with high probability.*

We now prove that learning from these higher-quality trajectories leads to superior policy improvement.

**Theorem F.11** (Policy Improvement Lower Bound). *Let $\pi_{old}$ be the current policy and $\pi_{new}$ be the policy after an update step with learning rate $\alpha$. The expected improvement in the objective function $J(\pi)$ is lower-bounded by the quality of the trajectories used in the update. Comparing Standard GRPO and Critique-GRPO:*

$$\Delta J_{std} \geq \alpha \cdot p^* \cdot \mathbb{E}\left[\frac{1}{\sigma_R}\pi_{old}(y^*)\|\nabla \log \pi(y^*)\|^2\right] - O(\alpha^2) \tag{19}$$

$$\Delta J_{crit} \geq \alpha \cdot P_{crit} \cdot \mathbb{E}\left[\frac{1}{\sigma_R}\pi_{old}(y^*)\|\nabla \log \pi(y^*)\|^2\right] - O(\alpha^2) \tag{20}$$

*Since $P_{crit} \gg p^*$, it follows that $\Delta J_{crit} \gg \Delta J_{std}$.*

*Proof.* **Step 1: Performance Difference.** From Kakade & Langford (2002), $J(\pi_{new}) - J(\pi_{old}) \approx \mathbb{E}_{y \sim \rho}[\hat{A}(y)]$, where $\rho$ is the sampling distribution. The update vector $\Delta\theta$ is proportional to the advantage-weighted gradient: $\Delta\theta \propto \sum_{y \in \text{Batch}} \hat{A}(y)\nabla \log \pi(y)$.

**Step 2: Dominance of Optimal Trajectories.** In a sparse reward setting, the advantage of an optimal solution $y^*$ is positive and large, while incorrect solutions have small negative advantages. The update direction is dominated by the term corresponding to $y^*$:

$$\Delta\theta \approx \eta \cdot \hat{A}(y^*)\nabla \log \pi(y^*). \tag{21}$$

The magnitude of this update depends on the frequency of observing $y^*$ in the batch.

**Step 3: Comparison of Sampling Distributions.**

- **Standard GRPO:** Samples $y \sim \pi_{old}$. The probability of observing $y^*$ is $p^*$. The expected update norm is proportional to $p^*\|\nabla \log \pi(y^*)\|$.

- **Critique-GRPO:** Samples from a mixture including refined outputs. The probability of observing $y^*$ is $P_{crit}$. The expected update norm is proportional to $P_{crit}\|\nabla \log \pi(y^*)\|$.

**Conclusion:** The superiority of Critique-GRPO lies in the fact that it learns from a distribution of *higher trajectories*. Even if standard exploration is capable of finding the solution (non-negligible $p^*$), the critique mechanism artificially amplifies the density of optimal solutions in the training data. Substituting the observation probabilities into the Taylor expansion of the objective function:

$$J(\pi_{new}) - J(\pi_{old}) \approx \nabla J(\pi_{old})^\top \Delta\theta \propto P(\text{observing } y^*) \cdot \|\nabla \log \pi(y^*)\|^2. \tag{22}$$

Given $P_{crit} \gg p^*$, the lower bound on policy improvement is strictly higher for Critique-GRPO. $\square$

**Theorem F.12** (Global Convergence via Policy Shaping). *Assuming the function approximation class of $\pi_\theta$ is sufficiently expressive and the learning rate satisfies Robbins-Monro conditions, Critique-GRPO converges to the optimal policy $\pi^*$. The convergence rate is accelerated by a factor of $O(P_{crit}/p^*)$ compared to standard GRPO.*

*Proof.* We utilize the property that Critique-GRPO simulates *Supervised Fine-Tuning (SFT)* on dynamically generated optimal data within the RL loop. From Theorem F.11, the policy probability mass on $y^*$ increases at each step. Since the gradient variance is bounded (due to the high probability of non-zero variance batches via Lemma F.10), stochastic gradient descent converges. Crucially, the effective learning rate on the optimal trajectory $y^*$ is scaled by the critique success rate. Thus, the model requires fewer iterations to reach the same level of performance as standard GRPO, effectively leveraging the higher quality of the critique-refined trajectories. $\square$

## F.6. Theoretical Analysis: Gradient Efficiency and Manifold Preservation

Now we provide a analysis of the optimization dynamics in $\mathcal{J}_{\text{Critique-GRPO}}$ after employing the shaping function on the refinements. We address the dual challenge of (1) efficiently extracting learning signals from off-policy refined responses ($y_{\text{ref}}$) and (2) preventing the catastrophic forgetting (entropy explosion) caused by distribution shift.

### F.6.1. GRADIENT DYNAMICS OF THE SHAPING FUNCTION

Standard Policy Gradient methods maximize $\mathbb{E}[\log \pi_\theta(y)]$. However, for refined responses where $\pi_\theta(y_{\text{ref}}) \ll 1$, standard gradients can be unstable or vanish. We employ the shaping function $\rho_t(\theta) = \frac{\pi_t}{\pi_t + \gamma}$.

**Lemma F.13** (Gradient Rescaling). *Let $\pi_t = \pi_\theta(a_t|s_t)$. The gradient of the shaped objective with respect to the policy parameters $\theta$ is a re-weighted version of the standard policy gradient:*

$$\nabla_\theta \rho_t(\theta) = \underbrace{\left[ \frac{\gamma \pi_t}{(\pi_t + \gamma)^2} \right]}_{\Psi(\pi_t)} \cdot \nabla_\theta \log \pi_t. \tag{23}$$

*Proof.* Applying the quotient rule to $\rho_t$ with respect to $\pi_t$:

$$\frac{\partial \rho_t}{\partial \pi_t} = \frac{1 \cdot (\pi_t + \gamma) - \pi_t \cdot 1}{(\pi_t + \gamma)^2} = \frac{\gamma}{(\pi_t + \gamma)^2}. \tag{24}$$

By the chain rule, $\nabla_\theta \rho_t = \frac{\partial \rho_t}{\partial \pi_t} \nabla_\theta \pi_t$. Using the log-derivative identity $\nabla_\theta \pi_t = \pi_t \nabla_\theta \log \pi_t$, we substitute to obtain:

$$\nabla_\theta \rho_t = \frac{\gamma \pi_t}{(\pi_t + \gamma)^2} \nabla_\theta \log \pi_t. \tag{25}$$

$\square$

**Analysis of the Modulation Term $\Psi(\pi_t)$.** The coefficient $\Psi(\pi_t)$ acts as a dynamic gain controller, theoretically aligning with the "hard negative mining" mechanism found in Focal Loss (Lin et al., 2017).

1. **Suppression of Easy Tokens ($\pi_t \to 1$):** $\lim_{\pi_t \to 1} \Psi(\pi_t) \approx \frac{\gamma}{(1+\gamma)^2}$. Since $\gamma \ll 1$, the gradient magnitude is significantly dampened. This prevents the model from overfitting to tokens it has already mastered, preserving the KL-divergence on trivial tokens.

2. **Peak Efficiency in the Learning Zone:** Solving $\frac{d\Psi}{d\pi} = 0$ yields a maximum at $\pi_t = \gamma$. By setting $\gamma \approx 0.1$, we explicitly maximize gradients for "correction tokens"—those that the current policy considers unlikely but plausible. This focuses the update on the *reasoning gap* between the initial and refined response.

### F.6.2. THE INSUFFICIENCY OF CLIPPING FOR OFF-POLICY DATA

Standard PPO/GRPO relies on clipping the probability ratio $r_t$ to $[1 - \epsilon, 1 + \epsilon]$ to enforce a trust region (Schulman et al., 2017). While effective for on-policy data, we prove that clipping is insufficient for off-policy refinements due to vector orthogonality.

**Proposition F.14** (Gradient Orthogonality). *Let $\mathcal{M}_{old}$ be the manifold of the current policy. The gradient update from refinements, $\Delta\theta_{ref}$, is often orthogonal to the update from initial responses, $\Delta\theta_{init}$.*

*Reasoning:* $\Delta\theta_{\text{init}}$ reinforces the current mode of the distribution. Conversely, $\Delta\theta_{\text{ref}}$ attempts to shift probability mass to a disjoint region of the sample space ($y_{\text{ref}}$). Optimizing $\Delta\theta_{\text{ref}}$ inherently requires reducing the probability of $y_{\text{init}}$. Therefore:

$$\langle \nabla_\theta \mathcal{J}_{\text{init}}, \nabla_\theta \mathcal{J}_{\text{ref}} \rangle < 0. \tag{26}$$

Clipping bounds the *magnitude* of the step but does not correct this conflicting *direction*. Without regularization, the accumulation of $\Delta\theta_{\text{ref}}$ updates drags the policy off the manifold $\mathcal{M}_{\text{old}}$, leading to entropy collapse.

F.6.3. MANIFOLD PRESERVATION VIA THE ANCHOR HYPOTHESIS

To prevent this collapse, we derive the necessity of the sample prioritization ratio $N_{\text{init}} \gg N_{\text{ref}}$.

**Theorem F.15** (The Anchor Condition). *Let $\mathbf{F}$ be the Fisher Information Matrix of the policy. To ensure the updated policy $\pi_{\theta'}$ remains within a trust region $D_{KL}(\pi_{\theta'}||\pi_\theta) \leq \delta$, the influence of refined samples must be bounded by the curvature of the on-policy manifold.*

*Proof.* We approximate the KL divergence using the second-order Taylor expansion (Amari et al., 2019; Kakade & Langford, 2002):

$$D_{\text{KL}}(\pi_{\theta+\Delta\theta}||\pi_\theta) \approx \frac{1}{2}\Delta\theta^T\mathbf{F}\Delta\theta. \tag{27}$$

The total update is a convex combination: $\Delta\theta \propto (1-\lambda)g_{\text{init}} + \lambda g_{\text{ref}}$, where $\lambda = N_{\text{ref}}/N$. Since $g_{\text{init}}$ is clipped on-policy, its contribution to the divergence is bounded by design. The risk arises from $g_{\text{ref}}$, which is high-variance and off-distribution. If $\lambda$ is large, the term $\lambda g_{\text{ref}}$ dominates. Since $g_{\text{ref}}$ is not aligned with the eigenvectors of $\mathbf{F}$ (which represent the current policy's geometry), the term $\Delta\theta^T\mathbf{F}\Delta\theta$ grows rapidly, signifying a departure from the trust region. By enforcing $N_{\text{init}} \gg N_{\text{ref}}$ (small $\lambda$), the term $(1-\lambda)g_{\text{init}}$ acts as a regularization term (an "Anchor"). It ensures that the primary update direction respects the local curvature $\mathbf{F}$, while $g_{\text{ref}}$ acts as a small perturbation vector that rotates the policy toward higher reward regions without shattering the manifold. $\square$

## G. Implementation Details

*Table 13.* Default hyperparameters and training configurations used in Critique-GRPO.

| Name | Value (for Qwen2.5-7B-Base or Qwen2.5-Math-7B-Base) | Value (for Qwen3-8B) | Description |
|---|---|---|---|
| **RL Finetuning** | | | |
| num_training_prompts | 4k | 4k | Default number of training prompts (unless specified otherwise). |
| training_steps | 400 | 300 | Total number of training steps. |
| eval_freq | 20 | 25 | Frequency of evaluations (in updates). |
| batch_size | 128 | 128 | Accumulated batch size during training. |
| lr | $1e^{-6}$ | $1e^{-6}$ | Learning rate. |
| max_prompt_length | 1024 | 1024 | Maximum input context length. |
| max_response_length | 8192 | 6144 | Maximum length of generated responses. |
| n_rollouts | 7 | 7 | Number of rollouts per prompt. |
| n_refinements | 1 | 1 | Number of refinements per prompt. |
| rewards | 1 or 0 | 1 or 0 | Scalar rewards for responses. |
| kl_loss_coef | 0.0 | 0.0 | Coefficient for KL divergence loss. |
| $\gamma$ | 0.1 | 0.1 | Hyperparameter in the policy shaping function. |
| train_temp | 1.0 | 1.0 | Sampling temperature during rollouts. |
| val_temp | 0.6 | 0.6 | Sampling temperature during validation. |
| total_epochs | 30 | 30 | Total number of training epochs. |
| **Evaluation** | | | |
| eval_temp | 0.0 | 0.0 | Sampling temperature during evaluation. |
| max_tokens | 8192 | 8192 | Inference token budget during evaluation. |

**Datasets and Evaluation Metrics.** We use randomly sampled subsets of 4k examples from a reorganized 45k subset (Yan et al., 2025) of OpenR1-Math-220k (Bakouch et al., 2025) as the training set (as described in Section 3). For validation, we use the curated validation set provided by (Yan et al., 2025). We evaluate the model on five well-established mathematical reasoning benchmarks: MATH-500 (Hendrycks et al., 2021), Minerva-Math (Lewkowycz et al., 2022), OlympiadBench (He et al., 2024a), AIME 2024 (Li et al., 2024), AIME 2025 (Li et al., 2024), and AMC 2023 (Li et al., 2024). For broader analysis, we assess the model's generalization ability on three scientific and general reasoning tasks: TheoremQA (Math, Physics, EE&CS, and Finance) (Chen et al., 2023), GPQA-Diamond (Physics, Chemistry, and Biology) (Rein et al., 2024), and MMLU-Pro (Business, Computer Science, Law, *etc.*) (Wang et al., 2024). During evaluation, we use greedy decoding (temperature = 0) and report pass@1 over three runs.

**Reward Design.** During RL fine-tuning, we use model-based evaluation to generate critiques and rule-based evaluation to provide binary scalar rewards, as described in Section 3.

**Compared Methods.** We compare Critique-GRPO against the following representative approaches, categorized into supervised learning and reinforcement learning-based finetuning. All differences are considered significant at $p < 0.01$.

***Supervised Learning-based Finetuning:***

(i) *Supervised Finetuning (SFT)*: Finetuning the base model on high-quality annotated training data using supervised learning.

(ii) *Reward rAnked Finetuning (RAFT)* (Dong et al., 2023): Finetuning on self-generated correct responses, sampled based on rule-based evaluation.

(iii) *Refinement Finetuning (Refinement FT)* (Chen et al., 2024): Finetuning on refined correct responses generated conditionally on the question, initial response, and CoT critiques.

(iv) *Critique Finetuning (Critique FT)* (Wang et al., 2025): Finetuning on annotated CoT critique data to train the model to critique a given query-response.

(v) *Critique-in-the-Loop Finetuning (CITL-FT)* (Xi et al., 2024): Finetuning on mixed data consisting of self-generated correct responses and refined correct responses, conditioned on the question-initial response-CoT critique triplet.

*Reinforcement Learning-based Finetuning:*

(vi) *R1-GRPO* (DeepSeek-AI et al., 2025): Finetuning the base model on its own generations using the GRPO algorithm with binary scalar rewards.

(vii) *R1-Dr.GRPO* (Liu et al., 2025a): Finetuning the base model on its own generations using the Dr.GRPO algorithm, which removes terms that cause biased optimization, with binary scalar rewards.

(viii) *Critique-GRPO (Indicative Critique)*: Fine-tuning the base model with Critique-GRPO by utilizing indicative critiques for refinements, as described in Appendix E.

(ix) *Critique-GRPO (Critique with Ground Truth)*: Fine-tuning the base model with Critique-GRPO by leveraging indicative critiques alongside ground-truth answers for refinements, as detailed in Appendix E.

**Implementation Details.** We conduct experiments using Qwen2.5-7B-Base, Qwen2.5-Math-7B-Base, and Qwen3-8B, with GPT-4o (which can be replaced by other reasoning-based reward models) serving as the reward model, as described in Section 3. For supervised finetuning baselines, models are finetuned until convergence, and the best performance is reported. For RL-based approaches, models are finetuned for 400 steps, and the best performance is recorded. To ensure a fair comparison: In R1-GRPO, 8 responses (rollouts) are sampled per training prompt with a temperature of 1. In Critique-GRPO, 7 responses are sampled per prompt, along with one refined response from the refinement sets. we present detailed hyperparameters and training configurations in Table 13.

All experiments are conducted on 40 NVIDIA A800 80G GPUs. To ensure consistency, we use only critiques generated by the reward model that align with rule-based evaluations; otherwise, the reward model is prompted to regenerate the critiques. Following prior works (Liu et al., 2025a), for evaluation, we adopt greedy sampling (temperature set to 0) to generate responses and report pass@1 as the evaluation metric. When reporting pass@k, we uniformly set the temperature to 0.6 and the top-p value to 0.95.

**Compared Methods in the Investigation of Math-Centric Backbone Models.** We evaluate the efficacy of RL fine-tuning with Critique-GRPO on the math-centric backbone model, Qwen2.5-Math-7B-Base. Specifically, we compare its performance against three representative RL fine-tuning approaches based solely on numerical feedback:

(i) *SimpleRL-Zero* (Zeng et al., 2025): an open-source reproduction of R1-GRPO.

(ii) *PRIME-Zero* (Cui et al., 2025a): fine-tuning the base model using both outcome binary rewards and process binary rewards.

(iii) *Oat-Zero* (Liu et al., 2025a): fine-tuning the base model with Dr.GRPO using outcome binary rewards.

## H. Detailed Results and Analysis of Self-Critiquing Mechanisms

*Table 14.* Potential for self-improvement through RL fine-tuning using Critique-GRPO with self-generated CoT critiques (self-critiquing) on Qwen3-8B.

| Method | w/ External Supervision | Math (ID) | | | | | Science & General (OOD) | | | Avg. |
| | | MATH 500 | Minerva MATH | Olympiad Bench | AMC23 | AIME24 | Theorem QA | GPQA Diamond | MMLU Pro | |
| --- | --- | --- | --- | --- | --- | --- | --- | --- | --- | --- |
| Qwen3-8B (w/ Think) | - | 82.00 | 41.20 | 44.10 | 67.50 | 40.00 | 46.90 | 35.86 | 68.25 | 53.23 |
| + SFT | ✓ | 83.20 | 43.80 | 46.40 | 82.50 | 40.00 | 48.90 | 38.38 | 66.81 | 56.25 |
| + R1-GRPO | ✓ | 91.00 | 52.60 | 65.60 | 82.50 | 50.00 | 57.90 | 40.40 | 70.00 | 63.75 |
| + Critique-GRPO (Ours) (Self-Critique& Self-Evaluation | × | 92.00 | 52.20 | 65.50 | 87.50 | 53.30 | 59.80 | 47.47 | **70.93** | 66.09 |
| + Critique-GRPO (Ours) (Self-Critique) | ✓ | **92.60** | **52.60** | **66.20** | **95.00** | **60.00** | **60.60** | **47.98** | 70.03 | **68.13** |

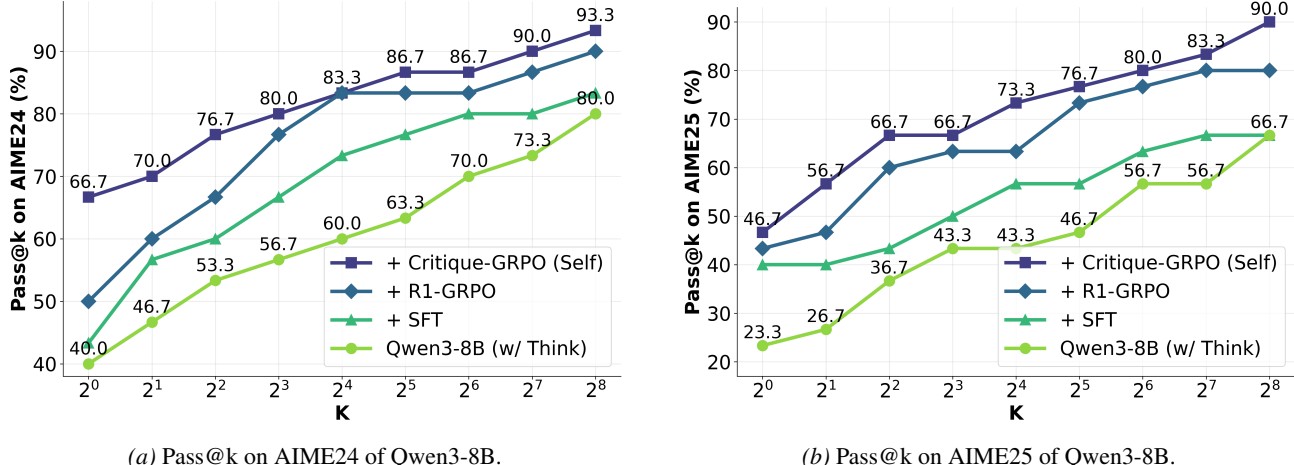

*(a) Pass@k on AIME24 of Qwen3-8B.*      *(b) Pass@k on AIME25 of Qwen3-8B.*

*Figure 9.* Comparison of Pass@k for self-improvement using RL fine-tuning via Critique-GRPO (self-critique), compared to methods relying on external numerical feedback (R1-GRPO), expert demonstrations (SFT), the base model Qwen3-8B.

To explore the potential of Critique-GRPO in enabling an LLM's self-improvement through self-critiquing, we prompt the model itself to serve as a reasoning-based reward model. Specifically, we investigate two types of self-critiquing: (i) Self-critique, where the model evaluates the correctness of its own responses using CoT critiques with ground truth answers as reference; and (ii) Self-critique & self-evaluation (Zhang et al., 2024), where the model evaluates its responses using CoT critiques *without* any reference. These approaches result in Critique-GRPO (self-critique) and Critique-GRPO (self-critique & self-evaluation), respectively. Details of the prompts are provided in Appendix M. Table 14 shows the evaluation results on Qwen3-8B, and Figure 9 presents pass@k performance changes on AIME24 and AIME25 (Li et al., 2024).

**Critique-GRPO enhances self-improvement through self-critiquing.** Table 14 RL fine-tuning with Critique-GRPO (self-critique) significantly outperforms fine-tuning with GRPO using external numerical feedback (R1-GRPO) and supervised fine-tuning with expert demonstrations (SFT). On average, Critique-GRPO (self-critique) improves pass@1 by +4.5% and +12.0% compared to R1-GRPO and SFT, respectively. Additionally, the unsupervised approach—Critique-GRPO (self-critique & self-evaluation)—achieves an average pass@1 improvement of 2.3% over R1-GRPO, highlighting the potential of leveraging self-critique for self-improvement *without any external supervision*.

**Self-critiquing aids effective exploration.** Figure 9 highlights the consistently superior performance of Critique-GRPO (self-critique) across pass@k metrics, with $k$ ranging from 1 to 256, indicating genuine improvements. Notably, Critique-GRPO (self-critique) achieves remarkable gains over R1-GRPO for pass@k with $k = 1$ to 4, yielding improvements of 10-16.7% on AIME24 (Figure 9a).

# I. Detailed Investigation of Policy Exploration During RL Finetuning

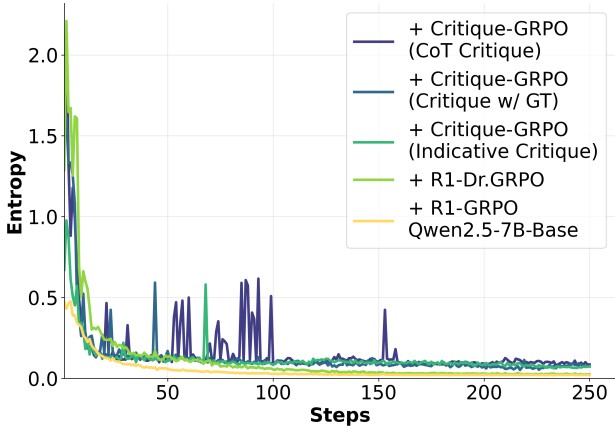
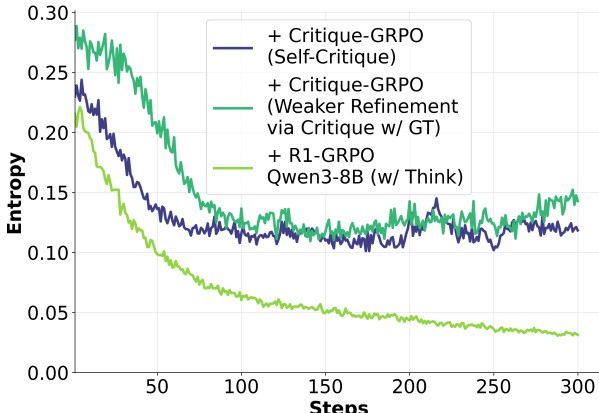

*(a)* Entropy dynamics for RL-based finetuning approaches over training steps on Qwen2.5-7B-Base.

*(b)* Entropy dynamics of self-improvement through self-critiquing and weak-to-strong generalization across training steps on Qwen3-8B.

*Figure 10.* Entropy dynamics during RL finetuning. A comparison of RL-based finetuning approaches for self-improvement on Qwen2.5-7B-Base (left) and self-improvement through self-critiquing with Critique-GRPO on Qwen3-8B (right).

To investigate policy exploration, we analyze two primary aspects of our RL-finetuned models: (i) entropy dynamics during RL fine-tuning for self-improvement using compared RL-based finetuning approaches on Qwen2.5-7B-Base and Critique-GRPO (self-critique) on Qwen3-8B (Figure 10), and (ii) changes in response length during fine-tuning (Figure 11).

**Learning from natural language feedback helps sustain exploration.** As shown in Figure 10a, the policy entropy of Critique-GRPO generally remains higher than that of R1-GRPO and R1-Dr.GRPO, suggesting more consistent exploration. The peaks in Critique-GRPO's entropy dynamics (before step 200) likely occur when its self-generated refinements deviate significantly from the initial sampled responses, leading to increased entropy and potentially beneficial distributional shifts. The subsequent decrease in entropy indicates that the model quickly internalizes these refinements, reducing the distributional deviation. This dynamic aligns with the observation that rare actions with high advantage can increase policy entropy (*i.e.,* unfamiliar but correct responses with high rewards promote *effective exploration*), whereas high-probability actions with high advantage tend to reduce entropy (Cui et al., 2025b). In contrast, R1-GRPO exhibits *entropy collapse*, where policy entropy drops sharply at the start of training and continues to decline monotonically to near zero. R1-Dr.GRPO initially exhibits higher entropy (before step 50) but rapidly drops to comparable near-zero values with R1-GRPO after step 150. Combined with the results in Table 2, the superior performance of Critique-GRPO over R1-Dr.GRPO and R1-GRPO highlights the importance of maintaining a certain level of entropy for better performance.

**Learning through self-critiquing facilitates policy exploration.** Figure 10b shows that Critique-GRPO (self-critique) avoids entropy collapse and maintains higher entropy than R1-GRPO. This finding aligns with the observation that increased exploration improves performance.

**Higher entropy does not always guarantee effective exploration.** Unexpectedly, as shown in Figure 10b, Critique-GRPO (weaker refinement via critique with ground truth), shown in green, achieves higher entropy than Critique-GRPO (self-critique), shown in dark blue, yet performs worse (average pass@1: 65.55% *v.s.* 68.13%). This discrepancy may be due to refinements from weaker models causing larger distributional shifts compared to self-refinements, while also being of lower quality. This suggests that the *quality* of exploration signals is more critical than the *extent* of exploration (as reflected solely by entropy).

**Critique-GRPO facilitates concise reasoning.** In Figure 11, Critique-GRPO achieves superior performance (Table 2) while minimally increasing response length on Qwen2.5-7B-Base (Figure 11a). This efficiency likely stems from its critique mechanism, which enables precise error identification and refinement, reducing the need for verbose reasoning. Additionally, Critique-GRPO tends to reduce response length on Qwen3-8B (Figure 11b). This trend can be attributed to the correction of Qwen3-8B's tendency toward redundant and ineffective self-reflection, as discussed in Appendix J.

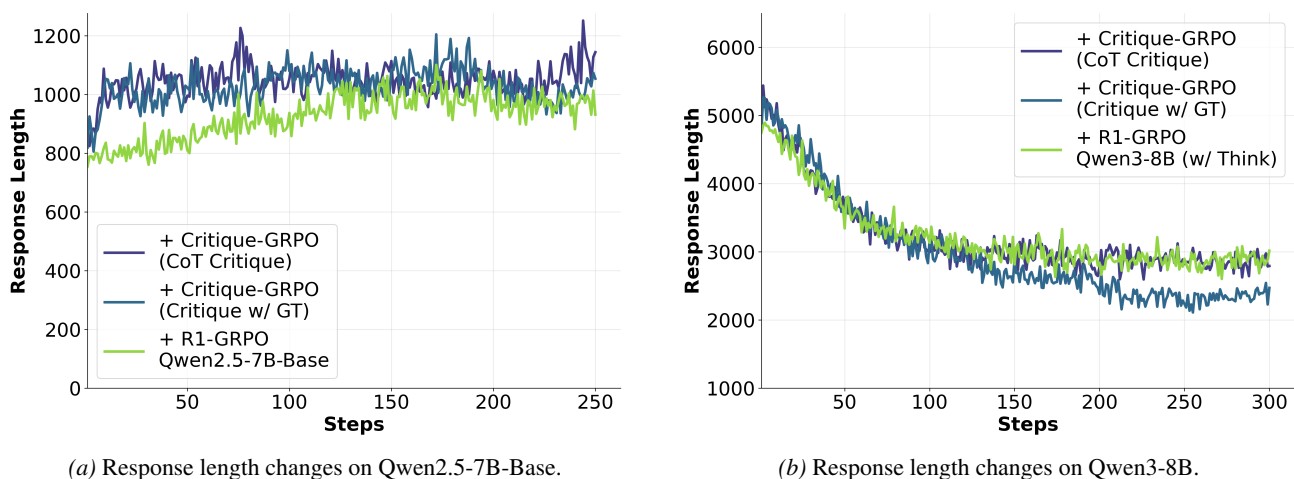

*(a)* Response length changes on Qwen2.5-7B-Base.

*(b)* Response length changes on Qwen3-8B.

*Figure 11.* Comparison of response length changes during RL finetuning on Qwen2.5-7B-Base (left) and Qwen3-8B (right).

## J. Qualitative Analysis

**Fine-Grained Analysis.** We conduct a fine-grained analysis of 100 generated responses on the Minerva-MATH dataset across four key dimensions: factuality, conciseness, correctness, and logicality, using the prompt in Appendix M. Figure 12 shows that fine-tuning with Critique-GRPO on Qwen2.5-7B-Base achieves the best performance across all four dimensions as well as in average performance. Additionally, the superior performance of RL fine-tuning with Critique-GRPO over R1-GRPO in terms of factuality and correctness indicates that CoT critiques help the model effectively identify errors and improve valid exploration. The inferior performance of SFT in logicality and conciseness may be attributed to the presence of redundant and sometimes illogical self-reflective reasoning behaviors in expert demonstrations, which could negatively impact user experience.

**Case Study.** We present a qualitative comparison between the responses generated by the base Qwen3-8B model and the RL-finetuned Qwen3-8B model using Critique-GRPO on the MATH-500 dataset. The given question is: `Evaluate` $\sin(\arcsin 0.4 + \arcsin 0.5), \sin(\arcsin 0.5 - \arcsin 0.4)$.

The base Qwen3-8B model produces an incorrect response due to an incorrect formulation of the expression as $\sin(A + B) \cdot$ $\sin(A - B)$, as shown in the red square. In contrast, the RL-finetuned Qwen3-8B model using Critique-GRPO generates a correct response by correctly formulating the expression as $\sin(\alpha + \beta) \cdot \sin(\beta - \alpha)$, as shown in the green square. Furthermore, the base Qwen3-8B model exhibits numerous redundant and ineffective self-reflection attempts (*e.g.,* "Wait, ..." highlighted in blue), which fail to help the model identify the actual errors and answer the question correctly. This results in an excessively long response (over 6000 tokens). In contrast, the RL-finetuned Qwen3-8B model using Critique-GRPO demonstrates concise and effective reasoning. It remains on the correct path to solve the problem, exhibits valid self-reflection to validate the answer (*e.g.,* "Let me check with approximate values. ..."), and ultimately generates the correct answer. Detailed responses are provided in Appendix O.

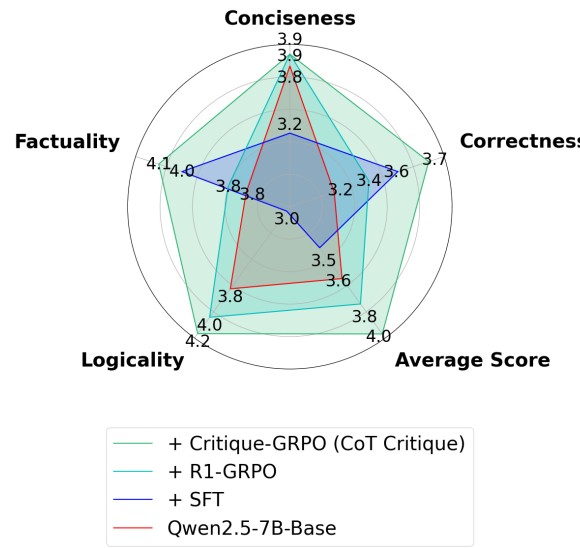

*Figure 12.* Qualitative analysis of generated responses on Minerva-Math across four dimensions: logicality, factuality, conciseness, correctness, and average performance.

**Response Generated by Base Qwen3-8B (Abridged)**

Okay, so I need to evaluate this expression: $\sin(\arcsin 0.4 + \arcsin 0.5)$ multiplied by $\sin(\arcsin 0.5 - \arcsin 0.4)$. Hmm, that looks a bit complicated with all those inverse sine functions. Let me try to break it down step by step. First, let me recall some trigonometric identities that might help here. I remember that there are formulas for $\sin(A + B)$ and $\sin(A - B)$. Maybe I can use those? Let me write them down: $\sin(A + B) = \sin A \cos B + \cos A \sin B$, $\sin(A - B) = \sin A \cos B - \cos A \sin B$

So if I let $A = \arcsin 0.4$ and $B = \arcsin 0.5$, then the expression becomes:

$$\sin(A + B) \cdot \sin(A - B)$$

Wait, so maybe I can compute $\sin(A + B)$ and $\sin(A - B)$ separately using those identities and then multiply them together? But before that, maybe I should figure out what $A$ and $B$ are? Well, $A$ is $\arcsin 0.4$, which means that $\sin A = 0.4$, and similarly $\sin B = 0.5$. But I need $\cos A$ and $\cos B$ to use the identities. Since $\sin^2 \theta + \cos^2 \theta = 1$, I can find $\cos A$ and $\cos B$. Let me compute $\cos A$ first. For angle $A$, $\sin A = 0.4$, so

$$\cos A = \sqrt{1 - \sin^2 A} = \sqrt{1 - 0.16} = \sqrt{0.84}.$$

Wait, $\sqrt{0.84}$... So $\cos B = \frac{\sqrt{3}}{2}$.

...

Okay, so now I can compute $\sin(A + B)$ and $\sin(A - B)$ using the identities.

...

So

$$\sin(A + B) = 0.2\sqrt{3} + \frac{\sqrt{21}}{10}$$

...

Okay, now moving on to $\sin(A - B)$:

$$\sin(A - B) = \sin A \cos B - \cos A \sin B$$

...

So

$$\sin(A - B) = \frac{2\sqrt{3} - \sqrt{21}}{10}$$

Therefore, the original expression is:

$$\sin(A + B) \cdot \sin(A - B) = \left(\frac{2\sqrt{3} + \sqrt{21}}{10}\right) \cdot \left(\frac{2\sqrt{3} - \sqrt{21}}{10}\right)$$

...

That seems correct. But let me verify with another approach. Maybe using another identity? Alternatively, I remember that $\sin(A + B) \cdot \sin(A - B)$ can be expressed as $\sin^2 A - \sin^2 B$. Wait, is that correct? Wait, let me recall the identity:

$$\sin(A + B) \cdot \sin(A - B) = \sin^2 A - \sin^2 B$$

Let me check that. Yes, because:

$$\sin(A + B) \cdot \sin(A - B) = [\sin A \cos B + \cos A \sin B][\sin A \cos B - \cos A \sin B]$$

$$= (\sin A \cos B)^2 - (\cos A \sin B)^2 = \sin^2 A \cos^2 B - \cos^2 A \sin^2 B$$

...

Let me compute $\sin^2 A - \sin^2 B$:

$$\sin^2 A - \sin^2 B = (0.4)^2 - (0.5)^2 = 0.16 - 0.25 = -0.09 = \frac{-9}{100}$$

...

**Response Generated by RL-Finetuned Qwen3-8B using Critique-GRPO (Abridged)**

Okay, so I need to evaluate this expression: $\sin(\arcsin 0.4 + \arcsin 0.5)$ multiplied by $\sin(\arcsin 0.5 - \arcsin 0.4)$. Hmm, let me think.

First, I remember that `arcsin` is the inverse sine function, so $\arcsin 0.4$ is an angle whose sine is $0.4$, and similarly for $\arcsin 0.5$. Let me denote some variables to make this easier. Let me set $\alpha = \arcsin 0.4$ and $\beta = \arcsin 0.5$. Then the expression becomes $\sin(\alpha + \beta) * \sin(\beta - \alpha)$.

Alright, so I need to compute $\sin(\alpha+\beta)$ and $\sin(\beta-\alpha)$ separately and then multiply them. Maybe I can use sine–addition formulas here. Let me recall that $\sin(A+B) = \sin A \cos B + \cos A \sin B$, and $\sin(A-B) = \sin A \cos B - \cos A \sin B$.

Since $\alpha$ and $\beta$ are `arcsin` values, I know that $\sin \alpha = 0.4$ and $\sin \beta = 0.5$. Therefore, I need to find $\cos \alpha$ and $\cos \beta$ to use in the formulas.

For $\cos \alpha$, since $\sin^2 \alpha + \cos^2 \alpha = 1$, we have

$$\cos \alpha = \sqrt{1 - \sin^2 \alpha} = \sqrt{1 - 0.16} = \sqrt{0.84}.$$

Note that $0.84 = \frac{84}{100} = \frac{21}{25}$, so $\cos \alpha = \sqrt{\frac{21}{25}} = \frac{\sqrt{21}}{5}$.

Similarly,

$$\cos \beta = \sqrt{1 - \sin^2 \beta} = \sqrt{1 - 0.25} = \sqrt{0.75} = \sqrt{\tfrac{3}{4}} = \frac{\sqrt{3}}{2}.$$

Now, let me compute $\sin(\alpha + \beta)$. Using the formula,

$$\sin(\alpha + \beta) = \sin \alpha \cos \beta + \cos \alpha \sin \beta = 0.4\left(\frac{\sqrt{3}}{2}\right) + \left(\frac{\sqrt{21}}{5}\right)(0.5).$$

Compute each term:

$$0.4 \cdot \frac{\sqrt{3}}{2} = \frac{2}{5} \cdot \frac{\sqrt{3}}{2} = \frac{\sqrt{3}}{5}, \quad \frac{\sqrt{21}}{5} \cdot 0.5 = \frac{\sqrt{21}}{10}.$$

Therefore

$$\sin(\alpha + \beta) = \frac{\sqrt{3}}{5} + \frac{\sqrt{21}}{10} = \frac{2\sqrt{3} + \sqrt{21}}{10}.$$

Similarly, compute $\sin(\beta - \alpha)$:

$$\sin(\beta - \alpha) = \sin \beta \cos \alpha - \cos \beta \sin \alpha = 0.5\left(\frac{\sqrt{21}}{5}\right) - \frac{\sqrt{3}}{2}(0.4).$$

Thus

$$\sin(\beta - \alpha) = \frac{\sqrt{21}}{10} - \frac{2\sqrt{3}}{10} = \frac{\sqrt{21} - 2\sqrt{3}}{10}.$$

Therefore, the original expression is

$$\left(\frac{2\sqrt{3} + \sqrt{21}}{10}\right)\left(\frac{\sqrt{21} - 2\sqrt{3}}{10}\right).$$

Let $A = 2\sqrt{3} + \sqrt{21}$ and $B = \sqrt{21} - 2\sqrt{3}$; then the product is $\frac{AB}{100}$. Note that $AB = (\sqrt{21})^2 - (2\sqrt{3})^2 = 21 - 12 = 9$, so the value is $\frac{9}{100}$.

...

# K. Limitations

While Critique-GRPO establishes a promising foundation for leveraging both natural language and numerical feedback, notable limitations remain.

**Performance limitations due to failed refinements.** Policy models sometimes fail to follow CoT critiques to refine their responses. We attribute this to the lack of deliberate training for self-refinement. An example of a failed refinement is provided in Appendix P. Future work could focus on improving the model's refinement capabilities or training a specialized model dedicated to refinement tasks.

**The role of critique detail in refinement quality.** We currently utilize three types of critiques (see Section 3), with CoT critiques demonstrating the greatest benefits for refinement. This advantage likely stems from their detailed step-by-step evaluations and concise improvement suggestions, which help models identify and correct errors in initial responses. It follows that more detailed critiques could result in higher-quality refinements. For simplicity, we use GPT-4o as the reasoning-based reward model, *not for expert knowledge distillation*. Consequently, the generated CoT critiques do not include expert demonstrations. Future work may explore alternative reasoning-based reward models. One might assume that directly incorporating expert demonstrations into critiques would significantly improve performance. However, our experiments reveal otherwise. Upon analyzing the generated refinements, we observe that both pre-trained models (*e.g.,* Qwen2.5-7B-Base) and alignment-tuned models (*e.g.,* Qwen3-8B) tend to produce conclusive sentences and correct answers as refinements, rather than detailed step-by-step reasoning to derive the correct answer. This behavior limits the effectiveness of expert demonstrations as critiques.

Future work could investigate, in greater depth, which types of critiques provide the most significant benefits for refinement, particularly in reasoning-intensive tasks.

# L. Analysis of Cognitive Behaviors

To systematically investigate this question, we characterize six key cognitive behaviors that contribute to self-improving reasoning during RL fine-tuning, as follows:

- **Subgoal Setting**: Breaking down complex problems into smaller, manageable steps or subtasks. For example, "Step 1... Step 2..."

- **Summarization**: Summarizing the current state by identifying completed subtasks and determining what remains to be done. This helps guide the next steps in reasoning. For example, "Now we have obtained..., next, we need to..."

- **Verification**: Systematically checking intermediate results or computations to ensure correctness. For example, "Let's verify this result by..."

- **Backtracking**: Identifying errors or dead-ends in reasoning and explicitly revising previous methods or approaches. For example, "This approach won't work because..., let's try another method..."

- **Backward Chaining**: Reasoning from desired outcomes back to initial inputs or steps required to achieve the result. This is particularly applicable to multiple-choice questions where answer options are provided. For example, "To get 24, I could do 24 ÷ 2 = 12..." (Gandhi et al., 2025)

- **Anticipation**: Anticipating potential inaccuracies or exhaustively considering multiple possibilities to solve a problem. For example, "Alternatively, this problem can be solved by..."

We analyze the reasoning (cognitive) behaviors using the prompts shown below.

When assessing the contributions of reasoning behaviors in Section 3 to successful problem-solving in RL fine-tuned models, we count each behavior appearing in the generated responses *only once*. For example, if the model produces multiple subgoals in a single response, the occurrence of "subgoal setting" is counted as one.

## Prompts for Analyzing Reasoning Behaviors

**System:** You are a helpful assistant.
**User:** The following is a chain-of-thought produced by a language model in response to a math & science problem:
**Question:** ¡Question Content¿
**Reasoning:** ¡Model Reasoning¿
**Ground Truth:** ¡Ground Truth Content¿

**Task 1: Answer Verification**
Determine whether the reasoning includes any *explicit or implicit answer verification steps* — moments where the model checks intermediate computations or final results for correctness.
Example: "Let's verify this result by..."
- Report the number of distinct answer verification steps using: `<count>n</count>`. If none are found, return `<count>0</count>`. - If such behavior is present and the final answer matches the ground truth, indicate whether the behavior contributed to the correct answer using the format: `contribution: yes/no`.

**Task 2: Backtracking Behavior**
Determine whether the reasoning demonstrates *backtracking* — where the model identifies an error or dead end and switches to a different approach.
Example: "This approach won't work because..., let's try another method..."
- Report the number of distinct backtracking instances using: `<count>n</count>`. If none are found, return `<count>0</count>`. - If such behavior is present and the final answer matches the ground truth, indicate whether the behavior contributed to the correct answer using the format: `contribution: yes/no`.

**Task 3: Subgoal Setting**
Determine whether the reasoning includes any *explicit subgoals* — intermediate steps that break the problem into smaller, manageable parts.
Example: "First, I'll try to..., then I'll..."
- Report the number of clearly defined subgoals using: `<count>n</count>`. If none are found, return `<count>0</count>`. - If such behavior is present and the final answer matches the ground truth, indicate whether the behavior contributed to the correct answer using the format: `contribution: yes/no`.

**Task 4: Backward Chaining**
Determine whether the reasoning includes *backward chaining* — starting from the target result and reasoning backward to infer inputs or steps.
Example: "To get 24, I could do $24 \div 2 = 12$..."
- Report the number of distinct backward chaining attempts using: `<count>n</count>`. If none are found, return `<count>0</count>`. - If such behavior is present and the final answer matches the ground truth, indicate whether the behavior contributed to the correct answer using the format: `contribution: yes/no`.

**Task 5: Anticipation**
Determine whether the reasoning includes *enumeration* or *anticipation and re-proposal* — suggesting alternative approaches or revising prior methods.
Examples: "Alternatively, this problem can be solved by...", "Let's try a different approach..."
- Report the number of such instances using: `<count>n</count>`. If none are found, return `<count>0</count>`. - If such behavior is present and the final answer matches the ground truth, indicate whether the behavior contributed to the correct answer using the format: `contribution: yes/no`.

**Task 6: Summarization**
Determine whether the reasoning includes *summarization* — identifying completed subtasks, summarizing progress, and determining the next steps.
Example: "Now we have obtained..., next, we need to..."
- Report the number of summarization instances using: `<count>n</count>`. If none are found, return `<count>0</count>`. - If such behavior is present and the final answer matches the ground truth, indicate whether the behavior contributed to the correct answer using the format: `contribution: yes/no`.

# M. Prompts

**Training Prompt.**    The following training prompt is used during all RL fine-tuning experiments:

---

**Training Prompt**

**System:** You are a helpful assistant.
**User:** `<Question Content>`
Please reason step by step and place your final answer within `\boxed`.

---

**Prompt for Generating Chain-of-Thought Critique.**    We adopt a prompt inspired by (Wang et al., 2025) to enable GPT-4o (Hurst et al., 2024) to generate CoT critiques. For quality control, we retained only those model-generated critiques whose evaluative conclusions (correct/incorrect) aligned with rule-based verification. When inconsistencies occurred, we prompted the critique model to regenerate the critiques.

---

**Prompt for Generating Chain-of-Thought Critique**

**System:** You are a science expert. A student is trying to solve a question. Please explain briefly (step-by-step) whether the student's solution is correct or not. Finally, conclude your judgment with:
"Conclusion: `correct/incorrect` [END]."
**User:**
Question: `<Question Content>`
Ground Truth Answer: `<Ground Truth>`
Student's Solution: `<Generated Solution>`
Critique:

---

**Prompt for Generating Chain-of-Thought Critique with Internal Knowledge**    The following prompt is designed to enable an LLM to leverage its internal knowledge and evaluate the correctness of its own generated responses through step-by-step CoT critiques.

---

**Prompt for Generating Chain-of-Thought Critique with Internal Knowledge**

**System:** You are a science expert. A student is trying to solve a question. Please explain briefly (step-by-step) whether the student's solution is correct or not. Finally, conclude your judgment with:
"Conclusion: `correct/incorrect` [END]."
**User:**
Question: `<Question Content>`
Student's Solution: `<Generated Solution>`
Critique:

---

**Refinement Prompt.**    The following refinement prompt is used to guide the model in improving its response by incorporating the critique.

---

**Refinement Prompt**

**System:** You are a helpful assistant.
**User:** Given the following inputs:
Question: `<Question Content>`
Previous Solution: `<Generated Solution>`
Critique: `<Critique Content>`
Please re-answer by:
- Correcting potential errors identified in the critique, if they exist.
- Providing clear, step-by-step reasoning.
- Placing your final answer within `\boxed`.
Ensure the revised solution addresses all issues raised in the critique.

---

Future work could explore designing prompts (Zhang et al., 2023) to enable LLMs to generate high-quality CoT critiques.

**Prompt for Qualitative Analysis.** We employ the following prompt to conduct qualitative analysis of the generated responses using GPT-4o.

---

**Prompt for Qualitative Analysis**

**System:** You are a science expert. You are provided with a question, the correct ground truth answer, and a student's solution. Please conduct a fine-grained qualitative analysis of the student's solution based on the following four aspects, rated on a scale of 1-5:
1. Correctness of the Final Answer: Is the final numerical answer correct based on your calculations or the ground truth? If incorrect, provide the correct answer and explain the discrepancy.
2. Verbosity: Is the reasoning path too verbose, too concise, or appropriate? Identify areas for condensation or expansion to improve clarity.
3. Factual Accuracy: Are all formulas, conversions, and physical principles factually accurate? Highlight any errors or misleading statements.
4. Logical Coherence: Does the reasoning flow logically from one step to the next? Identify gaps in logic, missing steps, or irrelevant details that detract from the solution.
End your analysis with:
"Conclusion:" Provide ratings (1-5 scale) for each aspect.
**User:**
Question: `<Question Content>`
Ground Truth Answer: `<Ground Truth>`
Student's Answer: `<Answer>`
Conclusion:

---

## N. An Example of Successful Refinement using a CoT Critique

The example below demonstrates a successful refinement using a CoT critique. This output was generated by the best-performing RL-finetuned Qwen3-8B model using GRPO algorithm in Section 3, utilizing the refinement prompt detailed in Appendix N.

---

**Question**

2. A sphere is inscribed in a cone, such that the area of the sphere and the area of the base of the cone are equal. Calculate the cosine of the angle of the axial section of the cone at the vertex which is also the apex of the cone.

---

**CoT Critique**

Let's analyze the student's solution step-by-step and identify any errors:
### Step 1: Geometry Understanding The student correctly identifies the geometry of the problem, noting that the sphere is tangent to the base and sides of the cone, and that the axial section forms an isosceles triangle.
### Step 2: Variable Definitions The student defines $r$ as the radius of the sphere, $R$ as the radius of the base of the cone, and $h$ as the height of the cone. These definitions are appropriate.
### Step 3: Relating Sphere and Cone The student attempts to relate $r$, $R$, and $h$ using similar triangles. However, the equation derived:

$$r = \frac{Rh}{h + R}$$

is not directly applicable to the problem as it does not consider the specific condition that the areas are equal.
### Step 4: Equating Areas The student correctly equates the areas of the sphere and the base of the cone:

$$4\pi r^2 = \pi R^2 \implies R = 2r$$

This step is correct.
### Step 5: Substitution and Error The student substitutes $R = 2r$ into the earlier equation, leading to:

$$r = \frac{(2r)h}{h + 2r}$$

This substitution is unnecessary and results in incorrect simplification. The correct approach would involve using the inradius formula for the triangle formed by the axial section of the cone.
### Step 6: Angle Calculation The student calculates $\tan \theta = 1$ and concludes $\theta = 45°$, leading to $\cos(90°) = 0$. This is incorrect because the angle calculation should be based on the relationship between $h$ and $R$, derived from the inradius formula.
### Step 7: Precise Calculation The student attempts to verify the angle calculation but repeats the same incorrect logic, leading to the same incorrect conclusion.
### Correct Solution Approach The correct approach involves using the inradius formula for the triangle formed by the axial section of the cone: - Calculate the inradius using the area and semi-perimeter of the triangle. - Substitute $R = 2r$ and solve for $h$ in terms of $r$. - Relate $h$ and $\theta$ using trigonometric identities. - Calculate $\cos(2\theta)$ using the double angle identity.
The correct final answer is:

$$\cos(2\theta) = \frac{7}{25}.$$

Conclusion: incorrect [END]

---

**Incorrect Initial Response**

To solve the problem, let's break it down step by step.

### Step 1: Geometry Understanding - **Sphere Inscribed in a Cone**: The sphere is tangent to the base and the sides of the cone. - **Axial Section**: The cross-section along the cone's central axis forms an isosceles triangle. - **Vertex Angle**: Let the vertex angle of the cone be $2\theta$, so each base angle is $\theta$.

### Step 2: Variable Definitions - $r$: Radius of the sphere. - $R$: Radius of the base of the cone. - $h$: Height of the cone.

### Step 3: Relating the Sphere and the Cone The sphere is inscribed in the cone, so: - The vertical distance from the center of the sphere to the base is $h - r$. - The horizontal distance from the center of the sphere to the cone's side is $r$. Using similar triangles, we relate $r$, $R$, and $h$:

$$\frac{r}{R} = \frac{h - r}{h} \implies rh = R(h - r) \implies r(h + R) = Rh \implies r = \frac{Rh}{h + R}.$$

### Step 4: Equating Areas The area of the sphere is $4\pi r^2$, and the area of the base of the cone is $\pi R^2$. From the problem:

$$4\pi r^2 = \pi R^2 \implies 4r^2 = R^2 \implies R = 2r.$$

### Step 5: Substituting $R = 2r$ Substituting $R = 2r$ into the earlier equation:

$$r = \frac{(2r)h}{h + 2r} \implies r(h + 2r) = 2rh \implies h + 2r = 2h \implies h = 2r.$$

### Step 6: Finding the Angle $\theta$ In the axial section of the cone, we have a right triangle with: - Height $h = 2r$. - Base radius $R = 2r$.

The tangent of $\theta$ is:

$$\tan\theta = \frac{R}{h} = \frac{2r}{2r} = 1 \implies \theta = 45°.$$

The cosine of the vertex angle $2\theta$ is:

$$\cos(2\theta) = \cos(90°) = 0.$$

### Step 7: Verifying with Precise Calculations For the axial section, the cone forms an isosceles triangle with: - Two equal slant heights $l$. - Base $2R = 4r$. - Height $h = 2r$.

Using the Pythagorean theorem, the slant height $l$ is:

$$l = \sqrt{h^2 + R^2} = \sqrt{(2r)^2 + (2r)^2} = \sqrt{8r^2} = 2r\sqrt{2}.$$

The tangent of the base angle $\theta$ is:

$$\tan\theta = \frac{R}{h} = \frac{2r}{2r} = 1 \implies \theta = 45°.$$

Thus, the vertex angle is $2\theta = 90°$, and:

$$\cos(90°) = 0.$$

### Final Answer

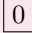 $\boxed{0}$

**Correct Refined Response (1/2)**

Let's revise the solution by addressing the issues identified in the critique and providing a clear, step-by-step reasoning.

### Step 1: Geometry Understanding The problem involves a cone with a sphere inscribed in it, such that the sphere is tangent to both the base and the sides of the cone. Additionally, the area of the sphere equals the area of the base of the cone.

### Step 2: Variable Definitions - $r$: Radius of the sphere. - $R$: Radius of the base of the cone. - $h$: Height of the cone.

### Step 3: Relating the Sphere and the Cone The sphere is inscribed in the cone, so: - The vertical distance from the center of the sphere to the base is $h - r$. - The horizontal distance from the center of the sphere to the cone's side is $r$. Using similar triangles, we relate $r$, $R$, and $h$:

$$\frac{r}{R} = \frac{h-r}{h} \implies r(h+R) = Rh \implies r = \frac{Rh}{h+R}.$$

### Step 4: Equating Areas The area of the sphere is $4\pi r^2$, and the area of the base of the cone is $\pi R^2$. From the problem:

$$4\pi r^2 = \pi R^2 \implies 4r^2 = R^2 \implies R = 2r.$$

### Step 5: Substituting $R = 2r$ Substitute $R = 2r$ into the earlier equation:

$$r = \frac{(2r)h}{h+2r} \implies r(h+2r) = 2rh \implies h + 2r = 2h \implies h = 2r.$$

### Step 6: Finding the Angle $\theta$ In the axial section of the cone: - The height is $h = 2r$. - The base radius is $R = 2r$. The tangent of $\theta$ is:

$$\tan\theta = \frac{R}{h} = \frac{2r}{2r} = 1 \implies \theta = 45°.$$

The cosine of the vertex angle $2\theta$ is:

$$\cos(2\theta) = \cos(90°) = 0.$$

However, this leads to an incorrect conclusion. Let's apply the **inradius formula** for a more accurate approach.

### Correct Approach: Using the Inradius Formula The axial section of the cone is an isosceles triangle with: - Height $h$. - Base $2R$. - Slant height $l = \sqrt{h^2 + R^2}$.

The inradius $r$ of this triangle is given by:

$$r = \frac{A}{s},$$

where $A$ is the area, and $s$ is the semi-perimeter.

### Area and Semi-Perimeter: The area $A$ of the triangle is:

$$A = \frac{1}{2} \times 2R \times h = Rh.$$

**Correct Refined Response (2/2)**

The semi-perimeter $s$ is:

$$s = \frac{2l + 2R}{2} = l + R = \sqrt{h^2 + R^2} + R.$$

### Inradius: Substitute $A$ and $s$:

$$r = \frac{Rh}{\sqrt{h^2 + R^2} + R}.$$

Given $R = 2r$ and $h = 2r$, substitute into the equation:

$$r = \frac{(2r)(2r)}{\sqrt{(2r)^2 + (2r)^2} + 2r} = \frac{4r^2}{\sqrt{8r^2} + 2r} = \frac{4r^2}{2r\sqrt{2} + 2r}.$$

Simplify:

$$r = \frac{4r}{2(\sqrt{2} + 1)} = \frac{2r}{\sqrt{2} + 1}.$$

### Rationalizing the Denominator:

$$r = \frac{2r}{\sqrt{2} + 1} \cdot \frac{\sqrt{2} - 1}{\sqrt{2} - 1} = \frac{2r(\sqrt{2} - 1)}{2 - 1} = 2r(\sqrt{2} - 1).$$

### Step 7: Find the Final Angle Given $h = 2r$ and $R = 2r$, the slant height $l$ is:

$$l = \sqrt{h^2 + R^2} = \sqrt{(2r)^2 + (2r)^2} = \sqrt{8r^2} = 2r\sqrt{2}.$$

The tangent of $\theta$ is:

$$\tan\theta = \frac{R}{h} = \frac{2r}{2r} = 1 \implies \theta = 45°.$$

The cosine of the vertex angle $2\theta$ is:

$$\cos(2\theta) = \frac{1 - \tan^2\theta}{1 + \tan^2\theta} = \frac{1 - \left(\frac{3}{4}\right)^2}{1 + \left(\frac{3}{4}\right)^2} = \frac{\frac{7}{16}}{\frac{25}{16}} = \frac{7}{25}.$$

### Final Answer:

$$\boxed{\frac{7}{25}}$$

## O. Responses Utilized in Qualitative Analysis

---

**Question**
Evaluate
$$\sin(\arcsin 0.4 + \arcsin 0.5) \cdot \sin(\arcsin 0.5 - \arcsin 0.4).$$

---

**Response Generated by Qwen3-8B (1/4)**

Okay, so I need to evaluate this expression: $\sin(\arcsin 0.4 + \arcsin 0.5)$ multiplied by $\sin(\arcsin 0.5 - \arcsin 0.4)$. Hmm, that looks a bit complicated with all those inverse sine functions. Let me try to break it down step by step. First, let me recall some trigonometric identities that might help here. I remember that there are formulas for $\sin(A + B)$ and $\sin(A - B)$. Maybe I can use those? Let me write them down:

$$\sin(A + B) = \sin A \cos B + \cos A \sin B$$

$$\sin(A - B) = \sin A \cos B - \cos A \sin B$$

So if I let $A = \arcsin 0.4$ and $B = \arcsin 0.5$, then the expression becomes:

$$\sin(A + B) \cdot \sin(A - B)$$

Wait, so maybe I can compute $\sin(A + B)$ and $\sin(A - B)$ separately using those identities and then multiply them together? But before that, maybe I should figure out what $A$ and $B$ are? Well, $A$ is $\arcsin 0.4$, which means that $\sin A = 0.4$, and similarly $\sin B = 0.5$. But I need $\cos A$ and $\cos B$ to use the identities. Since $\sin^2 \theta + \cos^2 \theta = 1$, I can find $\cos A$ and $\cos B$. Let me compute $\cos A$ first. For angle $A$, $\sin A = 0.4$, so

$$\cos A = \sqrt{1 - \sin^2 A} = \sqrt{1 - 0.16} = \sqrt{0.84}.$$

Wait, $\sqrt{0.84}$... Let me calculate that. $0.84$ is $\frac{84}{100}$, so $\sqrt{0.84} = \frac{\sqrt{84}}{10}$. Simplify $\sqrt{84}$... $\sqrt{84} = \sqrt{4 \cdot 21} = 2 \cdot \sqrt{21}$. So

$$\sqrt{0.84} = \frac{\sqrt{84}}{10} = \frac{2 \cdot \sqrt{21}}{10} = \frac{\sqrt{21}}{5}.$$

Wait, let me check that again. $\sqrt{84}$ is $\sqrt{4 \cdot 21}$ which is $2 \cdot \sqrt{21}$, so yes, $\sqrt{84}/10 = 2 \cdot \sqrt{21}/10 = \sqrt{21}/5$. So $\cos A = \sqrt{21}/5$. Wait, but $\sqrt{21}$ is approximately $4.583$, so $\sqrt{21}/5$ is about $0.9166$. Let me check if that's correct. Since $\sin A$ is $0.4$, then $\cos A$ should be $\sqrt{1 - 0.16} = \sqrt{0.84} \approx 0.9165$, which matches. So that's correct. Similarly, for angle $B$, $\sin B = 0.5$, so $\cos B = \sqrt{1 - 0.25} = \sqrt{0.75} = \frac{\sqrt{3}}{2} \approx 0.8660$. That's right because $\sin\left(\frac{\pi}{6}\right) = 0.5$, so $\cos\left(\frac{\pi}{6}\right) = \frac{\sqrt{3}}{2}$. So $\cos B = \frac{\sqrt{3}}{2}$. Okay, so now I can compute $\sin(A + B)$ and $\sin(A - B)$ using the identities. Starting with $\sin(A + B)$:

$$\sin(A + B) = \sin A \cos B + \cos A \sin B$$

Plugging in the values:

$$\sin A = 0.4, \quad \cos B = \frac{\sqrt{3}}{2}, \quad \cos A = \frac{\sqrt{21}}{5}, \quad \sin B = 0.5$$

So:

$$\sin(A + B) = 0.4 \left( \frac{\sqrt{3}}{2} \right) + \left( \frac{\sqrt{21}}{5} \right) (0.5)$$

**Response Generated by Qwen3-8B (2/4)**

Let me compute each term: First term:

$$0.4 \left( \frac{\sqrt{3}}{2} \right) = \left( \frac{0.4}{2} \right) \sqrt{3} = 0.2\sqrt{3}$$

Second term:

$$\left( \frac{\sqrt{21}}{5} \right) (0.5) = \left( \frac{\sqrt{21}}{5} \right) \left( \frac{1}{2} \right) = \frac{\sqrt{21}}{10}$$

So

$$\sin(A+B) = 0.2\sqrt{3} + \frac{\sqrt{21}}{10}$$

Hmm, maybe I can write $0.2$ as $\frac{1}{5}$? Let me check: $0.2$ is $\frac{1}{5}$, so

$$\frac{1}{5}\sqrt{3} + \frac{\sqrt{21}}{10}.$$

To combine these terms, maybe get a common denominator? Yes, common denominator is 10:

$$\frac{2\sqrt{3}}{10} + \frac{\sqrt{21}}{10} = \frac{2\sqrt{3} + \sqrt{21}}{10}$$

So

$$\sin(A+B) = \frac{2\sqrt{3} + \sqrt{21}}{10}$$

Okay, now moving on to $\sin(A - B)$:

$$\sin(A - B) = \sin A \cos B - \cos A \sin B$$

Which is similar to the previous one, but with a minus sign. So plugging in the same values:

$$\sin A \cos B - \cos A \sin B = 0.4 \left( \frac{\sqrt{3}}{2} \right) - \left( \frac{\sqrt{21}}{5} \right) (0.5)$$

Which is the same as: First term:

$$0.2\sqrt{3} - \frac{\sqrt{21}}{10}$$

Again, converting $0.2$ to $\frac{1}{5}$:

$$\frac{1}{5}\sqrt{3} - \frac{\sqrt{21}}{10} = \frac{2\sqrt{3} - \sqrt{21}}{10}$$

So

$$\sin(A - B) = \frac{2\sqrt{3} - \sqrt{21}}{10}$$

Therefore, the original expression is:

$$\sin(A + B) \cdot \sin(A - B) = \left( \frac{2\sqrt{3} + \sqrt{21}}{10} \right) \cdot \left( \frac{2\sqrt{3} - \sqrt{21}}{10} \right)$$

**Response Generated by Qwen3-8B (3/4)**

Hmm, this looks like a difference of squares. Because $(a+b)(a-b) = a^2 - b^2$. Let me check: Let me denote $a = 2\sqrt{3}$ and $b = \sqrt{21}$. Then the product is $(a+b)(a-b) = a^2 - b^2$. So compute $a^2 - b^2$:

$$a^2 = (2\sqrt{3})^2 = 4 \cdot 3 = 12$$

$$b^2 = (\sqrt{21})^2 = 21$$

Therefore, $a^2 - b^2 = 12 - 21 = -9$

Therefore, the product is $\dfrac{-9}{10 \cdot 10} = \dfrac{-9}{100}$

Wait, so the entire expression simplifies to $\frac{-9}{100}$? But let me check my steps again to make sure I didn't make a mistake. First, $\sin(A+B)$ and $\sin(A-B)$ were calculated as $\frac{2\sqrt{3}+\sqrt{21}}{10}$ and $\frac{2\sqrt{3}-\sqrt{21}}{10}$. Multiplying them gives:

$$\frac{(2\sqrt{3})^2 - (\sqrt{21})^2}{100} = \frac{12 - 21}{100} = \frac{-9}{100}$$

That seems correct. But let me verify with another approach. Maybe using another identity? Alternatively, I remember that $\sin(A+B) \cdot \sin(A-B)$ can be expressed as $\sin^2 A - \sin^2 B$. Wait, is that correct? Wait, let me recall the identity:

$$\sin(A+B) \cdot \sin(A-B) = \sin^2 A - \sin^2 B$$

Let me check that. Yes, because:

$$\sin(A+B) \cdot \sin(A-B) = [\sin A \cos B + \cos A \sin B][\sin A \cos B - \cos A \sin B]$$

$$= (\sin A \cos B)^2 - (\cos A \sin B)^2 = \sin^2 A \cos^2 B - \cos^2 A \sin^2 B$$

But that's not exactly $\sin^2 A - \sin^2 B$. Wait, unless there's another identity. Alternatively, maybe using product-to-sum formulas? Alternatively, let me compute $\sin^2 A - \sin^2 B$. Let me see:

$$\sin^2 A - \sin^2 B = (\sin A - \sin B)(\sin A + \sin B). \text{ Not sure if that helps.}$$

But let me check if $\sin(A+B) \cdot \sin(A-B)$ equals $\sin^2 A - \sin^2 B$. Let me compute $\sin^2 A - \sin^2 B$:

$$\sin^2 A - \sin^2 B = (0.4)^2 - (0.5)^2 = 0.16 - 0.25 = -0.09 = \frac{-9}{100}$$

Which is exactly the same as before! So that's a shortcut. Therefore:

$$\sin(A+B) \cdot \sin(A-B) = \sin^2 A - \sin^2 B = \frac{-9}{100}$$

Therefore, the answer is $\frac{-9}{100}$. So that confirms the previous result. Therefore, the value of the original expression is $\frac{-9}{100}$. But let me check once more with actual numbers to make sure. Let me compute $\arcsin(0.4)$ and $\arcsin(0.5)$ numerically, then compute the sines of their sum and difference, multiply them, and see if it's approximately $-0.09$. First, $\arcsin(0.5)$ is $\pi/6 \approx 0.5236$ radians. $\arcsin(0.4)$... Let me calculate that. Let me recall that $\sin(0.4115 \text{ radians}) \approx 0.4$. Let me check with calculator: Well, $\sin(0.4115) \approx \sin(0.4115) \approx 0.4$.

Let me check with calculator steps: But since I don't have a calculator here, maybe approximate. Alternatively, use known values. For example, $\sin(\pi/6) = 0.5$, $\sin(\pi/12) \approx 0.2588$, so $0.4$ is between $\pi/12$ and $\pi/6$. Let me approximate $\arcsin(0.4)$.

**Response Generated by Qwen3-8B (4/4)**

Let me use Taylor series or something? Maybe not. Alternatively, just take that $\arcsin(0.4) \approx 0.4115$ radians (as I thought before). Let me take that as approximate value. So $A \approx 0.4115$ radians, $B = \pi/6 \approx 0.5236$ radians. Compute $A + B \approx 0.4115 + 0.5236 \approx 0.9351$ radians. Compute $\sin(0.9351)$. Let me recall that $\sin(\pi/3) \approx 0.8660$, $\pi/3 \approx 1.0472$ radians. So $0.9351$ is less than $\pi/3$. Let me compute $\sin(0.9351)$. Let me use calculator-like approximation.

Alternatively, use the value we found earlier: $\frac{2\sqrt{3}+\sqrt{21}}{10}$. Let me compute that numerically. First, $\sqrt{3} \approx 1.732$, $\sqrt{21} \approx 4.5837$. So $2\sqrt{3} \approx 3.464$, $\sqrt{21} \approx 4.5837$. Then $2\sqrt{3} + \sqrt{21} \approx 3.464 + 4.5837 \approx 8.0477$. Divide by 10: $\approx 0.80477$. So $\sin(A + B) \approx 0.8048$. Similarly, $\sin(A - B) = \frac{2\sqrt{3}-\sqrt{21}}{10} \approx \frac{3.464-4.5837}{10} \approx \frac{-1.1197}{10} \approx -0.11197$. Therefore, multiplying them: $0.8048 \cdot (-0.11197) \approx -0.090$. Which is $-0.09$, which is $-\frac{9}{100}$. So that matches. Therefore, the exact value is $-\frac{9}{100}$, and the approximate decimal is $-0.09$. So that seems correct. Therefore, the answer is $-\frac{9}{100}$.

But let me check once more if there's any mistake in the sign. Wait, when I calculated $\sin(A - B)$, I had $\frac{2\sqrt{3}-\sqrt{21}}{10}$. Since $\sqrt{21} \approx 4.583$ and $2\sqrt{3} \approx 3.464$, so $2\sqrt{3} - \sqrt{21}$ is negative, so $\sin(A - B)$ is negative. Then $\sin(A + B)$ is positive, so multiplying them gives negative. Which matches with $-\frac{9}{100}$. So that's correct. Alternatively, if I think about angles: $A$ is $\arcsin(0.4)$, which is in the first quadrant, $B$ is $\arcsin(0.5)$ which is $\pi/6$, also first quadrant. So $A + B$ is the sum of two first quadrant angles. Depending on their measures, $A + B$ could be in the first or second quadrant. Similarly, $A - B$ could be positive or negative.

Let me check if $A - B$ is positive or negative. Since $\arcsin(0.4) \approx 0.4115$ radians, $\arcsin(0.5) \approx 0.5236$ radians. So $A - B \approx 0.4115 - 0.5236 \approx -0.1121$ radians. So a negative angle, which is equivalent to a positive angle by adding $2\pi$, but sine is an odd function, so $\sin(-x) = -\sin x$. Therefore, $\sin(A - B) = -\sin(B - A)$. So the negative value makes sense. Therefore, multiplying $\sin(A + B)$ which is positive with $\sin(A - B)$ which is negative gives a negative result. So $-\frac{9}{100}$ is correct. Therefore, I think the answer is $-\frac{9}{100}$. Let me check once more with another identity.

Alternatively, using complex numbers or other methods? Maybe not necessary. Since two different approaches gave me the same answer, and numerical approximation also matches, I think it's safe. But just to make sure, let me compute $\sin^2 A - \sin^2 B$:

$$\sin^2 A = (0.4)^2 = 0.16$$

**Response Generated by RL-Finetuned Qwen3-8B using Critique-GRPO (1/2)**

Okay, so I need to evaluate this expression: $\sin(\arcsin 0.4 + \arcsin 0.5)$ multiplied by $\sin(\arcsin 0.5 - \arcsin 0.4)$. Hmm, let me think.

First, I remember that `arcsin` is the inverse sine function, so $\arcsin 0.4$ is an angle whose sine is $0.4$, and similarly for $\arcsin 0.5$. Let me denote some variables to make this easier. Let me set $\alpha = \arcsin 0.4$ and $\beta = \arcsin 0.5$. Then the expression becomes $\sin(\alpha + \beta) * \sin(\beta - \alpha)$.

Alright, so I need to compute $\sin(\alpha+\beta)$ and $\sin(\beta-\alpha)$ separately and then multiply them. Maybe I can use sine–addition formulas here. Let me recall that $\sin(A+B) = \sin A \cos B + \cos A \sin B$, and $\sin(A-B) = \sin A \cos B - \cos A \sin B$.

Since $\alpha$ and $\beta$ are `arcsin` values, I know that $\sin \alpha = 0.4$ and $\sin \beta = 0.5$. Therefore, I need to find $\cos \alpha$ and $\cos \beta$ to use in the formulas.

For $\cos \alpha$, since $\sin^2 \alpha + \cos^2 \alpha = 1$, we have

$$\cos \alpha = \sqrt{1 - \sin^2 \alpha} = \sqrt{1 - 0.16} = \sqrt{0.84}.$$

Note that $0.84 = \frac{84}{100} = \frac{21}{25}$, so $\cos \alpha = \sqrt{\frac{21}{25}} = \frac{\sqrt{21}}{5}$.

Similarly,

$$\cos \beta = \sqrt{1 - \sin^2 \beta} = \sqrt{1 - 0.25} = \sqrt{0.75} = \sqrt{\tfrac{3}{4}} = \frac{\sqrt{3}}{2}.$$

Now, let me compute $\sin(\alpha + \beta)$. Using the formula,

$$\sin(\alpha + \beta) = \sin \alpha \cos \beta + \cos \alpha \sin \beta = 0.4\left(\frac{\sqrt{3}}{2}\right) + \left(\frac{\sqrt{21}}{5}\right)(0.5).$$

Compute each term:

$$0.4 \cdot \frac{\sqrt{3}}{2} = \frac{2}{5} \cdot \frac{\sqrt{3}}{2} = \frac{\sqrt{3}}{5}, \quad \frac{\sqrt{21}}{5} \cdot 0.5 = \frac{\sqrt{21}}{10}.$$

Therefore

$$\sin(\alpha + \beta) = \frac{\sqrt{3}}{5} + \frac{\sqrt{21}}{10} = \frac{2\sqrt{3} + \sqrt{21}}{10}.$$

Similarly, compute $\sin(\beta - \alpha)$:

$$\sin(\beta - \alpha) = \sin \beta \cos \alpha - \cos \beta \sin \alpha = 0.5\left(\frac{\sqrt{21}}{5}\right) - \frac{\sqrt{3}}{2}(0.4).$$

Thus

$$\sin(\beta - \alpha) = \frac{\sqrt{21}}{10} - \frac{2\sqrt{3}}{10} = \frac{\sqrt{21} - 2\sqrt{3}}{10}.$$

Therefore, the original expression is

$$\left(\frac{2\sqrt{3} + \sqrt{21}}{10}\right)\left(\frac{\sqrt{21} - 2\sqrt{3}}{10}\right).$$

Let $A = 2\sqrt{3} + \sqrt{21}$ and $B = \sqrt{21} - 2\sqrt{3}$; then the product is $\frac{AB}{100}$. Note that $AB = (\sqrt{21})^2 - (2\sqrt{3})^2 = 21 - 12 = 9$, so the value is $\dfrac{9}{100}$.

**Response Generated by RL-Finetuned Qwen3-8B using Critique-GRPO (2/2)**

Therefore, the value of the original expression is $\dfrac{9}{100}$, which is 0.09.

Let me check with approximate values. Take $\arcsin 0.5 = \frac{\pi}{6} \approx 0.5236\,\text{rad}$ and $\arcsin 0.4 \approx 0.4115\,\text{rad}$. Then $\alpha + \beta \approx 0.9351\,\text{rad}$, so $\sin(\alpha + \beta) \approx 0.805$; also $\beta - \alpha \approx 0.1121\,\text{rad}$, so $\sin(\beta - \alpha) \approx 0.1118$. Their product $0.805 \times 0.1118 \approx 0.09$, confirming $\dfrac{9}{100}$.

**Final Answer**

$$\boxed{\dfrac{9}{100}}$$

To evaluate the expression

$$\sin(\arcsin 0.4 + \arcsin 0.5) \cdot \sin(\arcsin 0.5 - \arcsin 0.4),$$

we begin by introducing a substitution to simplify the notation. Let

$$\alpha = \arcsin 0.4, \qquad \beta = \arcsin 0.5,$$

so that $\sin \alpha = 0.4$ and $\sin \beta = 0.5$.

We rewrite the expression as $\sin(\alpha + \beta)\,\sin(\beta - \alpha)$.

**Step 1: Use the Sine Addition and Subtraction Formulas**
Recall
$$\sin(\alpha + \beta) = \sin \alpha \, \cos \beta + \cos \alpha \, \sin \beta, \qquad \sin(\beta - \alpha) = \sin \beta \, \cos \alpha - \cos \beta \, \sin \alpha.$$

Using
$$\cos \alpha = \frac{\sqrt{21}}{5}, \qquad \cos \beta = \frac{\sqrt{3}}{2},$$

we compute
$$\sin(\alpha + \beta) = \frac{2\sqrt{3} + \sqrt{21}}{10}, \qquad \sin(\beta - \alpha) = \frac{\sqrt{21} - 2\sqrt{3}}{10}.$$

**Step 2: Multiply the Two Sine Values**
$$\left(\frac{2\sqrt{3} + \sqrt{21}}{10}\right)\left(\frac{\sqrt{21} - 2\sqrt{3}}{10}\right) = \frac{(\sqrt{21})^2 - (2\sqrt{3})^2}{100} = \frac{9}{100}.$$

**Final Answer**

$$\boxed{\dfrac{9}{100}}$$

## P. An Example of Failed Refinement

The following example demonstrates a failed refinement using a CoT critique, where RL-finetuned Qwen3-8B using Critique-GRPO fails to follow the critique to refine its responses.

---

**Question**
Example 15: Let $a, b, c$ be positive real numbers, and $abc + a + c = b$, find the maximum value of

$$p = \frac{2}{a^2 + 1} - \frac{2}{b^2 + 1} + \frac{3}{c^2 + 1}.$$

---

**CoT Critique**
The student's solution contains several errors and misunderstandings. Let's go through the mistakes step-by-step:
1. **Misinterpretation of the Constraint:** - The student correctly rearranges the constraint $abc + a + c = b$ to express $b$ in terms of $a$ and $c$, but then incorrectly assumes $b = \tan(\alpha + \gamma)$ without proper justification. The trigonometric substitution should be used more carefully, ensuring that the constraint is satisfied throughout.
2. **Incorrect Use of Trigonometric Identities:** - The student attempts to use trigonometric identities to simplify the expression for $p$, but the application is incorrect. The identities used do not lead to a valid simplification of the expression.
3. **Incorrect Derivative Calculation:** - The student attempts to find the maximum by taking derivatives, but the calculations are incorrect. The derivative should be taken with respect to the correct variables, and the critical points should be found accurately.
4. **Incorrect Evaluation of $p$:** - The student evaluates $p$ at specific points, but the calculations are inconsistent and do not match the constraint. The values chosen for $a$, $b$, and $c$ do not satisfy the constraint $abc + a + c = b$.
5. **Incorrect Final Answer:** - The student concludes with an incorrect maximum value of $p = \frac{105}{32}$, which is not supported by the calculations. The correct maximum value, as derived from a proper analysis, should be $\frac{10}{3}$.
**Conclusion:** wrong [END]

**Incorrect Initial Response (1/2)**

Okay, so I need to find the maximum value of this expression:

$$p = \frac{2}{a^2 + 1} - \frac{2}{b^2 + 1} + \frac{3}{c^2 + 1},$$

given that $a, b, c$ are positive real numbers satisfying $abc + a + c = b$. Hmm, let me start by understanding the constraint equation better.

The equation is:

$$abc + a + c = b.$$

Maybe I can rearrange this to express one variable in terms of the others. Let me try to solve for $b$. Let's see:

$$abc + a + c = b \implies b - abc = a + c \implies b(1 - ac) = a + c \implies b = \frac{a + c}{1 - ac}.$$

Wait, but since $a, b, c$ are positive real numbers, the denominator $1 - ac$ must be positive because the numerator $a + c$ is positive. Therefore:

$$1 - ac > 0 \implies ac < 1.$$

So, the product of $a$ and $c$ must be less than 1. That's an important condition.

—

Now, the expression for $p$ involves $a, b, c$. Since $b$ is expressed in terms of $a$ and $c$, perhaps I can substitute that into $p$ and then try to maximize with respect to $a$ and $c$. Let me try that.

Substituting $b = \frac{a+c}{1-ac}$ into $p$, we get:

$$p = \frac{2}{a^2 + 1} - \frac{2}{\left(\frac{a+c}{1-ac}\right)^2 + 1} + \frac{3}{c^2 + 1}.$$

This looks complicated. Maybe there's a substitution or some symmetry here. Let me check if there's a way to simplify the expression for $b$. Alternatively, maybe using substitution variables.

—

### Trigonometric Substitution

Let's consider substituting variables to simplify the constraint. For example, sometimes in equations with products, substitutions like $x = a, y = c$ are helpful, but I'm not sure in this case. Alternatively, maybe trigonometric substitution? Since we have terms like $a^2 + 1$, which reminds me of $\tan \theta$ substitutions. For example:
- Let $a = \tan \alpha$, $c = \tan \gamma$, where $\alpha, \gamma \in (0, \pi/2)$. - Then:

$$a^2 + 1 = \sec^2 \alpha \implies \frac{1}{a^2 + 1} = \cos^2 \alpha,$$

and similarly for $c$.

Let me try that. Substituting $a = \tan \alpha$ and $c = \tan \gamma$, and using the constraint $abc + a + c = b$, we get:

$$\tan \alpha \cdot b \cdot \tan \gamma + \tan \alpha + \tan \gamma = b.$$

Solving for $b$:

$$b(\tan \alpha \tan \gamma - 1) = \tan \alpha + \tan \gamma \implies b = \frac{\tan \alpha + \tan \gamma}{\tan \alpha \tan \gamma - 1}.$$

Wait, but earlier we had:

$$b = \frac{a + c}{1 - ac}.$$

Since $ac = \tan \alpha \tan \gamma$, so $1 - ac = 1 - \tan \alpha \tan \gamma$. Therefore:

$$b = \frac{\tan \alpha + \tan \gamma}{1 - \tan \alpha \tan \gamma}.$$

—

**Incorrect Initial Response (2/2)**
### Simplifying Further
Notice that:
$$\frac{\tan\alpha + \tan\gamma}{1 - \tan\alpha\tan\gamma} = \tan(\alpha + \gamma).$$

Thus:
$$b = \tan(\alpha + \gamma).$$

Therefore, since $b > 0$, $\tan(\alpha + \gamma) > 0$. Since $\alpha, \gamma \in (0, \pi/2)$, $\alpha + \gamma \in (0, \pi/2)$. This gives: $\alpha + \gamma < \frac{\pi}{2}$.

—
### Substituting Back into $p$
Using the substitution:
$$p = \frac{2}{a^2 + 1} - \frac{2}{b^2 + 1} + \frac{3}{c^2 + 1}.$$

Since $a = \tan\alpha$, we have:
$$\frac{1}{a^2 + 1} = \cos^2\alpha, \quad \frac{1}{c^2 + 1} = \cos^2\gamma, \quad \frac{1}{b^2 + 1} = \cos^2(\alpha + \gamma).$$

Thus: $p = 2\cos^2\alpha - 2\cos^2(\alpha + \gamma) + 3\cos^2\gamma$.

—
### Reducing the Problem
The problem now reduces to maximizing:
$$p = 2\cos^2\alpha - 2\cos^2(\alpha + \gamma) + 3\cos^2\gamma,$$

subject to $\alpha, \gamma > 0$ and $\alpha + \gamma < \pi/2$.
Let $\theta = \alpha + \gamma$. Then: $p = 2\cos^2\alpha - 2\cos^2\theta + 3\cos^2(\theta - \alpha)$.
—
### Symmetry Assumption
Assume $\alpha = \gamma$, which implies $\theta = 2\alpha$. Then:
$$p = 2\cos^2\alpha - 2\cos^2(2\alpha) + 3\cos^2\alpha = 5\cos^2\alpha - 2\cos^2(2\alpha).$$

Using the identity $\cos(2\alpha) = 2\cos^2\alpha - 1$, we get:
$$\cos^2(2\alpha) = (2\cos^2\alpha - 1)^2.$$

Let $x = \cos^2\alpha$, so $x \in \left(\frac{1}{2}, 1\right)$. Substituting:
$$p = 5x - 2(2x - 1)^2 = 5x - 2(4x^2 - 4x + 1) = -8x^2 + 13x - 2.$$

This is a quadratic equation, and its maximum occurs at:
$$x = \frac{-b}{2a} = \frac{13}{16}.$$

Substituting $x = \frac{13}{16}$ back into $p$:
$$p = -8\left(\frac{13}{16}\right)^2 + 13\left(\frac{13}{16}\right) - 2 = \frac{105}{32}.$$

—
### Final Answer
$$\boxed{\frac{105}{32}}$$

**Incorrect Refined Response (1/2)**

The given problem asks us to find the **maximum value** of the expression:

$$p = \frac{2}{a^2 + 1} - \frac{2}{b^2 + 1} + \frac{3}{c^2 + 1}$$

subject to the constraint:

$$abc + a + c = b$$

where $a, b, c$ are **positive real numbers**.

—

### Step 1: Analyze the Constraint

From the constraint:

$$abc + a + c = b$$

we can rearrange to isolate $b$:

$$b(1 - ac) = a + c \quad \Rightarrow \quad b = \frac{a + c}{1 - ac}.$$

Since $a, b, c > 0$, the denominator $1 - ac$ must be positive, so:

$$ac < 1$$

This condition will be important in limiting the domain of possible values for $a$ and $c$.

—

### Step 2: Trigonometric Substitution

To simplify the expression for $p$, we use the following substitutions:

- Let $a = \tan\alpha$ - Let $c = \tan\gamma$

Then, since $a^2 + 1 = \sec^2\alpha$, we have:

$$\frac{1}{a^2 + 1} = \cos^2\alpha, \quad \frac{1}{c^2 + 1} = \cos^2\gamma.$$

Also, from the tangent addition formula:

$$b = \frac{a + c}{1 - ac} = \tan(\alpha + \gamma).$$

Therefore:

$$\frac{1}{b^2 + 1} = \cos^2(\alpha + \gamma).$$

Substituting into $p$, we get:

$$p = 2\cos^2\alpha - 2\cos^2(\alpha + \gamma) + 3\cos^2\gamma.$$

Let $\theta = \alpha + \gamma$, and express $\gamma = \theta - \alpha$. Then:

$$p = 2\cos^2\alpha - 2\cos^2\theta + 3\cos^2(\theta - \alpha).$$

—

**Incorrect Refined Response (2/2)**
### Step 3: Assume Symmetry for Simplification
To simplify further, assume $\alpha = \gamma$, which implies $\theta = 2\alpha$. Then:

$$p = 2\cos^2 \alpha - 2\cos^2(2\alpha) + 3\cos^2 \alpha = 5\cos^2 \alpha - 2\cos^2(2\alpha).$$

Using the double-angle identity $\cos(2\alpha) = 2\cos^2 \alpha - 1$, we get:

$$\cos^2(2\alpha) = (2\cos^2 \alpha - 1)^2.$$

Let $x = \cos^2 \alpha$, so $x \in \left(\frac{1}{2}, 1\right)$. Then:

$$p = 5x - 2(2x - 1)^2 = -8x^2 + 13x - 2.$$

This is a quadratic in $x$, opening downward, so its maximum occurs at:

$$x = \frac{-b}{2a} = \frac{13}{16}.$$

Substituting back:

$$p = -8\left(\frac{13}{16}\right)^2 + 13\left(\frac{13}{16}\right) - 2 = \frac{105}{32}.$$

—

### Final Answer

$$\boxed{\frac{105}{32}}$$

