# OpenReview forum: "Advancing LLM Reasoning with Natural Language and Numerical Feedback"
_ICML.cc/2026/Conference — ICML 2026 spotlight_

### Official Review · Reviewer_5JuT · 2026-03-11

**Soundness:** 3
**Presentation:** 3
**Significance:** 3
**Originality:** 3
**Overall Recommendation:** 4
**Confidence:** 5

**Summary:**

- This paper identifies three critical issues in purely numerical feedback-based rlvr: performance plateaus, ineffective spontaneous self-reflection, and persistent failures in challenging problems.
- propose Critique-GRPO, an online RL framework that integrates both natural language and numerical feedback for policy optimization —— use natural language feedback to guide correct solutions and add to group
- Experimental results demonstrate substantial improvements over baseline methods across multiple model families, types and scales.

**Compliance With Llm Reviewing Policy:**

Affirmed.

**Final Justification:**

I appreciate the authors' rebuttal. I will keep my positive score.

**Key Questions For Authors:**

- In line 92, this paper refer "To bridge this gap, recent work has combined online RL with expert demonstrations (Yan et al., 2025; Lv et al., 2025; Lanchantin et al., 2025);". However, to my knowledge, the paper (Lv et al., 2025)  haven't use  expert demonstrations.
- Weak-to-strong supervision capability: In the current experimental setup, critiques are generated either by stronger models or through self-critique. An important question remains: Can effective improvement be achieved when critiques are provided by weaker models? i.e. weak to strong supervision.

**Limitations:**

yes

**Strengths And Weaknesses:**

**Strengths**
- Well-motivated problem identification and solution: The authors systematically observe, analyze, and quantify three important phenomena: (i) performance plateaus, (ii) ineffective spontaneous self-reflection, and (iii) persistent failures. The proposed Critique-GRPO framework directly and appropriately addresses these identified challenges.
- Comprehensive experimental validation: The experiments span diverse model configurations, including different scales (3B, 7B, 8B, 32B), different families (LLaMA, Qwen), and different types (reasoning or not). The extensive results convincingly demonstrate its effectiveness.
- High-quality presentation with thorough analysis: The paper is well-written with comprehensive content, including valuable practical insights such as training cost comparisons, which enhance the work's applicability.

**Weaknesses**
- Limited comparison with expert-response baselines  (If this is properly explained, I will increase my score)
    - Both using stronger models for critique generation and using expert-generated responses rely on more capable models, without fundamental differences.
    - Self-critique experiments are only conducted on a subset of models.
    - To substantiate the claimed superiority, the authors should either
         - extend self-critique experiments to all evaluated models
         - or provide comprehensive comparisons with methods that utilize expert-generated responses, such as Luffy (https://arxiv.org/abs/2504.14945 which also employs policy shaping) and QuestA(https://arxiv.org/abs/2507.13266)
- Insufficient evaluation on challenging benchmarks: Due to the limited number of problems in challenging datasets (e.g., AIME 2024 contains only 30 problems), the evaluation would benefit from including AIME 2025 to strengthen its soundness.
- Incomplete related work discussion: Line 780 mentions "some important related works" but lacks detailed elaboration. This section should be expanded to provide proper context and comparison with existing literature.

---

> ### Author Rebuttal · Authors · 2026-03-30
>
> Thank you for your feedback. We address your concerns and questions below.
>
> ---
>
> ### Responses to Weakness 1:
>
> We appreciate this question and take the opportunity to clarify a fundamental distinction. Methods such as LUFFY (Yan et al., 2025) and QuestA (Li et al., 2025) incorporate expert-generated solutions—complete reasoning traces—that the policy can directly imitate. In contrast, Critique-GRPO provides the model with natural language critiques (e.g., error localization and corrective suggestions), and the model must still generate a correct response autonomously by searching over the token space conditioned on these critiques. As shown in our theoretical analysis (Appendix F.4) and supported by prior work [1], the model's self-refinement capability is the operative mechanism, which is fundamentally different from trajectory-level imitation. Moreover, critiques of varying granularity yield different performance gains (Section 5.5), further confirming that it is the informative content of the feedback—not access to expert solutions—that drives improvement.
>
> To directly address the reviewer's suggestion, we provide additional self-critique results on Qwen-2.5-7B-Base below. Critique-GRPO (Self-Critique) significantly outperforms R1-GRPO (p < 0.01), confirming the superiority of joint numerical-and-language feedback without any reliance on stronger external models.
>
> | Method | MATH-500 | Minerva | Olympiad | AMC23 | AIME24 | AIME25 | TheoremQA | GPQA-D | MMLU-Pro |
> |---|---|---|---|---|---|---|---|---|---|
> | Qwen-2.5-7B-Base | 60.80 | 20.20 | 30.40 | 35.00 | 13.30 | 6.70 | 21.60 | 28.79 | 46.24 |
> | R1-GRPO | 74.00 | 32.00 | 38.50 | 42.50 | 16.70 | 10.00 | 40.60 | 33.33 | 51.81 |
> | Critique-GRPO (Self-Critique) | 77.00 | 36.00 | 39.70 | 60.00 | 20.00 | 13.30 | 44.00 | 36.87 | 54.88 |
> | Critique-GRPO (CoT-Critique) | 77.80 | 36.80 | 42.40 | 62.50 | 20.00 | 10.00 | 44.00 | 37.88 | 55.28 |
>
> ### Response to Weakness 2:
> We appreciate the suggestion to include AIME 2025. We have already included results for AIME 2025 in Section 5.4 and Appendix H of the current draft. Furthermore, we have added AIME 2025 to the new evaluation tables provided in this rebuttal (see above and below), demonstrating consistent improvements on this challenging benchmark.
>
> ### Response to Weakness 3
> Thank you for pointing this out. The current draft includes representative foundational works, but we agree that it would benefit from expansion. In the revised manuscript, we will provide a more comprehensive discussion of the literature.
>
> ---
>
> ### Responses to Questions
>
> 1. Response to Q1:
> Thank you for the careful reading. Upon rigorous checking, the training objective of the HPT algorithm in Lv et al. (2025) *does* include a Supervised Fine-Tuning (SFT) loss on the supervising trajectory (i.e., expert demonstrations in the training dataset); please refer to Algorithm 1 on Page 9 of their paper [2]. Nonetheless, to avoid any ambiguity, we will remove or rephrase this citation in the revised version.
>
> 2. Response to Q2:
> This is an excellent question. To test whether effective improvement can be achieved using critiques from a weaker model, we conducted a weak-to-strong supervision experiment. We used Qwen3-8B (a strong reasoning-based policy model) guided by critiques from Qwen3-8B-Base (a weaker model). As shown in the table below, the policy model successfully learns from the weaker model's critiques, yielding significant improvements over the R1-GRPO baseline (p < 0.01).
>
> | Method | MATH-500 | Minerva | Olympiad | AMC23 | AIME24 | AIME25 | TheoremQA | GPQA-D | MMLU-Pro |
> |---|---|---|---|---|---|---|---|---|---|
> | Qwen3-8B (w/ Think) | 82.00 | 41.20 | 44.10 | 67.50 | 40.00 | 23.30 | 46.90 | 35.86 | 68.25 |
> | R1-GRPO | 91.00 | 52.60 | 65.60 | 82.50 | 50.00 | 43.30 | 57.90 | 40.40 | 70.00 |
> | Critique-GRPO (Weaker Critic) | 92.00 | 52.90 | 66.20 | 87.50 | 63.30 | 46.70 | 58.90 | 47.47 | 70.86 |
> | Critique-GRPO (CoT-Critique) | 92.00 | 52.90 | 66.80 | 92.50 | 63.30 | 46.70 | 60.10 | 47.98 | 70.47 |
>
> ---
>
> We sincerely appreciate your efforts and hope these responses address your concerns.
>
> **References**
>
> [1] Provably Learning from Language Feedback
>
> [2] Towards a Unified View of Large Language Model Post-Training

---

> > ### Author Rebuttal · Reviewer_5JuT · 2026-04-03
> >
> > Thank you for the rebuttal. The authors have largely addressed my concerns, and I will maintain my positive score.

---

> > > ### Author Response · Authors · 2026-04-03
> > >
> > > Thank you once again for your valuable comments! Should there be any remaining questions requiring further clarification, please do not hesitate to let us know. If you are satisfied with our responses, we would greatly appreciate your consideration in adjusting the evaluation scores accordingly. We sincerely look forward to your feedback.

---

### Official Review · Reviewer_LKDM · 2026-03-13

**Soundness:** 2
**Presentation:** 3
**Significance:** 3
**Originality:** 3
**Overall Recommendation:** 4
**Confidence:** 3

**Summary:**

This paper studies online RL for reasoning with both numerical rewards and natural-language critiques. The main mrthods, Critique-GRPO, augments GRPO-style training with critique-guided refinement when all initial sampled attempts fail, and introduces an additional shaping term to place more weight on refined trajectories that turn failures into correct solutions. The paper is motivated by a real limitation of scalar-reward RL for reasoning: when all sampled rollouts are wrong, the learning signal is often too weak to drive meaningful correction. Empirically, the method reports substantial gains over reward-only GRPO baselines across mathematical and science/general-reasoning benchmarks.

**Compliance With Llm Reviewing Policy:**

Affirmed.

**Key Questions For Authors:**

- How much of the observed gain remains when critique quality is substantially weakened or made noisy?
- Can the authors better separate the contribution of the policy-update mechanism from the contribution of the external critique source itself?
- Is the “refine only when all initial attempts fail” rule close to optimal, or is it mainly a practical heuristic?
- Can the authors clarify more explicitly how their approach differs from language-feedback methods that directly shape or update reward functions?

Minor:

Some relate work on natural-language feedback and reward construction is missing, e.g., arXiv:2503.02077.

**Limitations:**

yes

**Strengths And Weaknesses:**

## Strengths
- how to make online reasoning RL benefit from richer feedback than a final scalar/verifiable reward.
- The method is conceptually clean. The training loop—sample, critique failures, refine, then optimize on the augmented candidate set—is easy to follow and reasonably well motivated.
- The empirical section appears good overall, especially the analyses suggesting that critiques improve exploration and repair rather than merely serving as post-hoc explanation.

## Weaknesses
1. **A substantial fraction of the gain may come from critic quality rather than the update rule itself.** In particular, the method appears to rely on strong critique generation, so the comparison to reward-only RL may be somewhat asymmetric in terms of injected task information.
2. Once ground-truth answers, verifiers, and high-quality critiques are available, part of the difficulty may shift from policy learning to constructing an effective feedback pipeline. The paper should be careful in how broadly it frames the contribution.
3. Refinement is invoked only when all initial attempts fail. This is computationally efficient, but it remains unclear whether partially-correct trajectories could also benefit from critique-guided repair.

---

> ### Author Rebuttal · Authors · 2026-03-29
>
> Thank you for your feedback. We address your concerns below.
>
> ---
>
> ### Response to Weakness 1:
>
> We appreciate this question and take the opportunity to clarify that our performance gains stem from the policy update mechanism rather than merely the injection of external task information, we provide both theoretical analysis (Section 4, Appendix F.4) and empirical evidence. Table 2 demonstrates that Critique-GRPO using simple, rule-based indicative critiques still yields statistically significant improvements (p < 0.01) over the GRPO baseline. Furthermore, our self-critique variant (Section 5.4) proves efficacy relies on the dual feedback mechanism itself, not an oracle critic.
>
> To further isolate the contribution of our online joint optimization, we compared Critique-GRPO against a sequential baseline: *R1-GRPO + Refinement-SFT (CoT Critique)*. In the table below, Critique-GRPO significantly outperforms this baseline (p < 0.01) across both in-distribution and out-of-distribution benchmarks, confirming the superiority of Critique-GRPO.
>
> | Method | Math (ID) | | | | | Science & General (OOD) | | | Avg |
> | :--- | :---: | :---: | :---: | :---: | :---: | :---: | :---: | :---: | :---: |
> | | MATH-500 | Minerva | Olympiad | AMC23 | AIME24 | TheoremQA | GPQA-Diamond | MMLU-Pro | |
> | Qwen-2.5-7B-Base | 60.80 | 20.20 | 30.40 | 35.00 | 13.30 | 21.60 | 28.79 | 46.24 | 32.04 |
> | R1-GRPO | 74.00 | 32.00 | 38.50 | 42.50 | 16.70 | 40.60 | 33.33 | 51.81 | 41.18 |
> | R1-GRPO + Ref.-SFT | 75.40 | 33.80 | 40.10 | 47.50 | 20.00 | 41.20 | 34.34 | 52.83 | 43.15 |
> | **Critique-GRPO** | **77.80** | **36.80** | **42.40** | **62.50** | **20.00** | **44.00** | **37.88** | **55.28** | **47.08** |
>
> ### Responses to Weakness 2:
> We agree and will carefully frame our contribution in the revision. As detailed in Section 4, our core contribution is an online RL framework that seamlessly integrates targeted natural language feedback while preserving standard exploration. By selectively incorporating refinements, we prevent entropy explosion and distribution shifts. Table 2 (Section 5.2) shows Critique-GRPO outperforms existing feedback pipelines (e.g., RAFT, Refinement-FT), highlighting the specific value of our online RL integration.
>
> ### Response to Weakness 3:
> Thanks for raising the question. When a response set contains both correct and incorrect trajectories, standard GRPO already effectively reinforces the correct ones. In our challenging math tasks (detailed in Section 3, Section 5, Appendix G), "partially correct" responses (an incorrect reasoning chain that coincidentally yields the correct final answer) are exceedingly rare; manual checking revealed virtually none. While our theoretical analysis (Appendix F.4) indicates such trajectories could benefit from critiques, refining only complete failures remains the most practical and efficient approach for these domains.
>
> ---
>
> ### Responses to Questions:
> Thank you for these questions. We respond as follows:
> 1. Response to Q1: Sections 5.4–5.5 evaluate four critique models of varying capability; self-critique matches GPT-4o performance (68.13 vs. 68.26; Table 4). Furthermore, Appendix C reports experiments with noisy and adversarial critiques under oracle numerical feedback, where we observe entropy explosion and performance degradation. This confirms that gains require valid constructive critiques rather than any arbitrary language signal.
>
> 2. Response to Q2: The strong performance of our self-critique variant (Table 4, Section 5.4) directly addresses this. By using the policy model itself to generate critiques (achieving an average score of 68.13, nearly matching the GPT-4o critique score of 68.26), we show that the performance gains stem fundamentally from the mechanism of learning from both numerical and natural language feedback, rather than relying on an external oracle.
>
> 3. Response to Q3: This is a practical heuristic balancing efficiency and performance. When correct solutions already exist in the response set, GRPO can reinforce them directly; refinement is most valuable when none exist. Our theoretical analysis (Appendix F.4) suggests extending refinement to partially correct trajectories could yield further gains, which we leave to future work.
>
> 4. Response to Q4: Existing language-feedback methods typically transform language critiques into scalar numerical rewards to guide policy optimization. This compression discards the rich, constructive information embedded within the text. In contrast, Critique-GRPO enables the model to learn *directly* from the natural language feedback (leveraging the textual information) while simultaneously utilizing the corresponding numerical feedback for effective policy optimization (see Appendix A and F.4 for details).
>
> 5. Thank you for pointing out the missing related work; we will ensure it is properly cited and discussed in the revised manuscript.
>
> ---
>
> We sincerely appreciate your efforts and hope these responses address your concerns.

---

> > ### Author Rebuttal · Reviewer_LKDM · 2026-04-03
> >
> > Thank you for the rebuttal. I will maintain my score.

---

### Official Review · Reviewer_5cbv · 2026-03-13

**Soundness:** 4
**Presentation:** 3
**Significance:** 4
**Originality:** 3
**Overall Recommendation:** 5
**Confidence:** 4

**Summary:**

This paper proposes Critique-GRPO, an extension of GRPO to learning from natural language feedback in the form of textual critiques. The method works by sampling initial responses, generating critiques, producing refined answers via in-context learning, and jointly optimizing over both initial and refined trajectories using a modified GRPO objective augmented with a policy shaping function.

**Compliance With Llm Reviewing Policy:**

Affirmed.

**Final Justification:**

The paper's core strengths remain compelling. The systematic diagnosis of GRPO's limitations in Section 3 is thorough and well-motivated, providing a strong foundation for the proposed methodand the integration of refined trajectories into the RL objective via a policy shaping function is a novel design choice that goes beyond the SFT-style approaches used in prior work.

My two main concerns was the fairness of the compute comparison with GRPO. The authors addressed both convincingly. On the first point, the new comparison against R1-GRPO + Refinement-SFT provides a much more controlled baseline that accounts for the additional critique and refinement steps.

Overall, I consider this a solid paper with a well-motivated method, strong empirical results, and sufficient ablations. I recommend acceptance.

**Key Questions For Authors:**

- The method only does refinement when the entire initial response set has zero correct solutions. Why not apply it even if there are some successes? As the authors noted, textual critiques can convey more information than scalars.

- How robust is the method to noisy or misleading critiques? This can be quite common in hard tasks or with weak critique models

- Table 8 shows that increasing the critique token length slightly hurts performance. Do the authors have an explanation for that?

- The theoretical section assumes critiques can localize the first error or provide corrective guidance. Can you be more explicit about which parts are only relevant under idealized assumptions versus which parts are relevant to the actual noisy critique used in experiments?

**Limitations:**

yes

**Strengths And Weaknesses:**

**Strengths**
- The paper's strongest contribution may be its careful analysis of regular GRPO. Rather than simply proposing a new method, Section 3 systematically diagnoses problems with the current training methods before proposing a solution.
- The way the authors chose to integrate the refined trajectories into the objective function is novel, with previous work focusing only on SFT-style objectives.
- Experiments are done across eight reasoning benchmarks on multiple models, demonstrating consistent and substantial improvements.

**Weaknesses**
- The comparison to baselines is not completely fair. Critique-GRPO performs extra critique generation, refinement, and so on, that the GRPO baseline doesn’t do. It would be useful to have a comparison where the algorithms are matched in wall-clock time or compute, not just training prompts.
- The largest performance gains come from CoT critiques, which in most experiments are generated by strong models. While the authors experimented with various critique models, the self-critique variant (the only one that requires no external oracle) underperforms substantially.

---

> ### Author Rebuttal · Authors · 2026-03-29
>
> Thank you for your feedback. We address your concerns below.
>
> ---
>
> ### Responses to Weakness 1
>
> We agree that a fair comparison is important. Critique-GRPO performs extra critique-guided refinement by leveraging natural language feedback, whereas the GRPO baseline relies solely on numerical feedback and lacks this refinement process. To address the compute difference, we provided extensive results in Section 3.1 and Appendix E.1 showing the GRPO baseline often plateaus even when scaling data/compute. Furthermore, Critique-GRPO demonstrates significantly better sample efficiency than various GRPO variants using only numerical feedback (Table 3, Section 5.3).
>
> Additionally, we compared Critique-GRPO against a matched sequential baseline: *R1-GRPO + Refinement-SFT (CoT Critique)*. This baseline involves: (1) running GRPO to convergence, (2) generating critiques/refinements for remaining failures, and (3) performing SFT on the filtered refinements. In the table below, Critique-GRPO significantly outperforms this baseline (p < 0.01) on both in-distribution math and out-of-distribution science/general benchmarks. This supports the superiority of online RL optimization with numeric and language feedback.
>
> | Method | MATH-500 | Minerva-MATH | Olympiad-Bench | AMC23 | AIME24 | TheoremQA | GPQA-Diamond | MMLU-Pro | Average |
> | :--- | ---: | ---: | ---: | ---: | ---: | ---: | ---: | ---: | ---: |
> | Qwen-2.5-7B-Base | 60.80 | 20.20 | 30.40 | 35.00 | 13.30 | 21.60 | 28.79 | 46.24 | 32.04 |
> | R1-GRPO | 74.00 | 32.00 | 38.50 | 42.50 | 16.70 | 40.60 | 33.33 | 51.81 | 41.18 |
> | R1-GRPO + Refinement-SFT (Based on CoT Critique) | 75.40 | 33.80 | 40.10 | 47.50 | 20.00 | 41.20 | 34.34 | 52.83 | 43.15 |
> | Critique-GRPO (CoT Critique) | **77.80** | **36.80** | **42.40** | **62.50** | **20.00** | **44.00** | **37.88** | **55.28** | **47.08** |
>
> ### Responses to Weakness 2
>
> We understand this concern. To ensure fair comparisons and ablate the effect of critique quality, we strictly filter critiques: we only use those generated by the reward model that align with our rule-based evaluations. If they do not align, the reward model is prompted to regenerate them (detailed in Appendix G).
>
> Regarding self-critique performance (Table 4, Appendix 5.4, Figure 1b), *Critique-GRPO using self-critiques achieves an average performance of 68.13* (the 10th column in Table 4), yielding a negligible underperformance compared to the GPT-4o CoT critique variant (68.26). Furthermore, Table 12 (Appendix H) shows that Critique-GRPO (using entirely self-generated critiques and self-evaluation, with no external oracle labels) significantly outperforms both SFT (using expert demonstrations) and R1-GRPO (using oracle numerical feedback). This confirms the effectiveness of Critique-GRPO and highlights its potential for self-improvement without external supervision.
>
> ---
>
> ### Responses to Questions
>
> Thank you for these questions. We respond as follows:
>
> 1. Response to Q1: We trigger refinement only when the initial response set yields zero correct solutions to balance training efficiency and performance. Critique-GRPO adds overhead from critique generation and refinement (Appendix B).
>
> 2. Response to Q2: As discussed in Appendix C, we evaluated Critique-GRPO using noisy and adversarial critiques under the guidance of oracle numerical feedback. In these adversarial settings, we observed entropy explosion and subsequent performance degradation. This confirms that the efficacy of Critique-GRPO stems from valid, constructive critique guidance rather than merely the presence of spurious natural language signals.
>
> 3. Response to Q3: As discussed in Appendix C, we hypothesize that the performance decline associated with increasing critique length is due to the model's difficulty in extracting critical information from longer contexts. Open-source models often struggle with long-context understanding and reasoning [1, 2].
>
> 4. Response to Q4:
> Thank you for this insightful question. Our theoretical analysis (Appendix F.4) accounts for two primary scenarios reflecting our empirical setups: (1) The critique identifies the first error step but provides no correction suggestions. This corresponds to standard CoT critiques (e.g., those generated by Llama3.1-405B, which achieved an average score of 46.79 in Table 5). (2) The critique identifies the error and provides valuable correction suggestions. This corresponds to CoT critiques from stronger models (e.g., GPT-4o, which achieved an average score of 47.08 in Table 5).
> Since even strong critique models like GPT-4o only sometimes provide correction suggestions in practice, our paper considers both scenarios. Our analysis demonstrates that critique-guided exploration yields better efficiency than numerical-feedback-guided exploration under both conditions.
>
> We sincerely appreciate your efforts and hope these responses address your concerns.
>
> **References:**
> [1] *Qwen2.5 Technical Report*
>
> [2] *Qwen3 Technical Report*

---

> > ### Author Rebuttal · Reviewer_5cbv · 2026-04-02
> >
> > I thank the authors for their response. After going over their response and the paper again, I maintain my opinion that this is a good paper and should be accepted.

---

### Official Review · Reviewer_ioUp · 2026-03-13

**Soundness:** 3
**Presentation:** 3
**Significance:** 3
**Originality:** 2
**Overall Recommendation:** 4
**Confidence:** 4

**Summary:**

The paper identified the key limitations of RL with numerical feedback with empirical results, including the performance saturation after longer RL steps or longer prompts, the limitation of LLM’s self-reflection abilities via thinking, and the consistent failure in certain problems. These limitations of RL with numerical feedback clearly motivates their design of critique-GRPO, where they sample refined responses from critique generated by external reward systems and do policy optimization over both initial and refined responses. Their extensive experiments of critique-GRPO validates that natural language feedback improves upon numerical feedback in math tasks, and Science & General tasks (an out-of-domain) example, showcasing consistent improvements over GRPO, SFT, and offline refinement baselines.

**Compliance With Llm Reviewing Policy:**

Affirmed.

**Key Questions For Authors:**

1. How does performance change when using weaker critique models, such as smaller open-source models that have not been post-trained for critiquing?

2. Can you share the compute cost of critique-GRPO compared to critique-free baselines? I am interested in whether the improvements are worth the extra cost, or if training R1-GRPO longer with the same compute would achieve similar results.

3. For Proposition 4.1, do you have empirical evidence showing that the complexity reduction actually occurs?

4. Have you tried a sequential baseline, such as training GRPO to convergence first, then generating critiques + refinements and SFT on the successful ones?

**Limitations:**

Yes

**Strengths And Weaknesses:**

Strengths:

1. The experiment coverage is quite comprehensive. For example, they cover a wide range of methodology baselines (SFT, RAFT, refinement FT, etc.), and two variations of GRPO implementations. Their Critique-GRPO method achieves consistently improved performance over baselines on the math benchmarks and good generalization over OOD tasks. Their ablations on different critic model types, objective modifications, and different model sizes are very informative and thorough. The conclusions and analyses are technically sound and thorough.

2. The overall narrative is good. The three limitations of numerical feedbacks clearly motivates the methodology. The methodology, experiment results, and analyses are easy to follow.

3. This paper addresses the important problem of learning from textual feedback in RL for LLMs. They not only show the critique injection in refined response sampling is crucial, but also show the potential of self-critiquing without external critique model (consistent improvements of all pass@k). These are practically significant especially for scaling up RL in test time.

Weaknesses:
1. The method is fundamentally dependent on the critique model capability, and the advantages/disadvantages of both reward systems are not fully discussed. For rule-based critiques, it is unclear what additional information the binary signal + textual label gives beyond a scalar reward. For model-based critiques, it requires a good critic model to gain meaningful CoT critiques and reliable reward assignment given the CoT. Even though Table 5 shows meaningful variance across critique models, but they did not analyze how critique quality degrades with weaker models (like open-sourced smaller models not post-trained on critiques). Plus, they did not report the computational cost of critique generation at training time, which should be part of the comparison with critique-free methods.

2. Proposition 4.1 is disconnected from your method implementation and not being discussed in the empirical analyses. According to your Proposition 4.1, the CoT critique feedback can reduce the complexity to O(|S|L), under the assumption that the critique model pinpoints which step is wrong and provides the correct replacement for that step independent of other steps. But in your Step 2 in Section 4.2, it performs whole-sequence regeneration conditioned on the critique, not editing individual steps as claimed in Proposition 4.1, so the complexity reduction never happens in reality. And there is no empirical discussion of the claimed complexity reduction in the results section. The gap between the two is large so I'm not convinced of the true theory behind your proposed method.

3. The paper proposes the joint optimization of both the responses and the refinement, but didn't test whether this joint integration is actually necessary. A natural and simpler baseline would be: (1) run standard GRPO with scalar reward to convergence, (2) then pick the problems the RL-trained model still fails, generate critiques and refinements, and do SFT after filtering those refinements. This baseline uses the same ingredients, but combines them sequentially rather than jointly.

4. The method you proposed contains critique-guided refinement, policy shaping, training on initial + refined responses, are mostly known ideas. The combination is reasonable but not surprising. The theoretical contribution (Proposition 4.1) could be the most novel element but is too disconnected from the true algorithm. But this paper can benefit from discussing what is genuinely new beyond the combining these pieces within GRPO.

---

> ### Author Rebuttal · Authors · 2026-03-29
>
> Thank you for your feedback. We provide the following responses to address your concerns:
>
> ---
>
> ## Response to Weakness 1
>
> ### Response to Weakness 1.1
> We agree that critique granularity matters. We analyze this effect theoretically using the Transfer Eluder Dimension framework [1] (Section 4.2; Appendix F.4) and empirically (Section 3.2; Appendix E.2). Our results show that adding a textual label—i.e., an explicit indication of whether the model’s answer is correct or incorrect—helps the model leverage critiques (even very coarse ones) to find correct solutions more efficiently than relying on scalar rewards alone.
>
> ### Response to Weakness 1.2
> We agree that critique quality is critical. To control critique reliability, we only use critiques generated by the reward model when they align with the same rule-based evaluations; otherwise, we prompt the reward model to regenerate the critique (detailed in Appendix G).
>
> We also evaluate critique quality across critique sources. In Section 5.4, we report results using self-critiques from Qwen3-8B (not post-trained on critiques), which shows negligible underperformance relative to GPT-4o critiques (Figure 1b). In addition, Section 5.5 (Table 5) reports Critique-GRPO results with two additional critique models, enabling a direct comparison of how critique model quality impacts final performance.
>
> ### Response to Weakness 1.3
> We agree training efficiency should be clarified. We provide a detailed training-time comparison between R1-GRPO (critique-free) and Critique-GRPO on both Qwen2.5-7B-Base (non-reasoning) and Qwen3-8B (reasoning) in Appendix B. We report total wall-clock time as GPU time; importantly, this total includes the additional critique-generation cost. We will state this more explicitly in the revision.
>
> ---
>
> ## Response to Weakness 2
> We appreciate this concern. Due to space limits, the main paper only summarizes the theoretical intuition in Section 4; the full theoretical analysis (Transfer Eluder Dimension framework [1]) is provided in Appendix F.4. Following [1], we model refinement as the policy revising specific steps during generation while conditioning on the critique in context (which may identify the erroneous step and/or provide an improvement suggestion).
>
> ---
>
> ## Response to Weakness 3
> Thanks for raising the question. We compare against the sequential baseline *R1-GRPO + Refinement-SFT (CoT critique)*: (1) run GRPO to convergence; (2) critique/refine remaining failures and SFT on filtered refinements. Critique-GRPO significantly outperforms this baseline (p < 0.01) on both in-distribution math and OOD science/general benchmarks, supporting *online* joint optimization with numeric + language feedback. This aligns with prior evidence that online RL can be more effective for policy optimization [2] and that SFT may hurt OOD generalization [3].
>
> | Method | MATH-500 | Minerva-MATH | Olympiad-Bench | AMC23 | AIME24 | TheoremQA | GPQA-Diamond | MMLU-Pro | Average |
> |---|---:|---:|---:|---:|---:|---:|---:|---:|---:|
> | Qwen-2.5-7B-Base | 60.80 | 20.20 | 30.40 | 35.00 | 13.30 | 21.60 | 28.79 | 46.24 | 32.04 |
> | R1-GRPO | 74.00 | 32.00 | 38.50 | 42.50 | 16.70 | 40.60 | 33.33 | 51.81 | 41.18 |
> | R1-GRPO + Refinement-SFT (Based on CoT Critique) | 75.40 | 33.80 | 40.10 | 47.50 | 20.00 | 41.20 | 34.34 | 52.83 | 43.15 |
> | Critique-GRPO (CoT Critique) | **77.80** | **36.80** | **42.40** | **62.50** | **20.00** | **44.00** | **37.88** | **55.28** | **47.08** |
>
> ---
>
> ## Response to Weakness 4
> We appreciate this concern. Our goal is to motivate and formalize online RL with richer feedback (natural language critiques) rather than scalar rewards alone. Implementing Critique-GRPO is non-trivial due to refinement-induced distribution shift (detailed in Section 4); we propose practical strategies (Section 4) and support them theoretically (Appendix F.4) and empirically (Sections 3–5).
>
> ---
>
> ## Responses to Questions
>
> Thank you for these questions. We respond as follows:
> 1. Response to Q1: We provide extensive results and discussion in Sections 5.4 and 5.5; see our response to Weakness 1.2.
> 2. Response to Q2: We report R1-GRPO results with longer training time and more training data (Section 3.1; Appendix E.1). These results suggest that RL with only numerical feedback often plateaus even when scaling data and compute.
> 3. Response to Q3: We provide empirical evidence of gains from critique-based self-refinement applied to plateaued scalar-reward RL across Qwen2.5-7B-Base, Qwen3-8B, and Qwen3-8B-Base (Section 3.2; Appendices E.2–E.4).
> 4. Response to Q4: Critique-GRPO significantly outperforms the requested sequential baseline (see Weakness 3-4).
>
> ---
> We sincerely appreciate your efforts and hope these responses address your concerns.
>
> **References:**
> [1] *Provably Learning from Language Feedback*
> [2] *Bridging Offline and Online Reinforcement Learning for LLMs*
> [3] *On the Generalization of SFT: A Reinforcement Learning Perspective with Reward Rectification*

---

> > ### Author Rebuttal · Reviewer_ioUp · 2026-04-04
> >
> > Thank the authors for the responses. They solved my concerns and I raise my score 4 -> 5

---

> > > ### Author Response · Authors · 2026-04-04
> > >
> > > Thank you again for your thoughtful reply and for your insightful comments throughout the review process. We are very glad to see that our rebuttal has addressed your concerns and clarified the main questions you raised.
> > >
> > > We also truly appreciate your positive feedback and your indication that you would be willing to **raise the score to 5**. We noticed that the score in the initial review may not yet have been updated, so we just wanted to kindly follow up in case this was simply overlooked.
> > >
> > > As the deadline for acknowledging author responses is approaching (April 3 AoE), we would be very grateful if you could kindly update **the 'Overall Recommendation' section** in your original review when convenient.
> > >
> > > Thank you once again for your time, constructive suggestions, and support of our work.

---

### Decision · Program_Chairs · 2026-04-30

**Decision:**

Accept (spotlight)

**Comment:**

This paper proposes Critique-GRPO, integrating natural language critiques into the GRPO training loop for LLM reasoning. The core observation is that purely numerical feedback leads to performance plateaus and persistent failures on a significant fraction of training problems where the model cannot self-correct from reward signal alone. The proposed solution generates critiques identifying specific errors in failed rollouts and uses them to guide refinement trajectories that re-enter the GRPO optimization. The experimental evaluation covers eight benchmarks across multiple model families (Qwen, Llama) and scales, and the self-critique variant where the policy model generates its own critiques achieves near-parity with GPT-4o critiques, which is an important result as it shows the framework does not require an external oracle. Concerns about comparison fairness with respect to the sequential alternative (GRPO followed by Refinement-SFT) were resolved in the rebuttal, where the integrated approach significantly outperformed the sequential baseline at marginal additional compute. I recommend acceptance. Authors should incorporate the sequential baseline comparison, weak-to-strong supervision results, and AIME 2025 numbers into the camera-ready version.